# Localizing Task Recognition and Task Learning in In-Context Learning via Attention Head Analysis

**Haolin Yang**[1]           **Hakaze Cho**[3,2,4]                    **Naoya Inoue**[3,4]
[1]University of Chicago        [2]Tohoku University        [3]RIKEN        [4]JAIST

## Abstract

We investigate the mechanistic underpinnings of in-context learning (ICL) in large language models by reconciling two dominant perspectives: the component-level analysis of attention heads and the holistic decomposition of ICL into **T**ask **R**ecognition (TR) and **T**ask **L**earning (TL). We propose a novel framework based on **T**ask **S**ubspace **L**ogit **A**ttribution (TSLA) to identify attention heads specialized in TR and TL, and demonstrate their distinct yet complementary roles. Through correlation analysis, ablation studies, and input perturbations, we show that the identified TR and TL heads independently and effectively capture the TR and TL components of ICL. Via steering experiments with geometric analysis of hidden states, we reveal that TR heads promote task recognition by aligning hidden states with the task subspace, while TL heads rotate hidden states within the subspace toward the correct label to facilitate prediction. We further show how previous findings on ICL's mechanism—including induction heads, task vectors, and more—can be reconciled with our attention-head-level analysis of the TR–TL decomposition. Our framework thus provides a unified and interpretable account of how LLMs execute ICL across diverse tasks and settings[1].

## 1 Introduction

A key property of **L**arge **L**anguage **M**odels (LLMs) is their ability to solve tasks from demonstrations embedded in the input—without further training. This phenomenon, known as **In-c**ontext **L**earning (ICL) (Brown et al., 2020; Radford et al., 2019), has reduced the need for large datasets and finetuning, enabling fast adaptation of LLMs to new tasks (Dong et al., 2024; Sun et al., 2022). Since its success cannot be explained by traditional gradient-based paradigms (Ren et al., 2024b), deciphering the mechanism behind ICL has become a central research question of great academic interest.

Two research paradigms dominate this pursuit. **(1)** The introspective paradigm designates internal model components or representations as critical drivers of ICL functionality. Pioneering works (Elhage et al., 2021; Olsson et al., 2022) formulate the output logits of Transformers as the sum of individual component outputs and highlight the significance of **Induction Heads (IHs)** in toy models, with follow-ups confirming their importance in larger models via ablation (Crosbie & Shutova, 2024; Halawi et al., 2024; Cho et al., 2025a). These studies inspired the concept of **task vectors**—compact representations distilled from hidden states or attention head outputs that steer zero-shot prompts toward ICL-level predictions (Hendel et al., 2023; Todd et al., 2024; Liu et al., 2024), and spurred further inquiries into the properties, behaviors, and emergence of IHs (Ren et al., 2024b; Singh et al., 2024; Yin & Steinhardt, 2025). **(2)** The holistic paradigm instead treats the LLM as an entirety and investigates ICL's properties by directly inspecting and probing how different demonstration configurations shape ICL performance. For instance, by perturbing the demonstration labels in context, Pan et al. (2023) factorize ICL into two core components: **Task Recognition (TR, recognizing the label space)** and **Task Learning (TL, learning the text–label mapping)**, each contributing to part of the ICL functionality (Figure 1 (A)). Min et al. (2022) also systematically explored the effect of the distribution of texts and labels in demonstrations individually, as well as the templates and number of demonstrations.

---

[1]Code Implementation: https://github.com/HLYang2001/Localizing_TR_TL

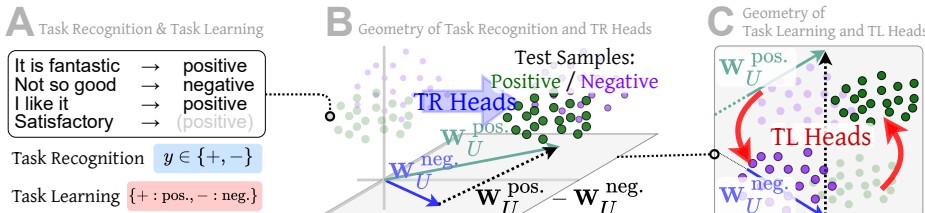

Figure 1: **(A)** Example of how LLMs deduce the label of a final query through ICL, which consists of two components: task recognition (identifying the label space) and task learning (mapping demonstration texts to labels). **(B)** The outputs of Task Recognition heads align with the task subspace spanned by candidate label unembeddings, thus they can steer the hidden states to align with the subspace by reducing the angle between the point clouds and the task subspace. **(C)** Task Learning heads act as rotations within the task subspace, aligning the query's hidden state with the unembedding of the correct label and enabling the correct prediction.

The two research paradigms offer complementary insights but also limitations. The introspective paradigm localizes ICL to individual attention heads, yet its reliance on ablation only measures **how much** performance changes when heads are removed, without explaining **how** these heads realize ICL or behave under varied inputs. The holistic paradigm provides a broad functional view, separating ICL into TR and TL, but cannot trace these roles back to concrete components. A unified framework is needed to combine mechanistic precision with functional clarity.

Therefore, we propose a TR/TL decomposition via attention-head analysis (Figure 1). **T**ask **S**ubspace **L**ogit **A**ttribution (TSLA) quantifies each head's effect on hidden states relative to task-label unembeddings, identifying **TR heads** and **TL heads** as ICL's two drivers. Geometric analyses show that TR heads align hidden states to the task-label subspace, enabling label recognition and suppressing irrelevant tokens (Figure 1 (B)), while TL heads rotate states within it toward the correct label, sharpening prediction (Figure 1 (C)). Together they implement ICL functionality.

We validate the framework using correlation analyses, ablations, input perturbations, and steering across ICL settings, including corrupted demonstrations and free-form generation. These results reconcile prior findings: IHs emerge as a subset of TR heads for label-space recognition, and zero-shot performance is traced to poor task-subspace alignment, explaining why TR-head outputs (hence IHs) form effective task vectors that restore alignment (Todd et al., 2024).

The three core contributions of this paper are:

1. We derive TSLA to identify TR and TL heads, localizing the two ICL components from input-perturbation studies to concrete attention heads.
2. We use correlation/overlap analyses and TR/TL-channel ablations to show separable control of task recognition vs. task learning, unifying ICL observations (e.g., IHs) in the TR/TL framework.
3. We perform steering and geometric analyses to explain how TR heads drive task-subspace alignment and TL heads drive within-subspace rotation toward labels in classification and generation.

## 2 RELATED WORKS

**TR & TL decomposition of ICL** Early ICL work relied on input-perturbation experiments (Min et al., 2022; Pan et al., 2023; Wei et al., 2023). By removing semantic labels (e.g., "negative" → "0") or scrambling the text–label mapping while still observing non-trivial accuracy, these works concluded that ICL comprises two components: Task Recognition (TR) and Task Learning (TL). However, this approach cannot provide mechanistic explanations or tie ICL functionality to model components. We bridge this gap with our theoretically derived TSLA method based on task-subspace geometry, identifying critical TR and TL heads for the two ICL components (Subsection 3.2).

**Circuit formulation of Transformer** Our TSLA method builds upon the circuit formulation of Transformers (Elhage et al., 2021) by reframing head contributions in task-subspace geometry, which decomposes output logits into contributions from attention heads and MLPs. This formulation has enabled precise attribution of LLM behaviors to individual components (Crosbie & Shutova, 2024) and spotlighted the Induction Heads (IH) as crucial for ICL (Song et al., 2025). In toy copying tasks

([X][Y][X][Y]...[X] → [Y]), IHs attend to earlier occurrences of [Y], mimicking label copying. Their importance for ICL has been confirmed in large-scale models through ablation studies that replace or remove their outputs and observe changes in the final logits (Cho et al., 2025a). In this work, we demonstrate how the significance of IHs fits into the TR & TL decomposition of ICL via extensive correlation analyses and investigations of the heads' distributional properties Subsection 4.1.

**Ablation of special attention heads**  The circuit formulation motivates identifying special heads by ablating their outputs and inspecting prediction changes. Examples include Function Vector Heads (Yin & Steinhardt, 2025)—heads whose ablation most harms ground-truth label logits and are thus viewed as causes of ICL functionality (Sun et al., 2025). Similar approaches extend to retrieval-augmented generation (Jin et al., 2024; Kahardipraja et al., 2025) and chain-of-thought reasoning (Cabannes et al., 2024). Yet they reveal only **how much** a head contributes, not **how** or **which** ICL component it affects; heatmap-based explanations remain too shallow for causal claims (Kahardipraja et al., 2025; Ren et al., 2024a). We address this by new TR/TL-channel metrics and fine-grained ablations that expose component-specific effects Subsection 4.2.

**Task Vector steering and geometric analysis**  An alternative to ablation is to aggregate head outputs into task vectors that steer zero-shot states to ICL-level accuracy (Hendel et al., 2023), with effective vectors viewed as ICL's mechanistic origins (Todd et al., 2024). However, interpretability issues remain: **(1)** candidates are chosen via ablation, inheriting its limitations; **(2)** effects are measured only at outputs, leaving their role in computation unclear (Merullo et al., 2024). Geometric analyses of layer-wise hidden-state evolution, revealing how task vectors reshape geometry, offer a promising alternative (Kirsanov et al., 2025). Jiang et al. (2025) identify a compress–expand pattern in task representations during ICL, while Yang et al. (2025) link IH outputs to alignment between hidden states and unembedding vectors of task-relevant labels. This perspective elucidates their influence on outputs. We perform a comprehensive analysis of the distinct geometric effects that task vectors constructed from TL and TR heads have on the model's hidden states Subsection 4.3.

## 3   METHODOLOGY

### 3.1   BACKGROUND

**Circuit formulation of Transformer**  In the circuit formulation of Transformer LLMs, an input query $q$ with $N$ tokens $[x_1, ..., x_N]$ (e.g., "I like this movie. Sentiment:" for a sentiment analysis task) is first transformed into layer-0 hidden states $h_1^0, ..., h_N^0$ via the embedding matrix $W_E \in \mathbb{R}^{|\mathbb{V}| \times d}$, where $d$ is the model dimension and $\mathbb{V}$ the vocabulary. These hidden states then pass through $L$ layers, where the update of the $i$-th token's hidden state at layer $l$ is:

$$h_i^l = h_i^{l-1} + \sum_{k=1}^{K} a_{i,k}^l + m_i^l,$$

with $a_{i,k}^l$ the output of the $k$-th attention head (denoted head $(l,k)$) in the attention sublayer, and $m_i^l$ the MLP sublayer output. $a_{i,k}^l$ is the weighted sum of the layer-$(l-1)$ hidden states of the first $i$ tokens, $H_{\leq i}^{l-1} = [h_j^{l-1}]_{j=1}^i$, transformed by the embedding matrices of head $(l,k)$. The final hidden state of the last token (":" in the previous example) can thus be written as:

$$h_N^L = h_N^0 + \sum_{l=1}^{L} \left( \sum_{k=1}^{K} a_{N,k}^l + m_N^l \right). \tag{1}$$

$h_N^L$ is multiplied by the unembedding matrix $W_U \in \mathbb{R}^{|\mathbb{V}| \times d}$ to form logits. Each head output thus contributes to the logits additively as $W_U a_{N,k}^l$, referred to as **Direct Logit Attribution (DLA)** (Olsson et al., 2022; Chughtai et al., 2024; Yu & Ananiadou, 2024; Lieberum et al., 2023).

**ICL and Induction Head**  In ICL, $m$ text-label demonstration pairs $t_1, y_1, ..., t_m, y_m$ are prepended to the query, forming the sequence $t_1, y_1, ..., t_m, y_m, q$ (e.g., "I hate this movie. Sentiment: negative. This movie is great. Sentiment: positive··· I like this movie. Sentiment:"). With these demonstrations, the attention head outputs $a_{N,k}^l$ to the final position, depending on all preceding tokens, producing logits that can lead the LLM to predict $y^*$, the correct label for $q$. An Induction Head (IH) is a special attention head that, at each position, searches for earlier occurrences of the current token, attends to

the immediately following tokens, and copies their information back to the current position. In the example above, an IH at the final position places its attention on the "positive" and "negative" tokens that follow previous ":" tokens and uses their hidden states to form its output.

## 3.2 Identifying TR and TL heads using Task Subspace Logit Attribution

Pan et al. (2023) decomposes ICL into two components. Task Recognition (TR) means recognizing the set of candidate task label tokens $\mathbb{Y}$ from demonstration labels, with $\{y_1, ..., y_m\} \in \mathbb{Y}$, without using the text-label mapping information to deduce the correct token. Task Learning (TL), in contrast, means learning the mapping from demonstration texts to task labels, $f : \mathbb{X} \to \mathbb{Y}$, to predict the only correct label for query $q$. To identify heads contributing to TR and TL, Lieberum et al. (2023) compute $W_U^{\mathbb{Y}} a_{N,k}^l$ and $a_{N,k}^{l,\top} W_U^{y^*}$ for all heads $(l,k)$, where $W_U^{\mathbb{Y}} \in \mathbb{R}^{|\mathbb{Y}| \times d}$ and $W_U^{y^*} \in \mathbb{R}^d$ are the unembedding matrix restricted to $\mathbb{Y}$ and $y^*$. Heads with the highest element-wise sum $\mathbf{1}^\top W_U^{\mathbb{Y}} a_{N,k}^l$ are considered TR heads, and those with the highest $a_{N,k}^{l,\top} W_U^{y^*}$ are TL heads.

This approach has two problems. **(1)** For TR heads, Lieberum et al. (2023) study four-choice tasks where the full label space is "A", "B", "C", "D". In general settings, demonstration labels are arbitrarily chosen and may not capture the full task semantics. Changing labels from positive/negative to favourable/unfavourable for sentiment analysis prompts does not alter the task, but heads amplifying logits for positive/negative may not do so for favourable/unfavourable. **(2)** For TL heads, the method ignores competition among label tokens: heads boosting $y^*$ may also boost incorrect labels $\mathbb{Y} \setminus \{y^*\}$, disqualifying them as true task-mapping heads. A more precise approach must (a) capture task semantics beyond surface tokens and (b) evaluate contributions relative to competing labels.

We therefore propose the **Task Subspace Logit Attribution (TSLA)** method. For TR heads, we compute the TR score:

$$\|\text{Proj}_{W_U^{\mathbb{Y}}} a_{N,k}^l\|_2, \tag{2}$$

where $\text{Proj}_{W_U^{\mathbb{Y}}} = W_U^{\mathbb{Y}} (W_U^{\mathbb{Y},\top} W_U^{\mathbb{Y}})^{-1} W_U^{\mathbb{Y},\top}$ is the $d \times d$ projection matrix onto $\text{span}(W_U^{\mathbb{Y}})$, the subspace spanned by unembedding vectors of demonstration labels. This subspace should capture the unembeddings of all tokens that could potentially serve as demonstration labels given the finding that LLMs encode inter-related semantics into subspaces (Saglam et al., 2025; Zhao et al., 2025) (further verified in Appendix F). Intuitively, TR heads are those whose outputs concentrate within this subspace rather than elsewhere, which is precisely what the TR score is designed to capture. We also have the following theoretical guarantee for this metric's effectiveness.

**Theorem 1** *Let $r = |\mathbb{Y}|$. Assume $n$ distinct $r$-dimensional subspaces drawn i.i.d. from the Grassmannian $Gr(r,d)$ are spanned by columns of $W_U$. If head $(l,k)$ has TR score $\gamma$, then with probability at least $1 - (n-1)(1 - I_{(\frac{\gamma}{\|a_{N,k}^l\|_2})^2}(\frac{r}{2}, \frac{d-r}{2}))$, $a_{N,k}^l$ has the largest projected $l_2$ norm onto $\text{span}(W_U^{\mathbb{Y}})$ among all such subspaces,*

where $I_x(\alpha, \beta) = \frac{B(x;\alpha,\beta)}{B(\alpha,\beta)}$ is the regularized incomplete beta function, monotone in $x$. Thus, a large TR score implies the head output with high probability projects onto the task-related subspace more than any other subspace, thereby qualifying it as a TR head. This formalizes the intuition above and explains why the metric identifies the heads advancing task recognition. The proof is in Appendix B.

For TL heads, we compute:

$$\frac{\text{Ave}_{y' \in \mathbb{Y}/\{y^*\}}(a_{N,k}^{l,\top}(W_U^{y^*} - W_U^{y'}))}{\|\text{Proj}_{W_U^{\mathbb{Y}}} a_{N,k}^l\|_2}. \tag{3}$$

The numerator measures the alignment between the head output and the average correct–incorrect label direction, reflecting the logit gap a head induces between the correct label and competing labels. The denominator is the TR score. Since $W_U^{y^*} - W_U^{y'} \in \text{span}(W_U^{\mathbb{Y}})$ for all $y' \in \mathbb{Y} \setminus \{y^*\}$, the TL score has a natural geometric interpretation with respect to the task subspace: it measures the component of the head output, after projection onto the task subspace, that lies along the average correct–incorrect label direction. Thus, the TL score isolates the contrastive within-subspace direction that favors $y^*$

over other competitor labels. Heads with high TL scores enlarge the correct–incorrect logit gap, effectively rotating hidden states within the task subspace toward the correct label and away from others (see Figure 1 (C)). This conforms to task learning, which identifies the correct label and excludes incorrect ones for an input. We provide further clues on this mechanism in Subsection 4.1: TL heads allocate more attention to query tokens, suggesting they absorb query semantics and match it to the label tokens to select the correct in-context mapping. Moreover, this TL score mitigates the DLA issue of interference from incorrect labels by disregarding heads that fail to differentiate and instead raise both logits. To verify this, we provide the detailed comparison between TSLA and DLA in Appendix C by demonstrating that TSLA solves the practical limitations of DLA as the theory predicts. For each dataset, we use ICL prompts from the first 50 queries to compute TR and TL scores for each head with TSLA, summing across prompts. Heads are ranked by these scores to identify TR and TL heads.

Having identified TR and TL heads with TSLA, we next ask whether these heads behave in the mechanistic ways TSLA claims. In particular, TSLA predicts (i) TR and TL heads should affect different ICL components in a separable manner, and (ii) their outputs should drive qualitatively different geometric changes in hidden states. Accordingly, Subsection 4.1 examines TR/TL specialization and their relation to IHs, Subsection 4.2 tests separability causally via TR/TL-channel ablations, and Subsection 4.3 probes the predicted geometric mechanisms through steering and layerwise analysis.

## 4 EXPERIMENTS

**Models**  We experiment on models with diverse architectures and sizes, including Llama3-8B, Llama3.1-8B, Llama3.2-3B (Grattafiori et al., 2024), Qwen2-7B, Qwen2.5-32B (Yang et al., 2024), and Yi-34B (01. AI et al., 2024). Unless otherwise noted, results are reported on Llama3-8B.

**Datasets**  We evaluate on the following datasets: SUBJ (Wang & Manning, 2012), SST-2 (Socher et al., 2013), TREC (Li & Roth, 2002), MR (Pang & Lee, 2005), SNLI (MacCartney & Manning, 2008), RTE (Dagan et al., 2005), and CB (De Marneffe et al., 2019). We further include an LLM-generated dataset introduced in Subsection 4.3, with curation details in Appendix D and Appendix K. We also experiment with the SubjQA dataset (Bjerva et al., 2020) in Appendix I.3.

**ICL setting**  We use 8-shot demonstrations for ICL. For implementation details (models, datasets, prompt templates, etc.), see Appendix D.

### 4.1 VALIDATING TR/TL HEAD SPECIALIZATION AND THE ROLE OF INDUCTION HEADS

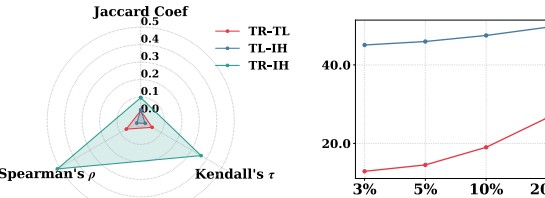

(a) Dataset-averaged Jaccard Coefficient, Kendall's $\tau$, and Spearman's $\rho$ for TR heads, TL heads, and IHs at the top 3% level.

(b) Conditional Mean Percentage at four top thresholds for the TR–IH and TL–IH pairs averaged over datasets.

Guided by TSLA, we first test whether the identified TR and TL heads exhibit the predicted specialization and how they relate to previously studied induction heads. Specifically, we analyze the overlap, rank correlation, and consistency between TR and TL head rankings, and include IHs as a reference point. Given the significance of IHs in mechanistic accounts of ICL (Zheng et al., 2024), we also ask whether IHs contribute primarily through TR, TL, or both. Following Todd et al. (2024) and Yang et al. (2025), we compute IH scores (i.e., the degree to which a head's attention pattern resembles that of IHs) as described in Appendix E.

Figure 2: Overlap, correlation, and consistency of three attention head types averaged across datasets. **(A)** TR heads exhibit substantially greater overlap and correlation with IHs compared to TR–TL or TL–IH pairs. **(B)** Top IHs consistently rank higher in the TR ranking than in the TL ranking. Results for other models are in Appendix G.2.

Adopting the methodology of Yin & Steinhardt (2025), we report Jaccard Coefficients[2] among the top 3% (further ablation studies for the choice of the threshold percentage are in Appendix G.1) of

---

[2]For two subsets $\mathbb{A}_1, \mathbb{A}_2 \subseteq \mathbb{A}$, the Jaccard Coefficient is $\frac{|\mathbb{A}_1 \cap \mathbb{A}_2|}{|\mathbb{A}_1 \cup \mathbb{A}_2|}$.

each head type to measure overlap at the top level. We also compute Kendall's $\tau$ and Spearman's $\rho$ among the three rankings to evaluate the global correlations levels among the head types. Finally, we introduce **Conditional Mean Percentage**, which measures the average rank percentile of the top 3%, 5%, 10%, and 20% IHs within the TR and TL rankings. This metric bridges the local and global perspectives and answers the question of how strongly, on average, the top IHs exhibit TR vs. TL functionality, which is important in subsequent ablation-based experiments in Subsection 4.2.

**Strong association between IHs and TR heads**
Figure 2a shows that TR heads and IHs are highly similar: **(1)** their overlap at the top 3% is much larger than either TR–TL or TL–IH pairs, and significantly above random baseline[3]; **(2)** their rank correlations are consistently higher. By contrast, TR–TL and TL–IH pairs show weaker overlap and correlation. This reveals how IHs, the magnitude of whose influence on ICL has been widely recorded, affect ICL: they influence ICL mainly by enabling recognition of the label space, rather than selectively amplifying the correct label. It reconciles conflicting previous findings with some reporting IHs as recognizing correct labels (Olsson et al., 2022; Cho et al., 2025b), others mentioning "false induction heads" that mislead (Halawi et al., 2024; Yu & Ananiadou, 2024). It also echoes Yin & Steinhardt (2025), who observed IHs overlap strongly with "function vector heads"[4], reinforcing the centrality of TR heads in ICL.

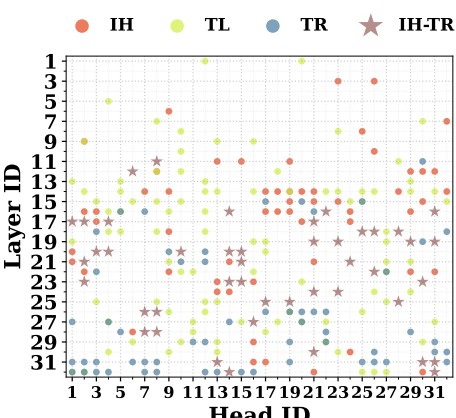

Figure 3: Distribution of the top 10% TR heads, TL heads, and IHs on SST-2. TR heads occur significantly deeper than TL heads and IHs. Overlaps between TR heads and IHs are also more frequent in deeper layers. Results for other models are in Appendix G.3.

Figure 2b further shows that top IHs consistently rank higher within TR rankings than TL rankings. For example, the top 10% IHs correspond to roughly the top 20% TR heads but only the top 50% TL heads. This further justifies the large accuracy implications of ablating top IHs, which would imply ablating fairly high-ranking TR heads and the failure of task recognition.

**Layer-wise distribution of special heads** Figure 3 shows the per-layer distribution of the top 10% TR, TL, and IH heads for SST-2. We observe: **(1)** TR heads appear significantly deeper than both TL heads and IHs, while the layer distributions of TL heads and IHs are more similar (see Appendix G.3 for significance tests) (We provide further explanations and experimental validations of why TR heads occur in deeper layers than TL heads in Appendix G.7). This partly echoes but also challenges Yin & Steinhardt (2025), who reported function vector heads as only "slightly deeper" than IHs. **(2)** The TR–IH overlaps are much greater than TL–IH or TR–TL overlaps,

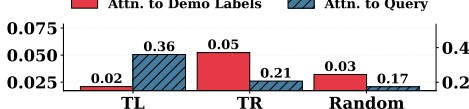

Figure 4: Average attention-weight distribution of top TL and TR heads. TR heads attend more to demonstration labels—supporting task recognition—whereas TL heads focus more on the query to extract semantics for correct prediction. Results for other models are in Appendix G.5.

and occur primarily in deeper layers. This indicates that the correlation between TR heads and IHs is systematic rather than haphazard: the overlaps conform to the general trend of TR heads being concentrated in deeper layers, instead of reflecting coincidental matches with scattered TR heads that occasionally appear in early layers.

**Attention distribution of TL and TR heads** A second property of the identified heads is their attention distribution, which reveals how they operationalize TL and TR. We quantify this using two metrics: **(1)** the total attention weight assigned to demonstration-label tokens, reflecting how much a head incorporates information about the task label space (supporting TR); and **(2)** the total attention weight assigned to query tokens, reflecting how much a head integrates the semantic content of the query (supporting TL). As shown in Figure 4, the top 3% TR heads allocate substantially more attention to demonstration labels than TL or random heads (the small magnitude is expected

---

[3]For two random $k\%$ subsets, the expected Jaccard Coefficient is asymptotically $\frac{k}{200-k}$, or 0.0152 for $k=3$.

[4]Heads revealed to have greatest impact on correct label logits through ablations.

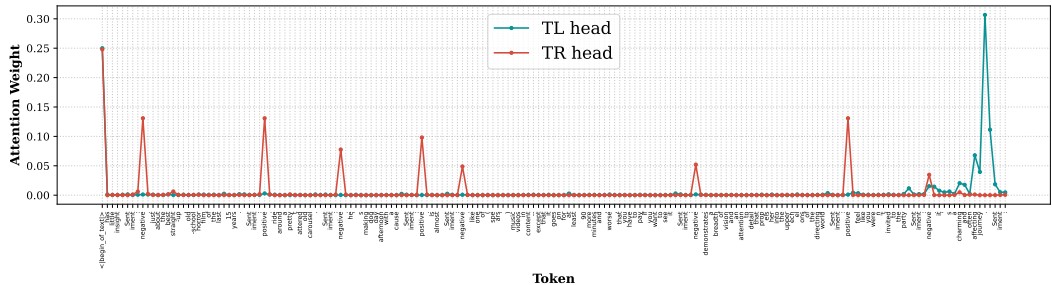

Figure 5: Attention distribution of the top 1 TL and TR head identified on SST-2 over an ICL prompt.

given the scarcity of label tokens in long ICL contexts), indicating that TR heads broadly summarize task-label information. In contrast, TL heads exhibit a far more local attention pattern, attending strongly to the query tokens to extract their semantic meaning and support correct label inference. Direct visualization (Figure 5) of the top SST-2 TL and TR heads on an SST-2 ICL prompt using the first query in the test set further highlight this contrast: TR heads focus on demonstration labels, whereas TL heads primarily attend to the queries. See Appendix G.6 for results using more queries.

**Cross-dataset correlation** To examine whether TR, TL, and IH heads generalize across tasks, we measure Jaccard, Kendall's $\tau$, and Spearman's $\rho$ among the top 3% heads on seven datasets, averaged across $\binom{7}{2} = 21$ dataset pairs. As shown in Figure 6, TR heads and IHs exhibit higher cross-task overlap and correlation than TL heads. *This underscores TR heads (and IHs) as task-invariant mechanistic foundations for label-space recognition, upon which TL heads specialize to learn dataset-specific mappings.* We further explore this discovery in Appendix H.4.

## 4.2 FINE-GRAINED ABLATION TESTS OF TR/TL SEPARABILITY

Having established behavioral specialization in Subsection 4.1, we now provide causal, fine-grained validation. If TR and TL heads realize distinct ICL components, selectively ablating them should yield different signatures for task recognition versus task learning. Accordingly, we ablate TSLA-identified TR and TL heads to test separable contributions. Prior work mainly evaluated ablations by overall ICL-accuracy drop (Crosbie & Shutova, 2024), which indicates performance change but obscures which ICL component is disrupted. To disentangle these effects, we introduce the **Task Recognition Ratio (TR ratio)**, defined as the proportion of predictions within the in-context label set. Formally, for $m$ ICL prompts with predicted labels $\hat{y}_1, ..., \hat{y}_m$ ,

$$\text{TR ratio} = \frac{1}{m} \sum_{i=1}^{m} \mathbf{1}_{\hat{y}_i \in \mathbb{Y}}.$$

Since accuracy is upper-bounded by the TR ratio, the two metrics together let us separately evaluate contributions of TR and TL heads. We conjecture: (1) ablating TR heads should reduce both accuracy and TR ratio, while (2) ablating TL heads should reduce accuracy but leave TR ratio largely intact—causing performance to approximate random guessing over all candidate labels with expected accuracy $\frac{1}{|\mathbb{Y}|}$ [5]. As a control, we also ablate 3% of randomly chosen heads disjoint from the identified TR/TL sets. We report the results averaged over the seven datasets.

**Separability of TR and TL functionality** Figure 7 confirms our conjecture. Removing top TR heads collapses the TR ratio from nearly 100% to ∼20%, leading to a drastic accuracy drop. In contrast, removing top TL heads

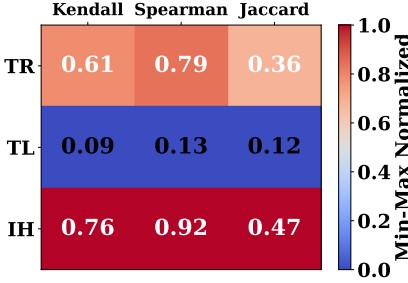

Figure 6: Pairwise overlap/correlation of TR/TL heads and IHs across datasets. TR heads and IHs are task-consistent, whereas TL heads vary widely. See Appendix G.4 for other models.

---

[5] For the seven datasets with 4 having 2 labels, 2 having 3 labels, 1 having 6 labels, the average random guessing level is 40.48%

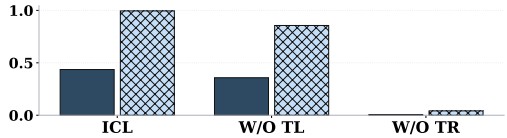
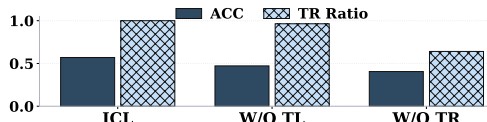

(a) Ablating with shuffled demonstration texts.   (b) Ablating with relabeled demonstrations.

Figure 8: Dataset-average effects of ablating TR and TL heads under input perturbations. **(A)** Shuffling character order of demonstration texts destroys TL, making TL ablation negligible while TR ablation still matters. **(B)** Relabeling demonstrations alters the label space, thereby greatly reducing the impact of TR head ablation.

lowers accuracy by ∼30% but only slightly reduces the TR ratio (by ∼10%). This highlights a key property: *separability*, i.e., TR and TL can be independently controlled and intervened upon, consistent with the conclusions in Pan et al. (2023) achieved through input perturbations. (see Appendix H.1 for other models).

**TR heads, IHs, and implications for zero-shot** Ablating IHs produces a pattern closely resembling TR head ablation: large accuracy losses primarily due to failed task recognition. This supports the conclusion that IHs influence ICL mainly by strengthening TR. Likewise, the root cause of poor zero-shot performance is insufficient task recognition. Thus, restoring ICL-level accuracy in zero-shot settings hinges on activating the TR functionality—a question we revisit in Subsection 4.3.

**Testing independence via input perturbations** To complement head ablations with an orthogonal causal test, we perturb ICL inputs to selectively disrupt TR or TL. Following Wei et al. (2023); Pan et al. (2023), we consider two cases: Case 1: Keep labels unchanged but randomize the character order of demonstration texts (e.g., "I like it: positive" → "tkl iieI : positive"). This destroys TL, since no meaningful mapping from such nonsensical texts to labels remains. Case

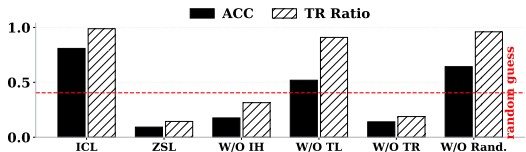

Figure 7: Effects of ablating the top 3% of different heads, averaged across datasets. TR head ablation severely reduces both accuracy and TR ratio, while TL head ablation primarily reduces accuracy.

2: Keep texts unchanged but replace demonstration labels with arbitrary tokens (e.g., "negative" → "0", "positive" → "1"), thereby altering the label space recognized by TR heads.

We hypothesize that: if the TR heads and TL heads maintain sufficient independence, then in Case 1, ablating TL heads should have little effect (TL is already disabled), while in Case 2, ablating TR heads should matter less (the original TR functionality is nullified).

As shown in Figure 8a, when texts are shuffled, TL ablation barely matters while TR ablation remains devastating. Conversely, in Figure 8b, TR ablation has little effect, compared to the significant impact shown in Figure 7, since the recognized label space has shifted, while TL ablation behaves as in the standard case. These results confirm the robustness of TR/TL independence across diverse ICL settings (see Appendix H.2 for other models).

## 4.3   STEERING WITH TR/TL HEADS: FUNCTIONAL AND GEOMETRIC INSIGHTS

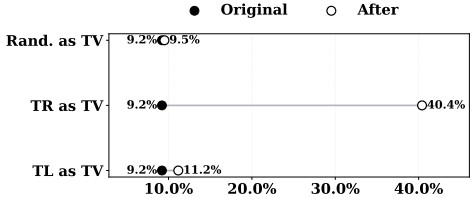

Figure 9: Zero-shot accuracy gains from steering with task vectors constructed from TR, TL, or random heads. TR-based task vectors consistently recover ICL-level accuracy, while TL-based vectors have weaker effects.

With fine-grained ablations establishing that TR and TL heads are necessary for their respective channels, we now test their sufficiency and TSLA-predicted geometric roles by injecting their outputs. Ablations show what happens when heads are removed, but not what happens when they are added. We therefore complement them with steering experiments that examine how TR and TL head outputs affect zero-shot behavior and hidden-state updates. Specifically, we test their suitability as task vectors (TVs) (Todd et al., 2024; Hendel et al., 2023) by extracting their outputs at the final token position from ICL prompts, summing them across prompts, and injecting them into the residual stream of zero-shot inputs to evaluate

recovery toward ICL-level performance. We use the top 3% TR/TL heads, compare against 3% random heads, and follow Appendix J.1 to construct task vectors.

**Task recognition as the key to zero-shot failure** Figure 9 mirrors our ablation findings (Figure 7): poor zero-shot performance stems primarily from weak task recognition. Injecting TR-based TVs restores this functionality and improves performance. TL-based vectors are less effective, reinforcing that TL operates based on TR (see Appendix I.1 for other models).

**Task-type dependence of steering effectiveness** Note that the relative ineffectiveness of TL heads as task vectors can be partly attributed to the classification datasets we use, where performance is tightly linked to task recognition and effectively upper-bounded by the TR ratio. In contrast, generation tasks differ fundamentally in that no fixed label space exists—the label space is indefinite and, in principle, infinite. As a result, model success in such tasks is less constrained by recognizing a closed set of labels, and instead depends more on learning and applying the correct input–output mapping. To examine this scenario, we consider a sentiment-controlled review generation task with prompts such as: *"Write a positive/negative review of a movie within 30 words."* Labels are coherent reviews with the desired sentiment[6]. We identify TR and TL heads on ICL-styled prompts from this task following Appendix J.2, and use their outputs as task vectors to influence the zero-shot generations. An LLM evaluator is then used to rate generations from 0 to 10 based on sentiment adherence and language coherence. As shown in Figure 10, TL-based vectors significantly outperform TR-based and random vectors, consistent with TL heads capturing mappings from demonstrations to the sentiment values and semantic coherence of the labels (see Appendix I.2 for other models). Nevertheless, given the relative simple and structured nature of this task and the fact that the label reviews are generated by GPT (OpenAI et al., 2024), we extend our investigation regarding the TV effectiveness using the SubjQA dataset (Bjerva et al., 2020), which we detail in Appendix I.3.

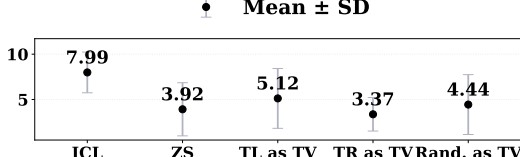

Figure 10: Ratings in the review generation task when steering with TR, TL, or random TVs. TL vectors yield the largest improvements, reflecting their strength in capturing in-context mappings.

**Geometric effects of TR and TL outputs** To understand the significance of TR/TL heads in ICL at a finer level than task vector experiments, we invoke the geometric analysis of hidden states (Kirsanov et al., 2025; Yang et al., 2025), which analyzes the evolution of ICL hidden states and the role of different components. Concretely, given an ICL prompt, we extract the summed outputs of the top 3% TR or TL heads, revert to the hidden state at an earlier layer, and steer it with these outputs. This mimics how head outputs are added to the residual stream during layer progression inside the model. We measure two geometric metrics before and after steering: **(1) Logit Difference**: inner product of the hidden state with the mean unembedding difference between correct and incorrect labels, reflecting label discrimination. **(2) Subspace Alignment**: cosine similarity between the hidden state and the subspace spanned by label unembeddings, reflecting alignment with task-related semantics [7].

The results in Figure 11 demonstrate the specialized geometric effects TR and TL heads have in the evolution of hidden states (for other models, see Appendix I.4). Steering with TR outputs causes hidden states to align significantly more with the task subspace. In contrast, TL outputs adjust the hidden state to align better with the unembedding direction of the correct label in the task space but not with the task subspace overall. This leads us to conjecture that TR outputs are *well-aligned with the task subspace*, thus increase hidden-state alignment with the subspace after addition by decreasing the angle in between. By contrast, TL heads create *pure rotation toward the*

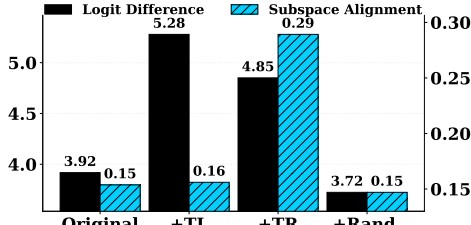

Figure 11: Geometric effects of TR and TL steering. TR outputs enhance alignment with the task subspace, while TL outputs rotate hidden states toward the correct label unembedding within the subspace.

---

[6]Example positive review: "Bold experimental narrative structure defies genre conventions delightfully. Socioeconomic themes challenge viewers' perceptions thoughtfully and respectfully."

[7]See Appendix J.3 for full definitions.

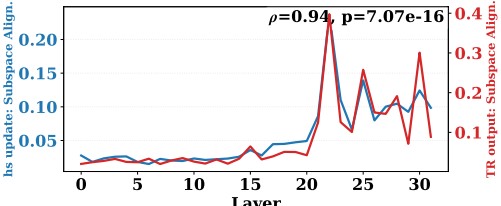 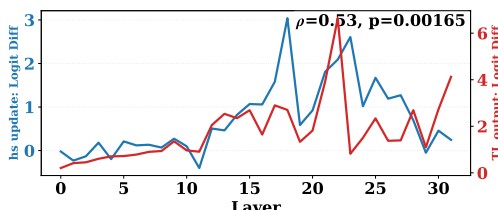

(a) Correlation between hidden state updates and TR head outputs in subspace alignment. Strong correlation confirms TR heads as main drivers of alignment.

(b) Correlation between hidden state updates and TL head outputs in logit difference. TL heads consistently drive discrimination toward correct labels.

Figure 12: Layerwise verification of TR and TL geometric effects. TR heads enforce alignment with the task subspace, while TL heads enforce rotations toward correct label directions.

*correct label unembedding direction*, fine-tuning hidden-state orientation toward the correct label without boosting subspace alignment.

**Layerwise verification of geometric influence**   To validate that TR and TL heads indeed primarily drive these geometric dynamics, we examine hidden state updates under ICL across layers. At each layer, we compute the mean subspace alignment of top-3 TR heads (i.e. heads with top-3 TR scores at the layer) outputs and the mean logit difference of top-3 TL heads outputs, then correlate them with the same metrics computed on the full hidden state updates. Since head outputs contribute directly to layer updates, their correlations with hidden-state geometry across layers indicate how strongly TR and TL heads drive the dynamics. As shown in Figure 12, the correlations are strong, confirming that TR and TL heads dominate layerwise geometric shaping of hidden states. For other models see Appendix I.5. Additional ablation-based verification is provided in Appendix I.6.

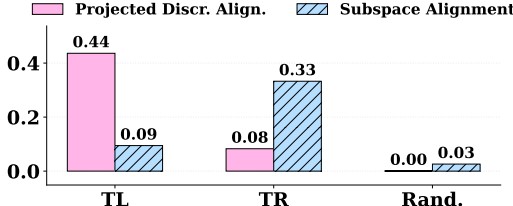

Figure 13: Decomposed geometric effects of TR and TL outputs. TR heads align hidden states to the task subspace; TL heads rotate states within the subspace toward correct label directions.

**TR heads align to task space, TL heads rotate within it**   To support our geometric intuition from Figure 11 that TR heads foster alignment while TL heads perform rotation, we consider two geometric measures of TR and TL head outputs. **(1) Subspace Alignment** — cosine similarity with the task subspace, and **(2) Projected Discriminant Alignment** — cosine similarity with the mean unembedding difference between the correct and incorrect labels after projection onto the task subspace. These measures dissect the geometric effects of head outputs into *steering towards the task space* and *steering within the task space*, enabling more fine-grained verification of the heads' distinct effects (Figure 1 (B), (C)). Figure 13 shows that TL head outputs have high cosine similarity with the mean unembedding difference after projection, confirming that TL heads, when restricted to the task subspace, propel rotation from wrong-label to correct-label unembedding directions. The high cosine similarity between TR heads and the task subspace itself strongly evidences their capability to steer hidden states towards the task subspace and support prediction of task-related labels (see Appendix I.7 for more models).

## 5   CONCLUSION

We presented a unified framework reconciling component and holistic views of in-context learning (ICL) by identifying attention heads specialized for Task Recognition (TR) and Task Learning (TL). Using TSLA, we showed that TR heads align hidden states with the task subspace to recognize labels, while TL heads rotate states within it toward the correct label. Ablation experiments confirmed separable roles: removing TR heads collapses task recognition, whereas removing TL heads mainly reduces accuracy. Steering experiments showed task dependence: TR-based vectors are crucial for classification with fixed labels, while TL-based vectors dominate in open-ended generation. Geometric analyses supported these findings, attributing alignment to TR heads and discriminative rotations to TL heads. Our results also clarify induction heads and task vectors as TR manifestations. Together, this work establishes TR and TL heads as mechanistic foundations of ICL.

ACKNOWLEDGMENTS

This work was supported by JST FOREST Program (Grant Number JPMJFR232K, Japan) and the Nakajima Foundation. We used ABCI 3.0 provided by AIST and AIST Solutions with support from "ABCI 3.0 Development Acceleration Use".

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

# Appendices

## A  STATEMENT OF LLM USAGE

In this work, LLMs are used to help with writing, experiment coding, and visualization of the results. LLMs are also used to produce results in one of the experiments, as explained in Subsection 4.3 and Appendix K.

## B  PROOF OF THEOREM 1

Let $S \in \mathbb{R}^{d \times r}$ be one of the $n$ distinct $r$-dimensional subspaces in $\text{span}(W_U)$ drawn uniformly i.i.d. from the Grassmannian $Gr(r,d)$. Denote $P_S$ as the projection matrix of $S$. Let $c = \dfrac{\|P_S a^l_{N,k}\|_2}{\|a^l_{N,k}\|_2}$ be the projected norm of the normalized head output $\dfrac{a^l_{N,k}}{\|a^l_{N,k}\|_2}$ onto $S$. Since the uniform distribution over $Gr(r,d)$ is induced by the Haar measure over the orthogonal group $O(d)$, the distribution is rotation-invariant; i.e., multiplying $S$ by an orthogonal matrix $U \in \mathbb{R}^{d \times d}$ does not change its distribution. Because orthogonal transformations preserve angles, we also have

$$\frac{\|P_{US} U a^l_{N,k}\|_2}{\|a^l_{N,k}\|_2} = \frac{\|P_S a^l_{N,k}\|_2}{\|a^l_{N,k}\|_2}.$$

Hence, without loss of generality, we pick an $U$ such that $U \dfrac{a^l_{N,k}}{\|a^l_{N,k}\|_2} = e_1$, the unit vector in the first coordinate, with $c' = \|P_{US} e_1\|_2$ having the same distribution as $c$.

Since $US = \text{span}(v_1, ..., v_r)$, where $v_1, ..., v_r$ are the first $r$ columns of a Haar orthogonal matrix $V$, let $V_r = [v_1, ..., v_r]$. Then $P_{US} = V_r V_r^\top$, and we have

$$c'^2 = e_1^\top V_r V_r^\top e_1 = \|V_r^\top e_1\|_2^2 = \sum_{i=1}^r \langle e_1, v_i \rangle^2 = \sum_{i=1}^r V_{1,i}^2,$$

where $V_{1,:}$ denotes the first row of $V_r$. Since $V_r$ is Haar orthogonal, $V_{1,:}$ is uniformly distributed on $\mathbb{S}^{d-1}$ and has the same distribution as $\dfrac{g}{\|g\|_2}$ with $g \sim \mathcal{N}(0, I)$. Therefore, $\sum_{i=1}^r V_{1,i}^2$ has the same distribution as

$$\frac{\sum_{i=1}^r g_i^2}{\|g\|_2^2} = \frac{\sum_{i=1}^r g_i^2}{\sum_{i=1}^d g_i^2} = \frac{\chi_r^2}{\chi_r^2 + \chi_{d-r}^2},$$

since $g_i \sim \mathcal{N}(0,1)$ for all $i$. Because $\chi_r^2 \perp\!\!\!\perp \chi_{d-r}^2$, we have

$$\frac{\chi_r^2}{\chi_r^2 + \chi_{d-r}^2} \sim \text{Beta}\left(\tfrac{r}{2}, \tfrac{d-r}{2}\right).$$

If $c^2 \sim \text{Beta}\left(\tfrac{r}{2}, \tfrac{d-r}{2}\right)$, then the tail probability is

$$\Pr(c \geq x) = 1 - I_{x^2}\left(\tfrac{r}{2}, \tfrac{d-r}{2}\right) = 1 - \frac{B(x^2; \tfrac{r}{2}, \tfrac{d-r}{2})}{B(\tfrac{r}{2}, \tfrac{d-r}{2})},$$

where $B$ is the Beta function. Since the TR score of the head is $\gamma$, the probability that $\|P_S a^l_{N,k}\|_2 \geq \gamma$ is

$$1 - I_{\left(\frac{\gamma}{\|a^l_{N,k}\|_2}\right)^2}\left(\tfrac{r}{2}, \tfrac{d-r}{2}\right).$$

Because there are $n - 1$ subspaces alongside $W_U^{\mathbb{Y}}$, the probability that the head output has the largest projected norm on $W_U^{\mathbb{Y}}$ is

$$1 - (n-1)\left(1 - I_{\left(\frac{\gamma}{\|a^l_{N,k}\|_2}\right)^2}\left(\tfrac{r}{2}, \tfrac{d-r}{2}\right)\right)$$

via the union bound.

## C    ABLATION EXPERIMENTS REGARDING THE IDENTIFICATION OF TR AND TL HEADS

In this section, we demonstrate the advantage of our Task Subspace Logit Attribution (TSLA) method over the naive approach of selecting TR and TL heads based on $\boldsymbol{a}_{N,k}^{l} \boldsymbol{W}_U^{\mathbb{Y}}$ and $\boldsymbol{a}_{N,k}^{l} \boldsymbol{W}_U^{y^*}$, i.e., Direct Logit Attribution (DLA) to the demonstration labels and the correct label. Specifically, Figure 14 shows the consequences of ablating the top 3% TR and TL heads identified via DLA, averaged across datasets on Llama3-8B. While ablating the identified TR heads achieves the intended effect of disabling task recognition by reducing the TR ratio, ablating the identified TL heads fails to induce the expected outcome of driving ICL toward random guessing over the label space. This indicates that the DLA approach cannot isolate distinct mechanistic causes for the TR and TL components of ICL, and reflects its inability to correct identify the real TL heads. Instead, it largely identifies heads that broadly amplify the logits of all demonstration label tokens, which may also increase the correct label logits but still primarily function through task recognition rather than true label differentiation.

To validate this statement, and following the setup of Figure 2, we report the dataset-averaged Jaccard Coefficient, Kendall's $\tau$, and Spearman's $\rho$ values between TR/TL heads identified by DLA and those identified by TSLA, as well as their relationship with IHs. As shown in Figure 15, both TL and TR heads selected via DLA strongly overlap with the TR heads identified by TSLA at the 3% level. This corroborates our conclusion that DLA fails to effectively recover genuine TL heads. Furthermore, the weak correlation between the TR/TL sets obtained from the two methods is reinforced by Figure 16, which displays overlap, correlation, and consistency analyses between DLA TR/TL heads and IHs. The strikingly high consistency between DLA TR and TL heads across all three metrics demonstrates the lack of a meaningful distinction between them. Meanwhile, the low correlation between DLA heads and IHs highlights another limitation of DLA: it cannot provide mechanistic explanations for the well-documented importance of IHs in ICL.

Finally, to justify our second critique of the DLA approach in Subsection 3.1 regarding its sensitivity to the concrete set of demonstration labels as a hyperparameter and its inability to comprehensively capture the task semantics, we consider the following experiment on SST-2. We replace "positive" and "negative", i.e. the default demonstration labels used to create ICL prompts from the dataset, with "unfavourable" and "favourable", which do not alter the essence of the task. Then we test how the ablation of TR heads identified with DLA and our TSLA using the ICL prompts with the original labels will impact the ICL accuracy and TR ratio with the new labels. The results in Figures 25–30 confirm the robustness of our TSLA method against demonstration label shifts in identifying TR heads. On all models except Qwen2.5-32B, ablating the TR heads selected using our TSLA approach causes a significantly larger impact on ICL performance and TR ratio with the new demonstration labels on the SST-2 dataset, with the gap being most prominent for the three Llama family models.

## D    IMPLEMENTATION DETAILS

**Models**    We use the official HuggingFace implementations of all models. Models with more than 10B parameters are quantized to 4-bit for efficiency.

**Datasets**    We use the official HuggingFace implementations of all datasets, except for the Review dataset, which we curated ourselves. The Review dataset was generated using ChatGPT-4o (OpenAI et al., 2024) and contains 200 datapoints. Each datapoint consists of a prompt instructing the model to generate a movie review in the format: `''Write a positive review for a movie. The positive review should be within 30 words.''` The 30-word limit was chosen to set the `max_new_tokens` parameter (set to 45) when calling the generation function. Labels are ChatGPT-4o–generated reviews that comprehensively assess a movie from multiple aspects in the requested positive/negative tone. For example: *"Bold experimental narrative structure defies genre conventions delightfully. Rich orchestral score enhances every pivotal moment. Progressive messages inspire reflection on equality and justice. Raw vulnerability on screen fosters sincere emotional investment."* as a positive review. Details of dataset curation are provided in Appendix K. The dataset is balanced, with 100 positive and 100 negative reviews.

**ICL setting**    For each dataset (except the Review dataset), we select demonstrations from the training set and queries from the test set, or the validation set if ground-truth test labels are unavailable. For

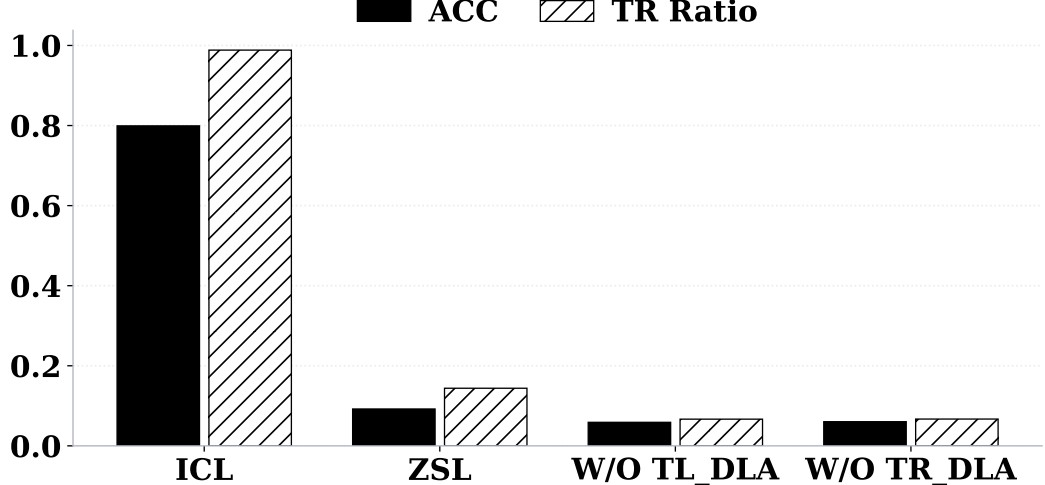

Figure 14: Effects of ablating the top 3% TR and TL heads identified using DLA, averaged across datasets on Llama3-8B. While TR heads reduce task recognition as expected, TL heads do not replicate the behavior predicted for task-learning components.

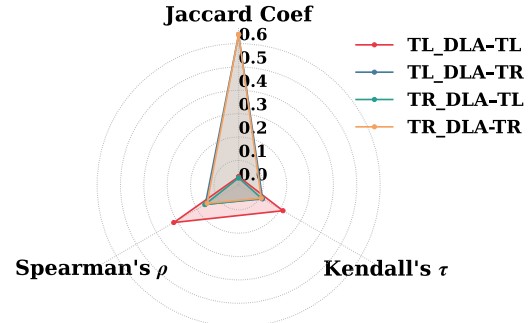

Figure 15: Dataset-averaged Jaccard Coefficient, Kendall's $\tau$, and Spearman's $\rho$ between TR and TL heads identified using DLA and TSLA at the top 3% level. DLA heads overlap substantially with TR heads, confirming their inability to recover distinct TL heads.

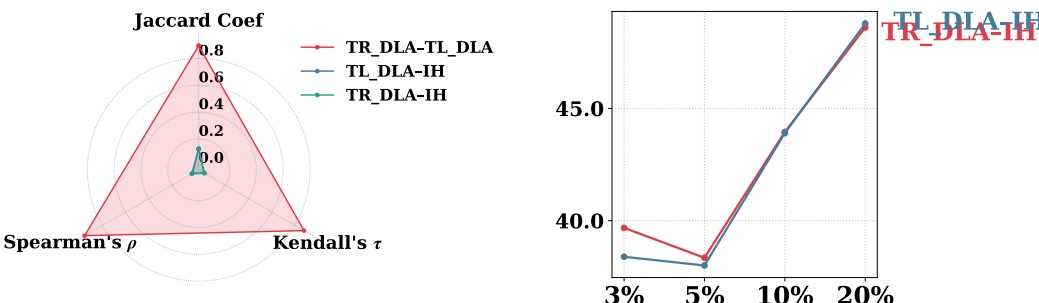

(a) Dataset-averaged Jaccard Coefficient, Kendall's $\tau$, and Spearman's $\rho$ values for TR heads TL heads identified using DLA and IHs at the top 3% level.

(b) Conditional Average Percentage at the top 3%, 5%, 10%, and 20% levels for TR (DLA)–IH and TL (DLA)–IH pairs, averaged across datasets.

Figure 16: Dataset-averaged overlap, correlation, and consistency analyses of TR and TL heads identified using DLA and their relationship with IHs. Results show high redundancy between DLA TR and TL heads and weak association with IHs, underscoring the limitations of DLA in separating TR and TL mechanisms or explaining IH significance.

demonstration selection, we retain at most the first 10,000 training examples. For evaluation, we use the first 1,000 test or validation examples. For the Review dataset, we use the first 50 examples for demonstration selection and the remaining 150 for testing. Prompt templates used to construct ICL prompts are listed in Table 1.

**Devices** All experiments were conducted on an H200 GPU.

**Random label mappings** For the experiment in Appendix G, where demonstration labels are replaced with numbers, we use the mappings in Table 2.

When demonstration labels are replaced with numeric symbols, we also modify the prompt templates in Table 1. Specifically, for SNLI and CB, "True or maybe or false" is replaced with "0 or 1 or 2," and for RTE, "True or false" is replaced with "0 or 1."

**Flipped label mappings** For the experiment in Subsection 4.2, where demonstration labels are flipped, we use the mappings in Table 3.

## E    EXPERIMENT DETAILS CONCERNING THE IDENTIFICATION OF **IH**S

For each dataset, we use the first 50 queries to identify the top IHs. Let the queries be $q_1, \ldots, q_{50}$, where $q_i$ has token length $s(q_i)$. For each $q_i$, the LLM outputs an attention tensor $\boldsymbol{Attn}_i \in \mathbb{R}^{L \times N_h \times s(q_i) \times s(q_i)}$, with $L$ being the number of layers and $N_h$ the number of attention heads per layer. The $n_h$-th head in layer $l$ has an attention matrix $\boldsymbol{Attn}(l, n_h)_i \in \mathbb{R}^{s(q_i) \times s(q_i)}$, where $\boldsymbol{Attn}(l, n_h)_{i,j,k}$ denotes the attention from the $k$-th token to the $j$-th token in $x_i$.

**Identification of IHs** For each $q_i$, we randomly sample 8 demonstrations and prepend them to $q_i$. The resulting ICL prompt, $Q_i$ (length $s(Q_i)$), follows the format $\langle t_{i,1} \rangle : \langle y_{i,1} \rangle, \ldots, \langle t_{i,8} \rangle : \langle y_{i,8} \rangle, \langle q_i \rangle :$ where $\langle t_{i,k} \rangle$ is the sentence part of demonstration $k$ (e.g., "I like this movie. Sentiment"), and $\langle y_{i,k} \rangle$ is the label (e.g., "positive"), separated by a colon. $\langle q_i \rangle$ is the sentence for the query. At the position of the final colon, an IH is expected to attend to tokens after previous colons—that is, the label tokens $\langle y_{i,1} \rangle, \ldots, \langle y_{i,8} \rangle$. Let $\mathbb{I}_i$ be the set of label token indices in $Q_i$. The IH score for head $(l, n_h)$ over the 50 queries is defined as $\sum_{i=1}^{50} \sum_{k \in \mathbb{I}_i} Attn(l, n_h)_{i,k,s(Q_i)}$, i.e., the total attention a head assigns at the final ":" position to the positions of all the label tokens, summed over all 50 queries. We calculate the IH scores for all $(l, n_h)$ pairs and choose the top 3% attention heads as the identified **IH**s.

## F    VERIFYING WHETHER THE UNEMBEDDINGS OF SEMANTICALLY RELATED LABELS EXIST IN COMMON SUBSPACES

The theoretical and practical validity of our TSLA method relies on the assumption that LLM unembeddings of semantically related tokens lie in common low-dimensional subspaces, an observation supported by prior work (Zhao et al., 2025; Saglam et al., 2025). To further verify this assumption, we conduct the following experiment. We consider a set $\mathbb{T}$ of semantically related sentiment tokens, with $\mathbb{T} =$[positive, negative, favourable, unfavourable, pleasing, disappointing, enjoyable, unpleasant, satisfying, dissatisfying, delightful, distasteful, uplifting, depressing, entertaining, regrettable, excellent, terrible]. For each token $t \in \mathbb{T}$, we measure the norm of its unembedding projected onto the subspace spanned by the unembeddings of the remaining tokens in $\mathbb{T}$, i.e.,

$$\|\mathrm{Proj}_{\boldsymbol{W}_U^{\mathbb{T}/\{t\}}} W_U^t\|_2.$$

Then we randomly draw a token set $\mathbb{T}'$ of the same size as $\mathbb{T} \setminus \{t\}$ from the full vocabulary (i.e., $|\mathbb{T}'| = |\mathbb{T} \setminus \{t\}|$), and compute

$$\|\mathrm{Proj}_{\boldsymbol{W}_U^{\mathbb{T}'}} W_U^t\|_2.$$

If unembeddings of semantically related tokens indeed concentrate in common subspaces, we should observe

$$\|\mathrm{Proj}_{\boldsymbol{W}_U^{\mathbb{T}/\{t\}}} W_U^t\|_2 \; > \; \|\mathrm{Proj}_{\boldsymbol{W}_U^{\mathbb{T}'}} W_U^t\|_2,$$

meaning that the unembedding of $t$ aligns more strongly with the subspace spanned by its semantic peers than with a randomly selected subspace. We compute both norms for every $t \in \mathbb{T}$ across all models and report the results in Figures 19–24. The results show that projection norms are indeed

substantially larger when $t$ is projected onto the subspace spanned by its semantic correlates, thereby validating the core assumption behind our TSLA approach. To further support these visual findings, we conduct a Wilcoxon signed-rank test on the paired norm values. The results in Table 5 show that the differences are highly statistically significant.

## G SUPPLEMENTARY MATERIALS FOR SUBSECTION 4.1

### G.1 ABLATION STUDIES FOR THE PERCENTAGE THRESHOLD OF SELECTING TOP TL AND TR HEADS

Throughout the paper, we select the top 3% TL and TR heads based on their respective scores and use them in subsequent analysis, ablation studies, and steering-based experiments. To assess whether this particular threshold influences our conclusions, we sweep the percentage from 1% to 10% in increments of 1%. For Llama3-8B, which has 1024 heads, each 1% corresponds to 10 heads. For ablation experiments, we remove different percentages of top TL or TR heads and evaluate the effect on accuracy and TR ratio. For task-vector steering experiments, we construct task vectors from the outputs of TL or TR heads selected at each percentage level. The results in Figure 52 and Figure 53 show that our findings are robust to the choice of threshold. Specifically, ablating different percentages of TL heads impacts accuracy but leaves the TR ratio largely unchanged, whereas ablating TR heads affects both accuracy and the TR ratio. Moreover, using more top TR heads when constructing task vectors increases accuracy (with saturation around 7%), whereas including more TL heads does not yield accuracy gains.

These observations not only reinforce our main conclusions but also shed light on how attention heads collectively realize TL and TR functionality: heads at different top-percentage levels make additive contributions—albeit with varying strengths—rather than the behavior being dominated by only a few exceptional heads. This further justifies selecting heads based on a percentage threshold rather than attempting to isolate only a handful of specific heads.

### G.2 REPLICATION OF FIGURE 2 FOR OTHER MODELS

Figures 31–35 replicate the experiments from Figure 2, demonstrating the robustness of our findings across different models. In every case, TR heads show markedly stronger overlap, correlation, and consistency with IHs than the other head pairs. The consistently higher values of the TR–IH pair in terms of Jaccard Coefficient, Kendall's $\tau$, Spearman's $\rho$, and Conditional Average Percentage across all levels and architectures confirm our conclusion.

### G.3 REPLICATION OF FIGURE 3 FOR OTHER MODELS

Figures 36–40 replicate the experiments from Figure 3 on additional models. These visualizations support our claims in Subsection 4.1 that: **1)** TR heads generally reside in deeper layers than TL heads and IHs; **2)** The overlap between TR heads and IHs is larger and predominantly occurs in deeper layers.

To complement these figures, Tables 6–11 report the mean layer indices of the top 3%, 5%, and 10% TR heads, TL heads, and IHs averaged across datasets. We also conduct Mann–Whitney U tests to assess whether the differences in layer distributions between TR heads and IHs, and between IHs and TL heads, are statistically significant. The results show that the distributional differences between IHs and TL heads are often not significant ($p \geq 0.05$). Even when significant, the $p$-values are much larger than those observed between TR heads and IHs, indicating that the TR–IH distinction is far more robust.

### G.4 REPLICATION OF FIGURE 6 FOR OTHER MODELS

Figures 41–45 extend the analysis of Figure 6 to the remaining models. The results reinforce our conclusion that TR heads and IHs identified across datasets or tasks are largely consistent, whereas TL heads vary substantially.

To further test this, we evaluate the cross-dataset transferability of TR heads. Specifically, we ablate top 3% TR heads identified using SST-2 prompts and measure their impact on accuracy and TR ratio for the six remaining datasets. The results for all models in Figures 73-78 in general confirm the transferrability of TR heads across datasets, but some interesting variations among datasets and models are also worth noting. First, on the three Llama family models, ablating the TR heads identified on the SST-2 dataset can effectively drive both the accuracy and TR ratio on RTE, CB, and MR datasets to near zero, and to a lesser extent impact the two metrics on TREC and SNLI. On Yi-34B, the ablation instead significantly impact the model performance on SNLI rather than MR. For the remaining two Qwen family models the consequence of the ablation over datasets is similar to the case of Llama models but to a lesser degree overall.

## G.5    REPLICATION OF FIGURE 4 FOR OTHER MODELS

In Figures 54–58 we report the results of quantifying the attention weight distributions of top TL and TR heads on other models, which are largely similar to Figure 4. TR heads assign larger weights to demonstration labels to collect information about the task label space, whilst TL heads attend to the query to leverage its specific semantics to facilitate its matching to the proper label token.

## G.6    REPLICATION OF FIGURE 5 FOR OTHER MODELS

In Figure 5, we visualize the distinct attention patterns of TL and TR heads by showing the attention distributions of the top-1 TL and TR heads over an ICL prompt formed using the first test query of SST-2. In Figures 59–67, we report the corresponding attention distributions over ICL prompts formed using the second through tenth test queries of SST-2; these largely resemble Figure 5 and thus support its validity.

## G.7    DEMYSTIFYING THE LAYER ORDER OF TL AND TR FROM THE PERSPECTIVE OF THE LAYERWISE EVOLUTION OF ICL HIDDEN STATES

Our results in Figure 3 suggest that attention heads responsible for TL emerge in earlier layers than those responsible for TR. This raises a question: how can the model perform task learning before it has recognized the task? The key to resolving this lies in our definition of TL and TR (and their corresponding heads) based on how they are carried out in the actual forward computation process of the model, which ultimately results in the promotion of certain tokens' logits at the output layer and their prediction. From a logit-centric viewpoint, TL corresponds to the process by which the model increases the logit margin between the correct label and the incorrect candidate labels *within* the task's label set. In contrast, TR corresponds to increasing the logit margin between the task's label set as a whole and all other non–task-related tokens, enabling the model to detect which task it should perform. Under this interpretation, TL and TR are fundamentally distinct and need not occur simultaneously, which explains the low overlap and weak correlation between TL and TR heads observed in Figure 2a.

To further validate our claim that TL heads precede their TR counterparts, we analyze the layerwise logit dynamics of correct labels, incorrect candidate labels, and non–task-related tokens using the ICL hidden-state evolution. Specifically, we use the Logit Difference metric introduced in Subsection 4.3 to measure (i) the logit difference between the correct and incorrect candidate labels by projecting intermediate hidden states onto the corresponding unembedding differences, and (ii) the logit difference between the maximum logit among task-label tokens and the maximum logit among irrelevant tokens, which reflects task recognition capability. The results, averaged across datasets and models and shown in Figures 46–51, reveal a striking contrast between these two margins. The margin between correct and incorrect task labels starts around 0 and becomes positive after only a few layers—indicating that the model quickly performs task learning and distinguishes between the candidate labels. In contrast, the margin between task labels and non-task labels is strongly negative in early layers and only becomes positive much later, indicating that the model initially fails to identify the task but begins to perform task recognition in deeper layers. This pattern aligns precisely with our observed ordering of TL and TR heads.

## H    SUPPLEMENTARY MATERIALS FOR SUBSECTION 4.2

### H.1    REPLICATION OF FIGURE 7 FOR OTHER MODELS

In Figures 68–72, we replicate the ablation experiments on additional models and examine their effects on dataset-average accuracy and TR ratio. The results echo our observations in Subsection 4.2: **1)** Ablation of TR heads and TL heads impacts the TR and TL components of ICL separately. **2)** Ablating IHs produces effects similar to ablating TR heads. **3)** The primary cause of low accuracy in the zero-shot case—as well as in cases where TR heads or IHs are ablated—is the failure to adequately activate the TR functionality.

### H.2    REPLICATION OF FIGURE 8 FOR OTHER MODELS

In Figures 79–83, we repeat the experiments from Figure 8 on other models, focusing on the ablation of TR and TL heads when ICL inputs are subjected to perturbations. The results closely mirror those in Figure 8: when the in-context text–label mapping is destroyed or reversed, ablating TL heads has no effect—or even a positive effect—on accuracy. By contrast, since these perturbations do not alter the demonstration label space, the TR component of ICL remains unaffected.

### H.3    ASSESSING THE INDEPENDENCE OF TR AND TL WITH FLIPPED DEMONSTRATION LABELS

To further validate the independence of TR and TL and the mechanisms of their associated heads, we analyze the effect of ablating TR/TL heads under flipped demonstration labels. Specifically, we apply a mapping $g : \mathbb{Y}' \to \mathbb{Y}'$ that reverses the demonstration labels (e.g., "negative" → "positive," "positive" → "negative"), as listed in Table 3. Since label flipping invalidates the original text–label mapping captured by TL heads, we conjecture that ablating top TL heads will *increase* accuracy, while the effect of ablating TR heads will remain unchanged because the label space itself is preserved.

The dataset-average results in Figures 84–89 confirm this conjecture: ablating top TL heads indeed raises accuracy, whereas ablating top TR heads still drives accuracy close to zero, as observed in the standard setting of Figure 7.

### H.4    COMPOSITE TASK LEARNING AND RECOGNITION IN ICL

In this section, we discuss how the findings in Li et al. (2025) relate to our framework of Task Learning (TL) and Task Recognition (TR), together with the corresponding experimental results. The main findings of Li et al. (2025) are threefold:

1. For an ICL prompt, only the hidden states or head outputs at certain specific token positions serve as effective TVs. For instance, in the prompt "I like this movie: positive, I don't like it: negative, I love it:", only the hidden states or head outputs at the two ":" positions can serve as TVs because they encode the information needed to predict "positive" and "negative" as the next tokens. This suggests that a TV encodes only the information needed for next-token prediction.
2. Consequently, in settings involving multiple or composite tasks, a TV only supports predicting the label of the first task, but not subsequent ones. For example, in the prompt "France, big → Paris, small", which combines a Country–Capital task and an Antonym task, a TV can correctly produce "Paris" for the first task but not "small" for the second.
3. Nevertheless, hidden states from prompts corresponding to different composite tasks that share the same initial task remain well-separated and linearly classifiable.

The limitation of task vectors to single-token prediction is directly relevant to our work, since we identify TL and TR heads based on their outputs at a single token position, and similarly construct TVs from those outputs to inject at a single position. Moreover, the effectiveness of TVs only for the first task, together with the linear separability of hidden states from different composite tasks, suggests that under composite tasks the model must repeatedly perform task learning for each constituent task while simultaneously maintaining a recognition mechanism that tracks the full scope of the composite structure. This unusual composite setting therefore provides an opportunity to extend our TL & TR framework to more complex scenarios.   To investigate TL and TR in composite task learning, we

conduct the following experiment. We consider the Country–Capital + Antonym composite task with ICL prompts of the form "France, big → Paris, small. China, high → Beijing, low ... Germany, quick →". Using 8-shot ICL prompts, we obtain two sets of top 3% TR and TL heads, one for each constituent task. For Task 1 (Country–Capital), we identify the heads whose outputs at the final "→" position promote the logits of different capital names, as well as those whose outputs increase the logit margin between the correct capital label "Berlin" and the remaining capital names (the Task 1 TR and TL heads). Then, we supplement the query with the first task's label, yielding the prompt "France, big → Paris, small. China, high → Beijing, low ... Germany, quick → **Berlin,**", and repeat the procedure to identify the TR and TL heads for Task 2 (Antonym), whose label space consists of all adjectives used in the Antonym task. After identifying the relevant heads, we ablate the top 10% **Task 2** TR and TL heads to measure their impact on the accuracy and TR ratio of **Task 1** (for prompts such as "Germany, quick →"). Likewise, we ablate the top 3% **Task 1** TR and TL heads to evaluate the performance of **Task 2** (for prompts such as "Germany, quick → Berlin,"). The results in Figure 17 and Figure 18 reveal a striking pattern: ablating TL heads across tasks has minimal effect on accuracy and TR ratio, whereas ablating TR heads across tasks significantly affects both, even though the TR heads are identified from entirely different task label spaces. We also report the Jaccard coefficient between the two TR (and TL) sets, as well as the Spearman's $\rho$ between the Task 1 and Task 2 TL (and TR) scores across all heads, in Table 4. These statistics show that TR heads across tasks have substantially higher overlap and much stronger correlation than TL heads, further confirming the cross-task similarity of TR heads and the cross-task dissimilarity of TL heads. Based on these findings, we provide the following mechanistic explanation for ICL under composite tasks. To correctly infer each task's label, a distinct set of TL heads specialized for that specific task (see (Yin & Steinhardt, 2025) for discussion of such specialization in pretraining) activates to promote the correct task-specific label using the task labels learned from the demonstrations. In contrast, at the broader level of solving the entire composite task, a unified set of TR heads tracks the label spaces of all constituent tasks. These heads provide the task-recognition foundation that enables each task-specific TL set to selectively promote its respective labels as the prompt progresses through the subtasks. This interpretation closely aligns with our findings in Figure 6, where TR heads—but not TL heads—identified across datasets exhibit a high degree of overlap and correlation.

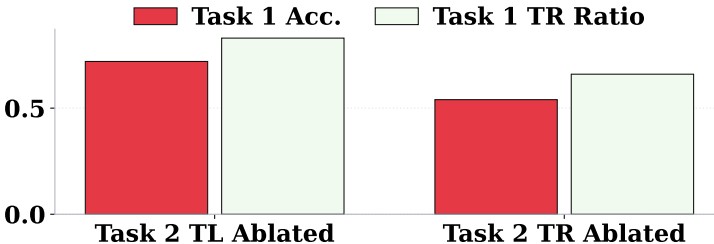

Figure 17: Effects of ablating the Task 2 TR and TL heads on the accuracy and TR ratio of Task 1.

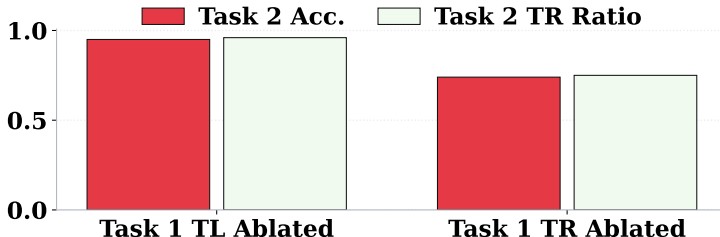

Figure 18: Effects of ablating the Task 1 TR and TL heads on the accuracy and TR ratio of Task 2.

# I   SUPPLEMENTARY MATERIALS FOR SUBSECTION 4.3

## I.1   REPLICATION OF FIGURE 9 FOR OTHER MODELS

Figures 90–94 replicate the steering experiments from Figure 9, evaluating the effectiveness of task vectors constructed from special attention head outputs in other models. For all models except the two Qwen-family models, the results are consistent with Subsection 4.2: task vectors built from TR heads are substantially more effective than those from TL heads. In the Qwen models, however, TL heads match TR heads as task vectors. This deviation can be explained by the high zero-shot accuracy of the Qwen models (Figure 93, Figure 92), which exceeds 20%—considerably higher than the other models. Because these models already achieve strong task recognition in the zero-shot setting, injecting TR-head-based task vectors (which primarily encode recognition) provides less additional benefit.

## I.2   REPLICATION OF FIGURE 10 FOR OTHER MODELS

Figures 95–99 evaluate how task vectors built from different types of heads affect the quality of generated reviews across models. Consistently, TL heads outperform TR heads and random heads as task vectors. An exception is Yi-34B, where steering reduces the average rating below the original zero-shot level. For Qwen2-7B and Qwen2.5-32B, TL-head task vectors even push ratings above the ICL-level baseline. Interestingly, the zero-shot reviews of these models score higher than their ICL reviews. Closer inspection reveals why: ICL reviews, though coherent and stylistically faithful, sometimes contradict the sentiment required in the query. TL heads appear to filter out such inconsistencies by correctly capturing the text–label mapping and discarding misleading signals, thereby boosting zero-shot review quality beyond ICL.

In addition to cross-model replication, Table 12 presents sampled outputs under ICL, zero-shot, and steering with different task vectors. These examples highlight the TL heads' ability to extract the correct text–label mapping and use it for generation. In contrast, zero-shot or TR-head steering often yields generic, off-topic sentences loosely related to the concept of "review."

## I.3   EVALUATING TASK VECTOR PERFORMANCE ON A MORE COMPLEX REVIEW-GENERATION TASK

Because the movie review generation task in the main text has a relatively simple structure and a synthetic two-way label space, we further evaluate task vectors constructed from TL and TR head outputs on a more complex setting: the SubjQA dataset (Bjerva et al., 2020). We use the "book" split, where each datapoint contains a sentiment label and a human-written book review. Unlike the movie task—whose labels are limited to "positive" and "negative"—this dataset features a much richer and more diverse sentiment label space, including labels such as "captivating," "anticlimactic," and "wrenching," among many others. The human-written reviews also introduce greater linguistic complexity and semantic variability. Following the same TSLA-based procedure used to identify TL and TR heads in the main text, we compute TL and TR scores under this enlarged sentiment label space. We then construct task vectors from the outputs of the top TL or TR heads and use them to steer book-review generation in a zero-shot setting. GPT is subsequently asked to rate each generated review on a 10-point scale based on coherence and how well it reflects the intended sentiment label.   The average ratings across models, shown in Figures 100–105, mirror the patterns observed in the movie-review task: TL-based task vectors consistently outperform TR-based vectors and random baselines. This demonstrates that TL heads capture abstract associations between demonstration/query texts and sentiment labels strongly enough to yield effective task vectors even in substantially more complex, real-world generation scenarios, thereby validating the robustness of our TSLA-based identification of TL heads.

## I.4   REPLICATION OF FIGURE 11 FOR OTHER MODELS

Figures 106–110 extend the geometric analysis of Figure 11 to other models. The results largely confirm our earlier observation: TL heads tend to align hidden states with label-unembedding difference directions, while TR heads align hidden states with the broader task subspace.

### I.5 REPLICATION OF FIGURE 12 FOR OTHER MODELS

Figures 116–120 report layer-wise correlations between mean TR/TL head outputs and full layer updates, measured by logit difference and subspace alignment. Across models, we observe clear and consistent correlation patterns, reinforcing that TR and TL heads are the primary drivers of the geometric shaping of hidden states in layer updates.

### I.6 ABLATING TOP TR AND TL HEADS PER LAYER TO VERIFY THEIR GEOMETRIC SIGNIFICANCE

In Figure 12, we validated the geometric importance of TR/TL heads by correlating their outputs with full layer updates. Here, we provide an alternative perspective. Specifically, we ablate the top three TR/TL heads per layer and then remeasure layer-wise hidden state updates under the same two metrics. Figures 121–126 show that TR and TL heads are indeed crucial: without top TR heads, hidden states fail to gradually align with the task subspace, crippling task recognition; without top TL heads, logit differences collapse, preventing hidden states from rotating toward the correct label's unembedding direction. By contrast, ablating three random heads per layer has negligible impact.

### I.7 REPLICATION OF FIGURE 13 FOR OTHER MODELS

Figures 111–115 replicate the analysis of Figure 13 across models. The results are consistent: TL heads excel in projected discriminant alignment, rotating hidden states toward the correct label unembedding and away from incorrect ones. TR heads, conversely, excel in subspace alignment, keeping hidden states well-positioned within the task subspace. Both substantially outperform randomly chosen heads on their respective strengths.

## J EXPERIMENT DETAILS RELATED TO TASK VECTORS

### J.1 CONSTRUCTION AND APPLICATION OF TASK VECTORS

For each dataset, we first construct 8-shot ICL prompts using the last 50 queries. The demonstrations are identical to those used when evaluating the 8-shot ICL accuracy for each dataset. Following the procedure of Todd et al. (2024), we compute the average output (across the 50 prompts) of each identified top 3% TR, TL, or random head at the final token position. We then sum these average outputs across heads to form the task vector.

In the steering experiment, the task vector is added to the hidden state of the final token of each zero-shot query at the midpoint layer (e.g., layer 16 for the 32-layer Llama3-8B). The modified hidden states are then propagated through the subsequent layers, and accuracy as well as TR ratio are measured at the final layer.

### J.2 IDENTIFYING TR AND TL HEADS ON THE MOVIE REVIEW DATASET

A key difficulty in identifying TR and TL heads for free-form generation tasks is the unbounded label space, since labels are not restricted to a finite set of tokens. To address this, we define the relevant label tokens as "positive" and "negative," reflecting the sentiment nature of the review-generation task. Specifically, the TR score of a head is defined as the projection norm of its output onto the span of the unembedding vectors of "positive" and "negative" when processing ICL prompts from the review dataset. The TL score is defined as the inner product between the head output and the difference between the unembeddings of "positive" and "negative," normalized by its TR score. After identifying TR and TL heads, we construct task vectors from their outputs following the procedure in Appendix J.1.

### J.3 MATHEMATICAL DETAILS OF THE MEASURES IN SUBSECTION 4.3 AND CALCULATION PROCEDURE

1. **Logit Difference** Given a hidden state $\boldsymbol{h}$, we compute $\text{Ave}_{y' \in \mathbb{Y}/\{y^*\}}(\boldsymbol{h}^\top(\boldsymbol{W}_U^{y^*} - \boldsymbol{W}_U^{y'}))$, where $\boldsymbol{W}_U$ is the unembedding matrix, $y^*$ is the correct label, and $\mathbb{Y}$ is the demonstration label space.

2. **Subspace Alignment** We compute $\frac{\boldsymbol{h}^\top \mathrm{Proj}_{\boldsymbol{W}_U^{\mathbb{Y}}}^\top \boldsymbol{h}}{\|\mathrm{Proj}_{\boldsymbol{W}_U^{\mathbb{Y}}}^\top \boldsymbol{h}\|_2 \|\boldsymbol{h}\|_2}$, which is the cosine similarity between

$\mathrm{Proj}_{\boldsymbol{W}_U^{\mathbb{Y}}}^\top \boldsymbol{h}$ and $\boldsymbol{h}$.

For evaluation, we take the hidden state of the final position at the layer corresponding to 75% of the model depth (e.g., layer 24 in Llama3-8B). Reported metric values are averaged across the first 30 ICL prompts of each dataset.

## K CURATION DETAILS OF THE REVIEW DATASET

We use the following template, adapted from Zhao et al. (2025), to prompt ChatGPT-4o to generate movie reviews.

| | |
|---|---|
| **Prompt** | Compose a concise 30-word movie review that addresses the following four aspects: plot, sound and music, cultural impact, and emotional resonance. Use a positive tone throughout the review. For the plot, comment on its structure or originality. For sound and music, describe how they enhance the storytelling. For cultural impact, mention any relevant social commentary. Finally, highlight how the film resonates emotionally. Ensure the positive tone is consistent throughout and include positive descriptions of the movie. |
| **Samples** | *Inventive non-linear storyline weaves intrigue with clever twists. Soaring vocal melodies heighten the film's emotional arcs. Relevant socioeconomic themes challenge viewers' perceptions thoughtfully and respectfully. Joyful humor interwoven with drama creates comforting resonance.* |

We use the following template to ask ChatGPT-4o to rate the movie reviews.

| | |
|---|---|
| **Prompt** | Rate the following movie review on a scale of 10. Your rating should be based on two criteria: (1) whether the text is indeed a movie review, and (2) whether it conveys the positive or negative sentiment indicated by the label. Review: Inventive non-linear storyline weaves intrigue with clever twists. Soaring vocal melodies heighten the film's emotional arcs. Relevant socioeconomic themes challenge viewers' perceptions thoughtfully and respectfully. Joyful humor interwoven with drama creates comforting resonance. Sentiment: Positive |
| **Response** | *10* |

Table 1: Prompt templates and labels for different datasets.

| Dataset | Template | Labels |
|---------|----------|--------|
| SST-2 | {Sentence} Sentiment: {Label} | positive / negative |
| SUBJ | {Sentence} Type: {Label} | subjective / objective |
| TREC | Question: {Sentence} Type: {Label} | abbreviation / entity / description / human / location / number |
| MR | {Sentence} Sentiment: {Label} | positive / negative |
| SNLI | The question is: {Premise}? True or maybe or false? The answer is: {Hypothesis} {Label} | true / maybe / false |
| RTE | The question is: {Premise}? True or false? The answer is: {Hypothesis} {Label} | true / false |
| CB | The question is: {Premise}? True or maybe or false? The answer is: {Hypothesis} {Label} | true / maybe / false |

Table 2: Mappings used to replace ground-truth labels with numeric symbols.

| Dataset | Label Mapping |
|---------|---------------|
| SST-2 | negative/positive $\rightarrow$ 0/1 |
| SUBJ | objective/subjective $\rightarrow$ 0/1 |
| MR | negative/positive $\rightarrow$ 0/1 |
| TREC | abbreviation/entity/description/person/number/location $\rightarrow$ 0/1/2/3/4/5 |
| SNLI | true/maybe/false $\rightarrow$ 0/1/2 |
| RTE | true/false $\rightarrow$ 0/1 |
| CB | true/maybe/false $\rightarrow$ 0/1/2 |

Table 3: Mappings used to flip the demonstration labels for each dataset.

| Dataset | Label Mapping |
|---------|---------------|
| SST-2 | negative/positive $\rightarrow$ positive/negative |
| SUBJ | objective/subjective $\rightarrow$ subjective/objective |
| TREC | abbreviation/entity/description/person/number/location $\rightarrow$ entity/description/person/number/location/abbreviation |
| MR | negative/positive $\rightarrow$ positive/negative |
| SNLI | true/maybe/false $\rightarrow$ maybe/false/true |
| RTE | true/false $\rightarrow$ false/true |
| CB | true/maybe/false $\rightarrow$ maybe/false/true |

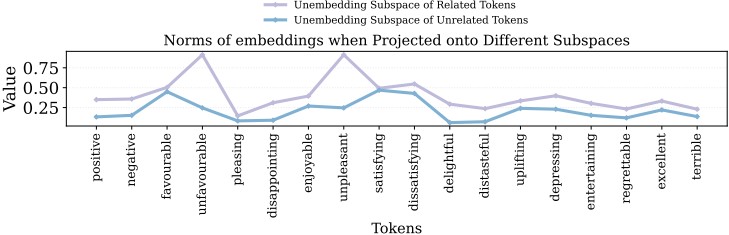

Figure 19: Norms of Llama3-8B token unembeddings when projected onto the unembedding subspace spanned by semantically related tokens vs semantically unrelated tokens.

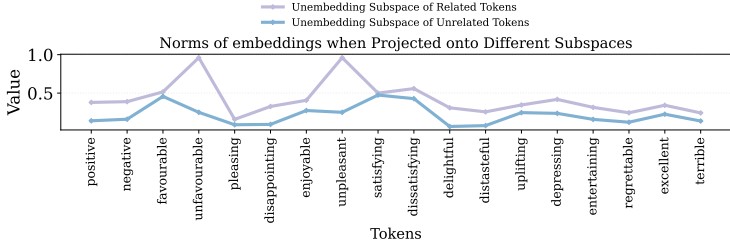

Figure 20: Norms of Llama3.1-8B token unembeddings when projected onto the unembedding subspace spanned by semantically related tokens vs semantically unrelated tokens.

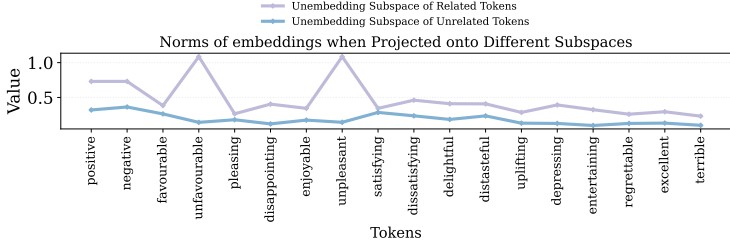

Figure 21: Norms of Llama3.2-3B token unembeddings when projected onto the unembedding subspace spanned by semantically related tokens vs semantically unrelated tokens.

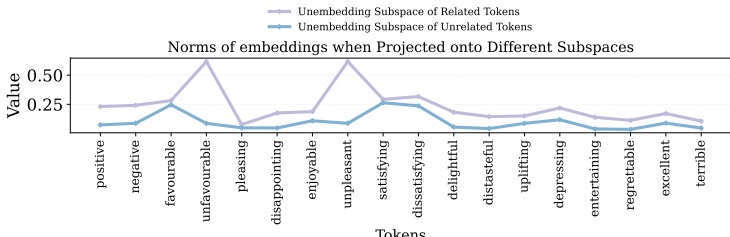

Figure 22: Norms of Qwen2-7B token unembeddings when projected onto the unembedding subspace spanned by semantically related tokens vs semantically unrelated tokens.

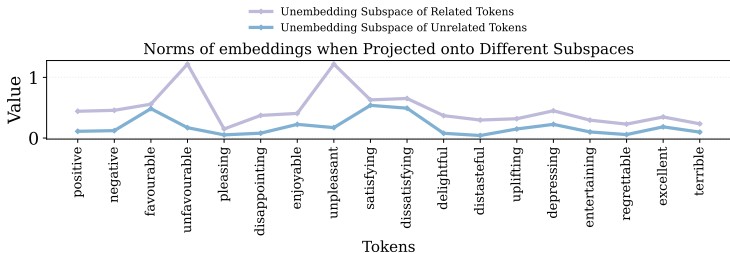

Figure 23: Norms of Qwen2.5-32B token unembeddings when projected onto the unembedding subspace spanned by semantically related tokens vs semantically unrelated tokens.

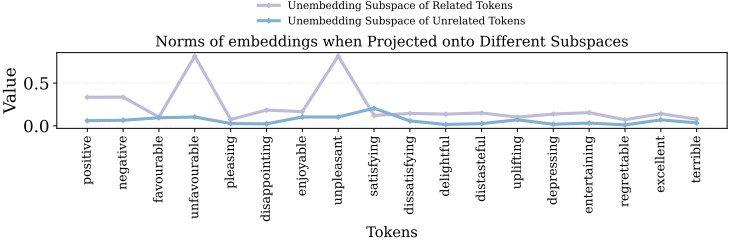

Figure 24: Norms of Yi-34B token unembeddings when projected onto the unembedding subspace spanned by semantically related tokens vs semantically unrelated tokens.

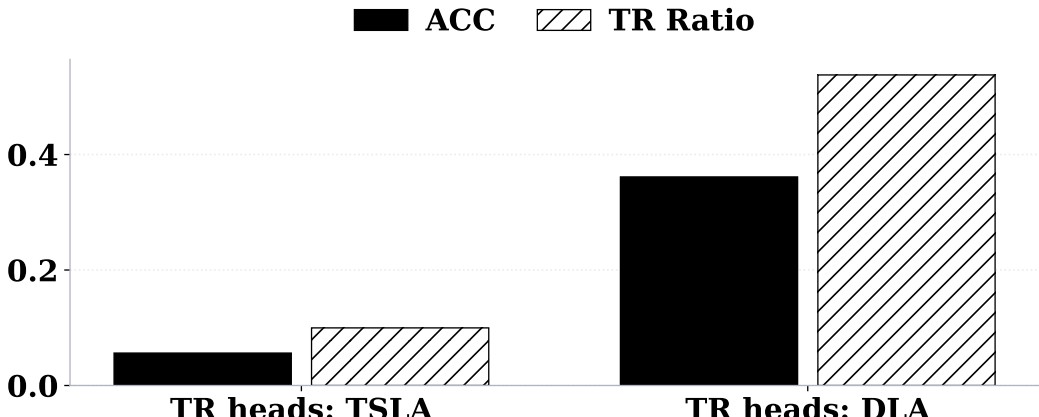

Figure 25: Results on LLama3-8B: Effects of ablating top 10% TR heads identified using TSLA or DLA when the SST-2 demonstration labels are shifted from positive/negative to favourable/un-favourable.

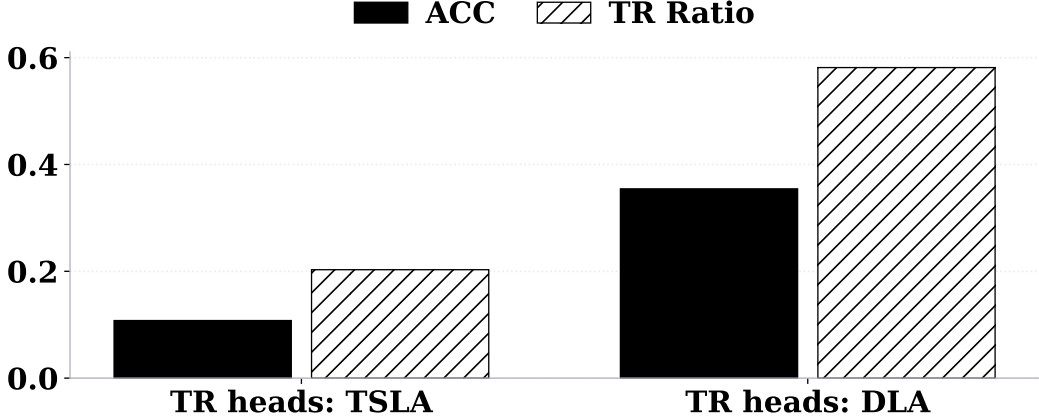

Figure 26: Results on Llama3.1-8B: Effects of ablating top 10% TR heads identified using TSLA or DLA when the SST-2 demonstration labels are shifted from positive/negative to favourable/un-favourable.

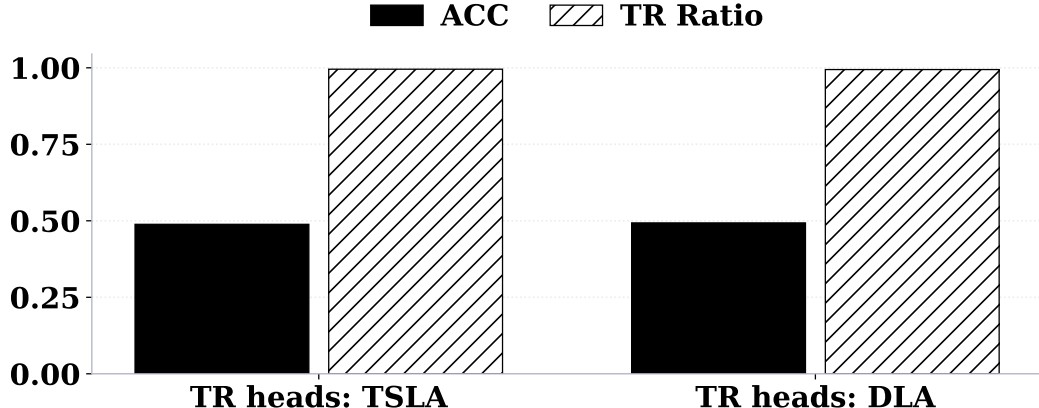

Figure 27: Results on Llama3.2-3B: Effects of ablating top 10% TR heads identified using TSLA or DLA when the SST-2 demonstration labels are shifted from positive/negative to favourable/un-favourable.

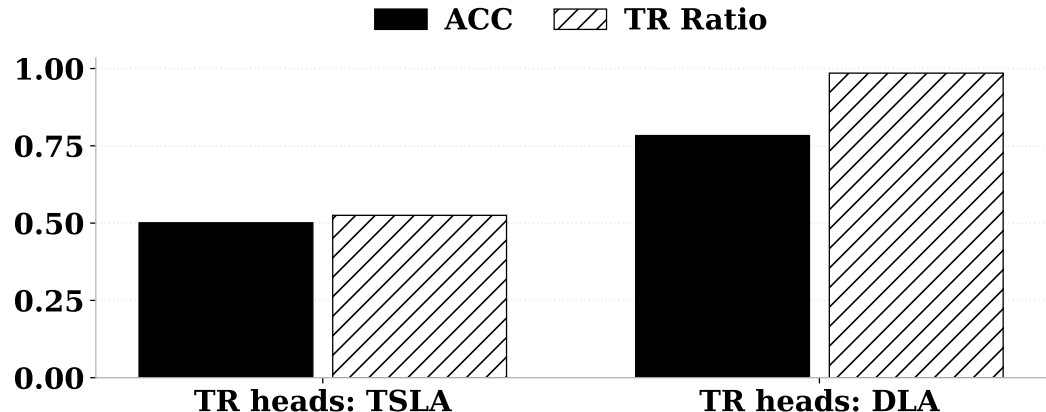

Figure 28: Results on Qwen2-7B: Effects of ablating top 10% TR heads identified using TSLA or DLA when the SST-2 demonstration labels are shifted from positive/negative to favourable/unfavourable.

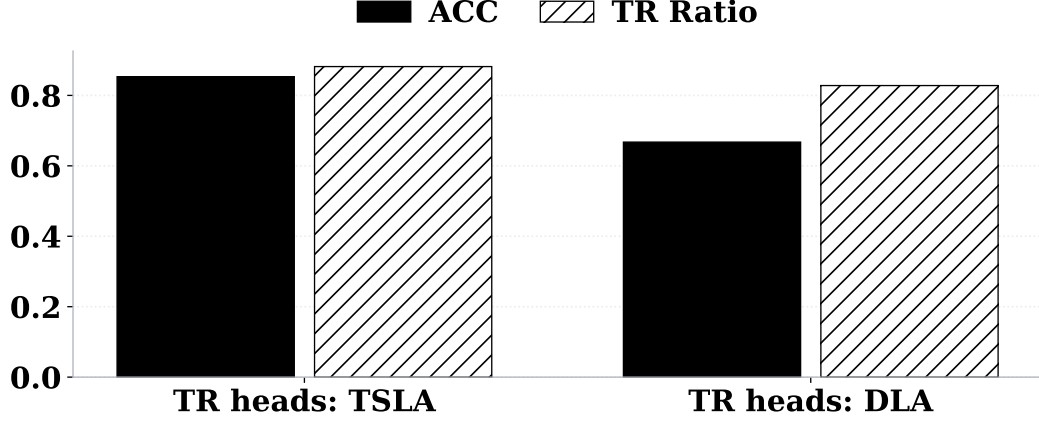

Figure 29: Results on Qwen2.5-32B: Effects of ablating top 10% TR heads identified using TSLA or DLA when the SST-2 demonstration labels are shifted from positive/negative to favourable/un-favourable.

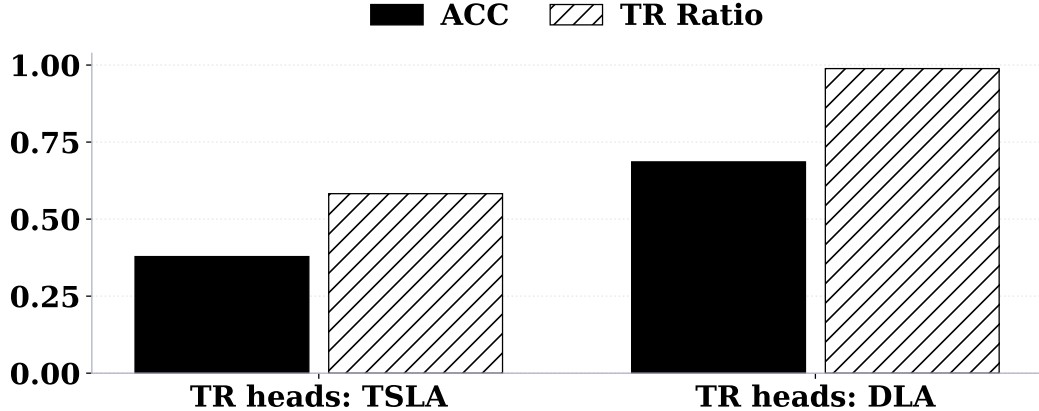

Figure 30: Results on Yi-34B: Effects of ablating top 10% TR heads identified using TSLA or DLA when the SST-2 demonstration labels are shifted from positive/negative to favourable/unfavourable.

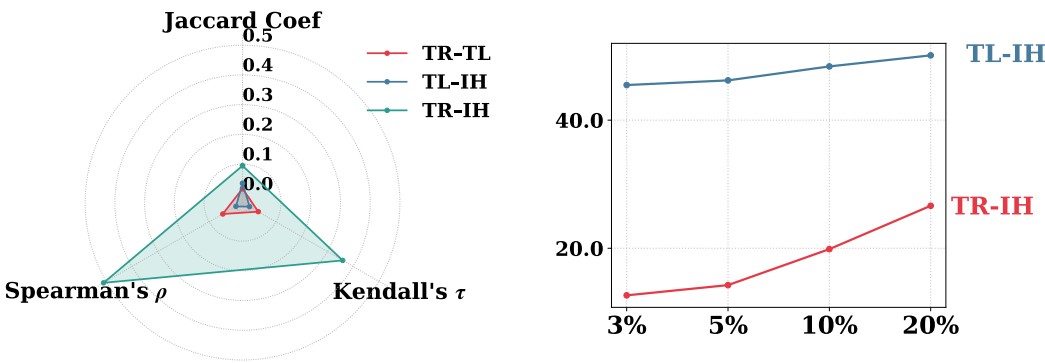

(a) Jaccard Coefficient, Kendall's $\tau$, and Spearman's $\rho$ values for TR heads, TL heads, and IHs.

(b) Conditional Average Percentage at four top levels for TR-IH and TL-IH pairs.

Figure 31: Results of overlap, correlation, and consistency analysis of attention head types averaged across datasets on Llama3.1-8B.

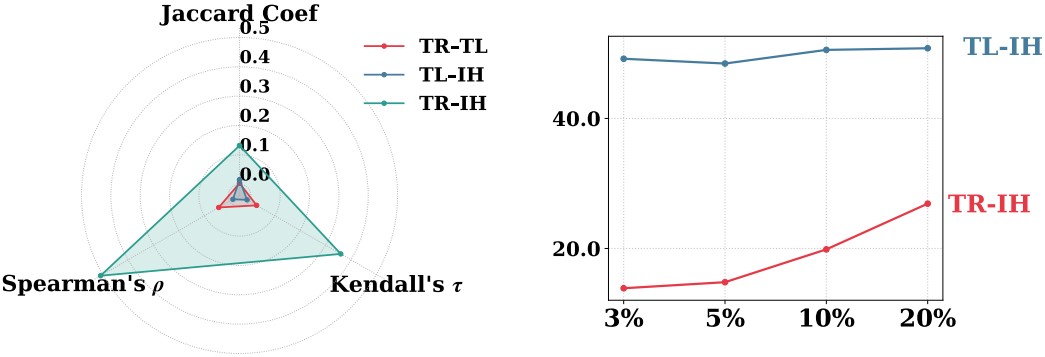

(a) Jaccard Coefficient, Kendall's $\tau$, and Spearman's $\rho$ values for TR heads, TL heads, and IHs.

(b) Conditional Average Percentage at four top levels for TR-IH and TL-IH pairs.

Figure 32: Results of overlap, correlation, and consistency analysis of attention head types averaged across datasets on Llama3.2-3B.

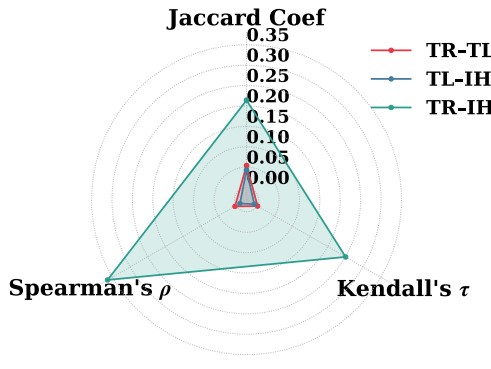

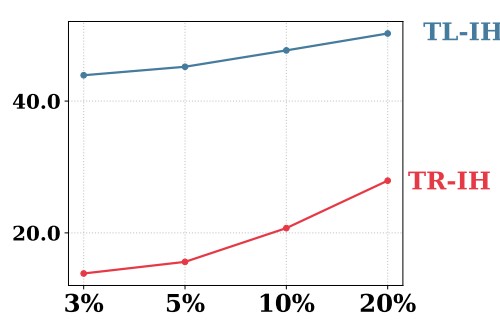

(a) Jaccard Coefficient, Kendall's $\tau$, and Spearman's $\rho$ values for TR heads, TL heads, and IHs.

(b) Conditional Average Percentage at four top levels for TR-IH and TL-IH pairs.

Figure 33: Results of overlap, correlation, and consistency analysis of attention head types averaged across datasets on Qwen2-7B.

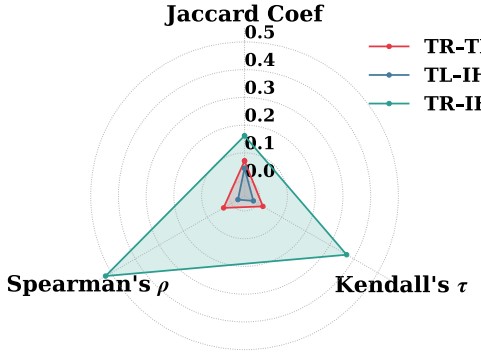

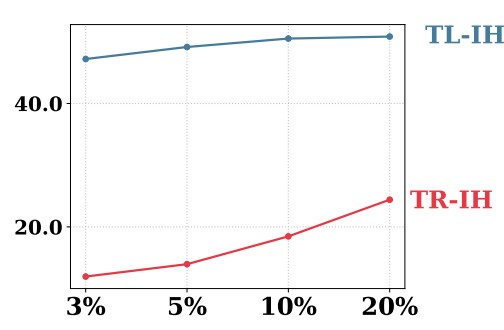

(a) Jaccard Coefficient, Kendall's $\tau$, and Spearman's $\rho$ values for TR heads, TL heads, and IHs.

(b) Conditional Average Percentage at four top levels for TR-IH and TL-IH pairs.

Figure 34: Results of overlap, correlation, and consistency analysis of attention head types averaged across datasets on Qwen2.5-32B.

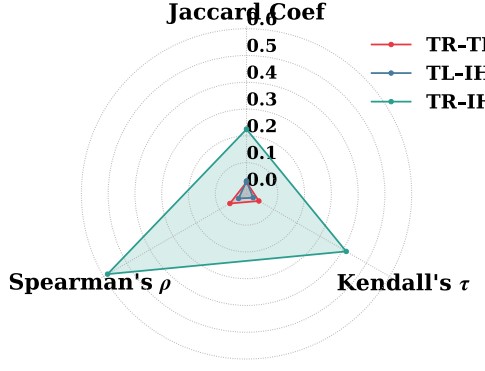

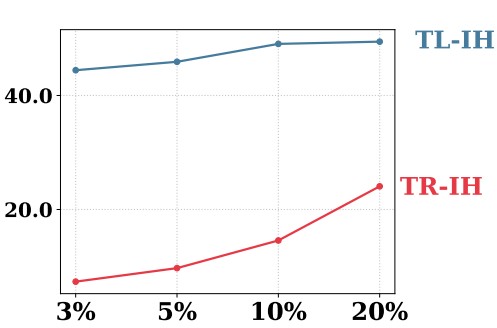

(a) Jaccard Coefficient, Kendall's $\tau$, and Spearman's $\rho$ values for TR heads, TL heads, and IHs.

(b) Conditional Average Percentage at four top levels for TR-IH and TL-IH pairs.

Figure 35: Results of overlap, correlation, and consistency analysis of attention head types averaged across datasets on Yi-34B.

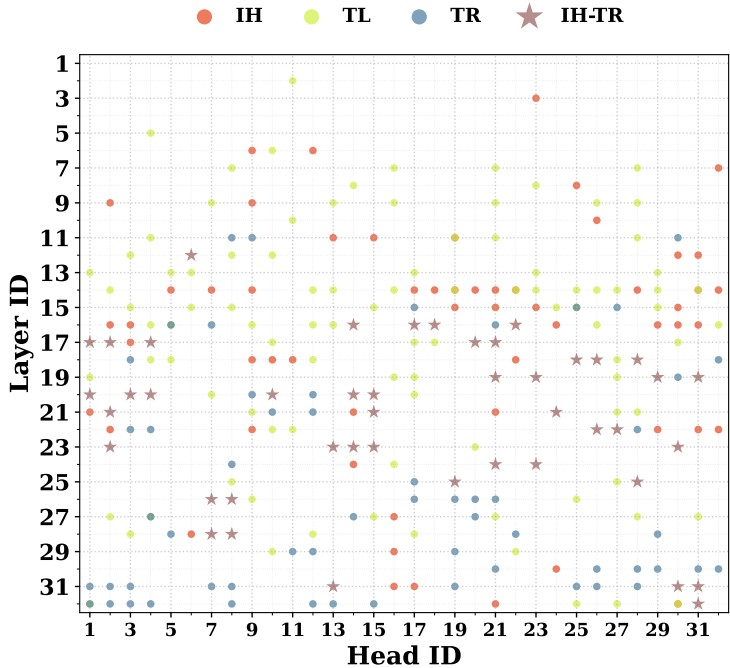

Figure 36: Distribution of the top 10% TR heads, TL heads, and IHs across layers for the SST-2 dataset on Llama3.1-8B.

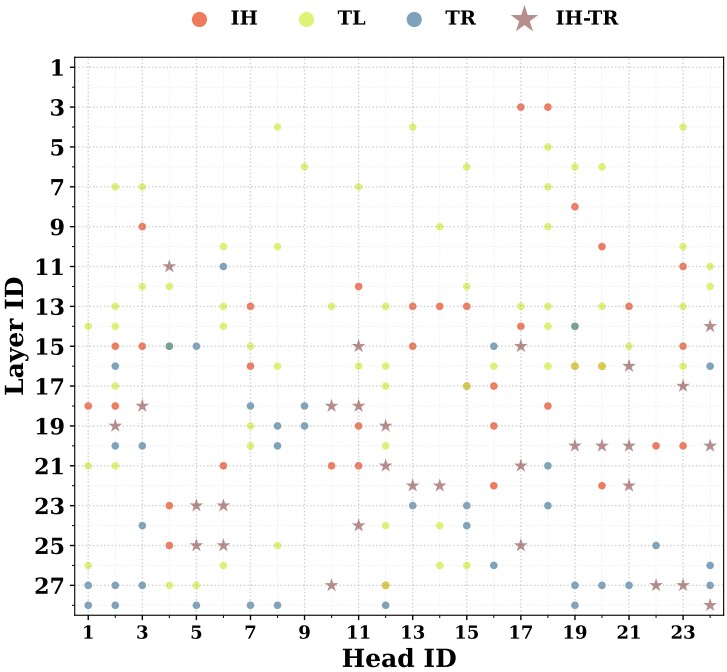

Figure 37: Distribution of the top 10% TR heads, TL heads, and IHs across layers for the SST-2 dataset on Llama3.2-3B.

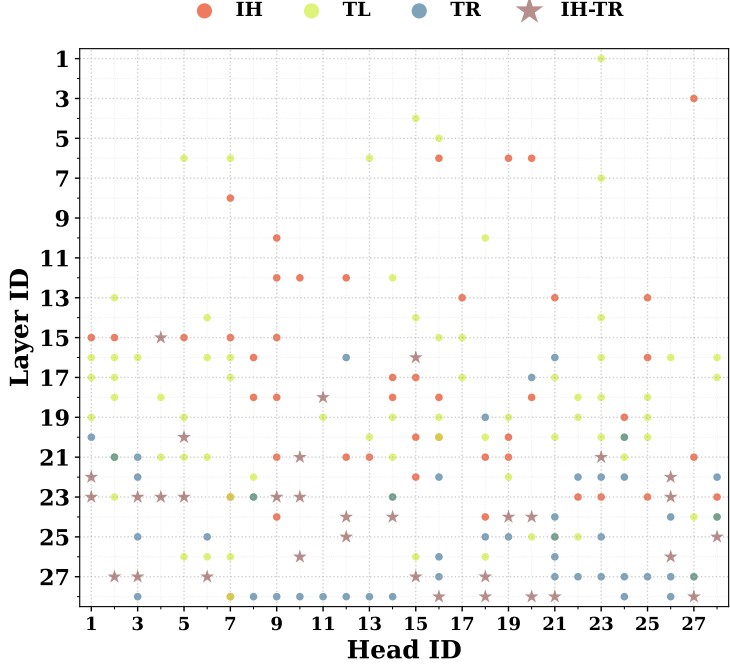

Figure 38: Distribution of the top 10% TR heads, TL heads, and IHs across layers for the SST-2 dataset on Qwen2-7B.

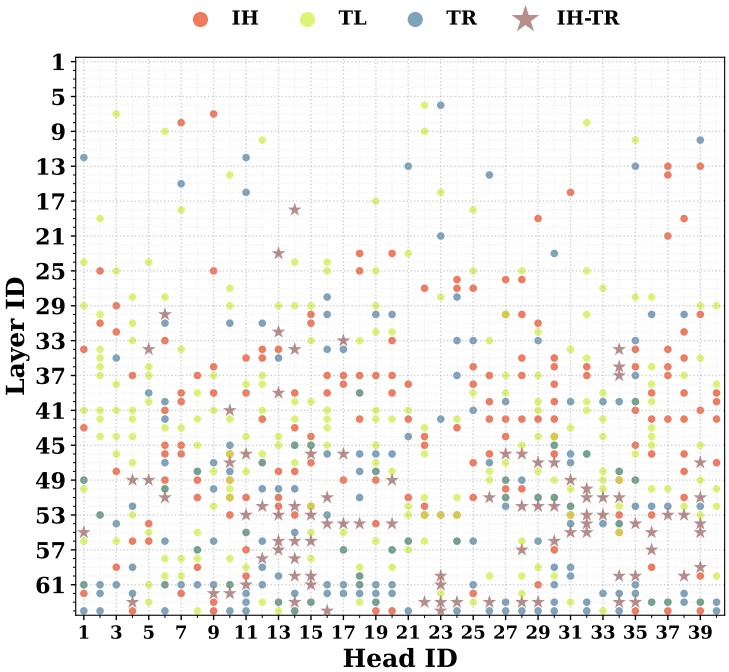

Figure 39: Distribution of the top 10% TR heads, TL heads, and IHs across layers for the SST-2 dataset on Qwen2.5-32B.

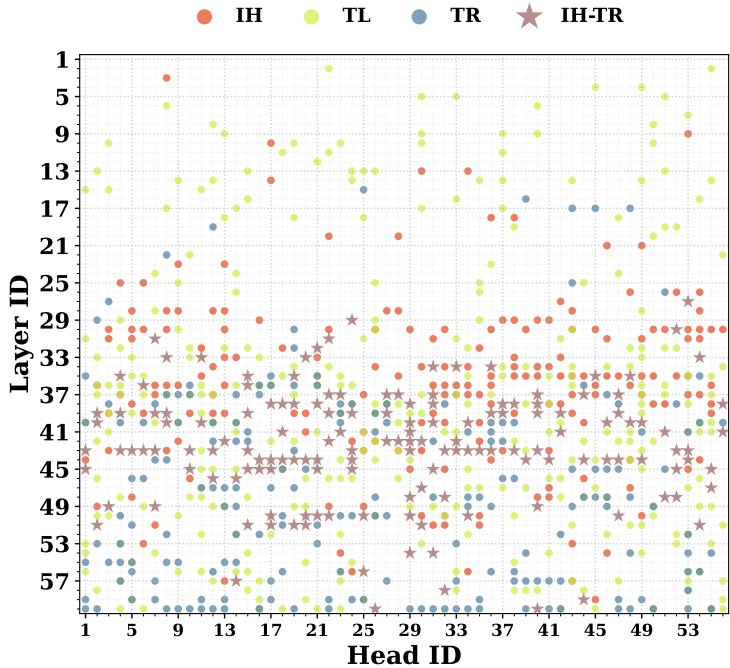

Figure 40: Distribution of the top 10% TR heads, TL heads, and IHs across layers for the SST-2 dataset on Yi-34B.

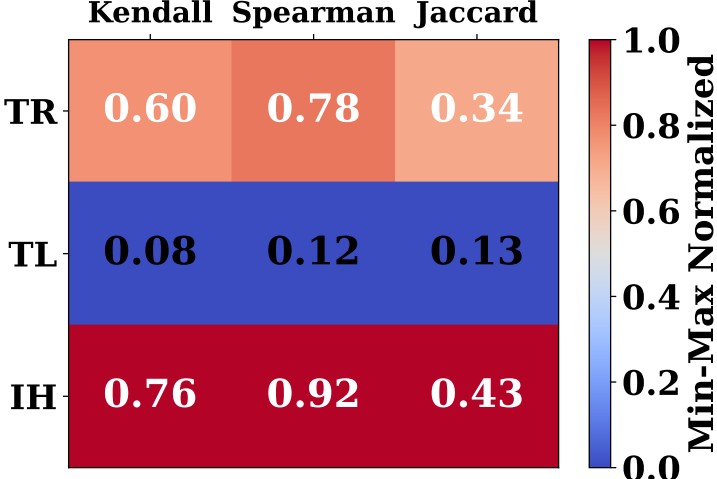

Figure 41: Overlap and correlation of the TR heads, TL heads, and IHs across datasets on Llama3.1-8B.

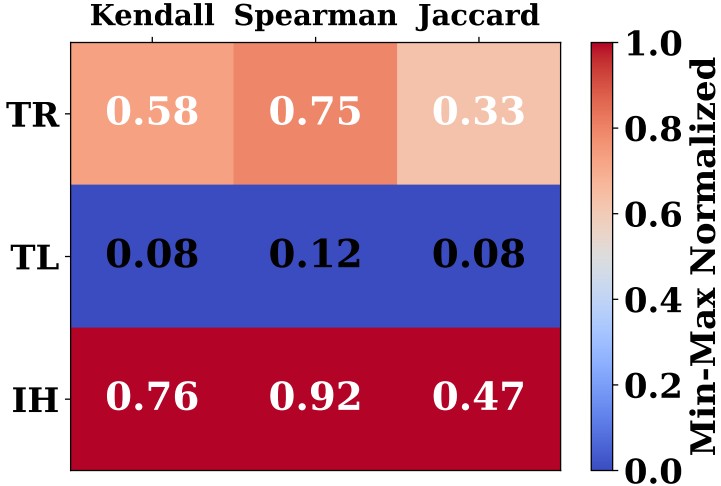

Figure 42: Overlap and correlation of the TR heads, TL heads, and IHs across datasets on Llama3.2-3B.

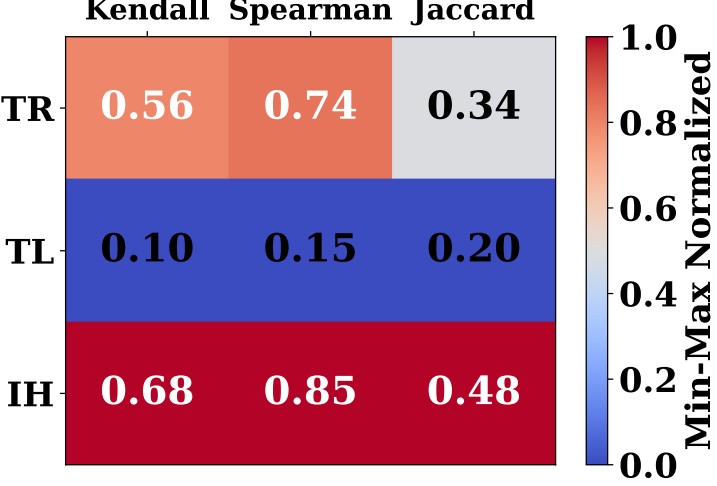

Figure 43: Overlap and correlation of the TR heads, TL heads, and IHs across datasets on Qwen2-7B.

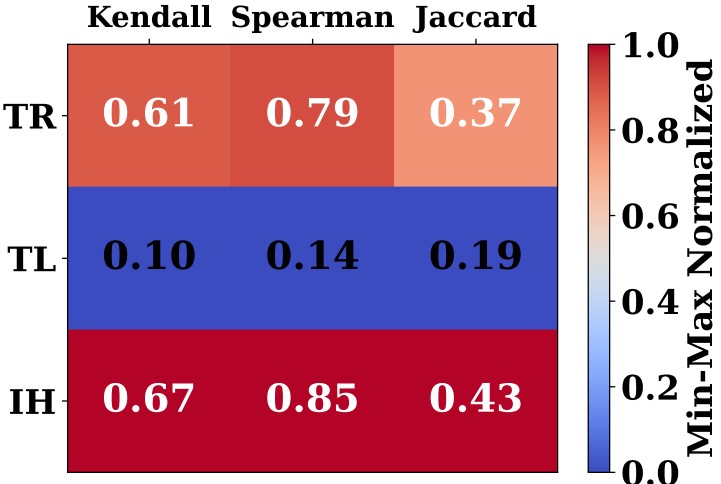

Figure 44: Overlap and correlation of the TR heads, TL heads, and IHs across datasets on Qwen2.5-32B.

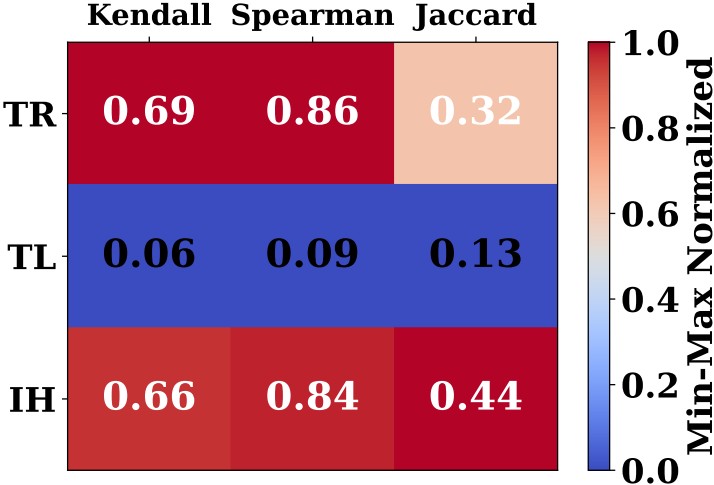

Figure 45: Overlap and correlation of the TR heads, TL heads, and IHs across datasets on Yi-34B.

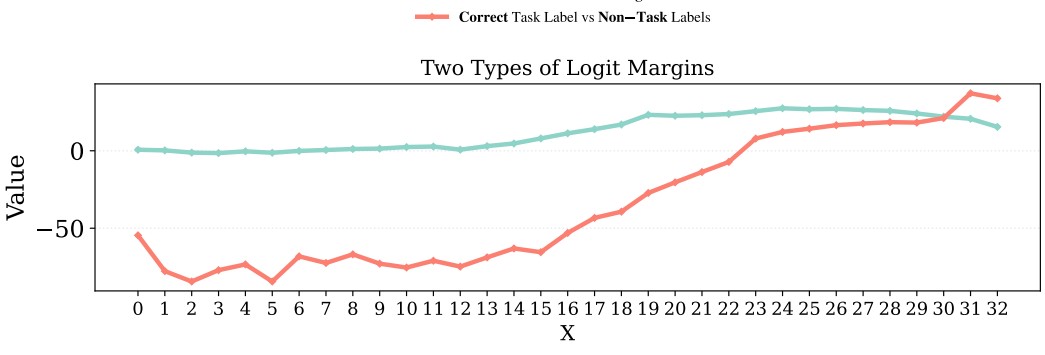

Figure 46: Dynamics of logit margin between 1) correct and incorrect task labels and 2)task labels and task-irrelevant labels across layers on Llama3-8B.

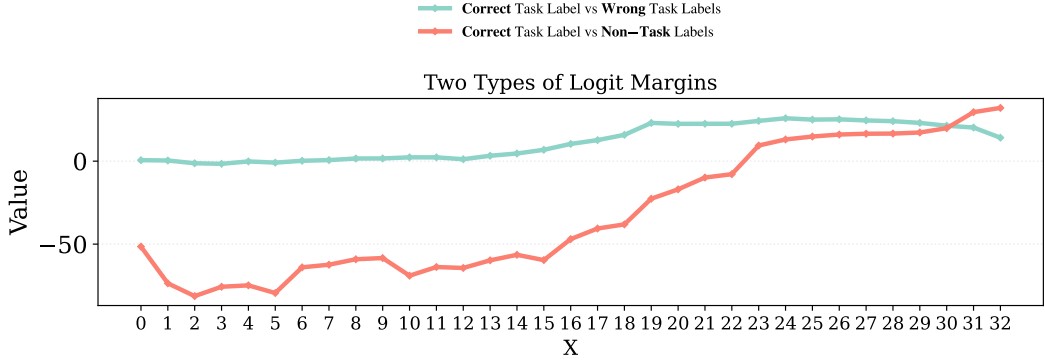

Figure 47: Dynamics of logit margin between 1) correct and incorrect task labels and 2)task labels and task-irrelevant labels across layers on Llama3.1-8B.

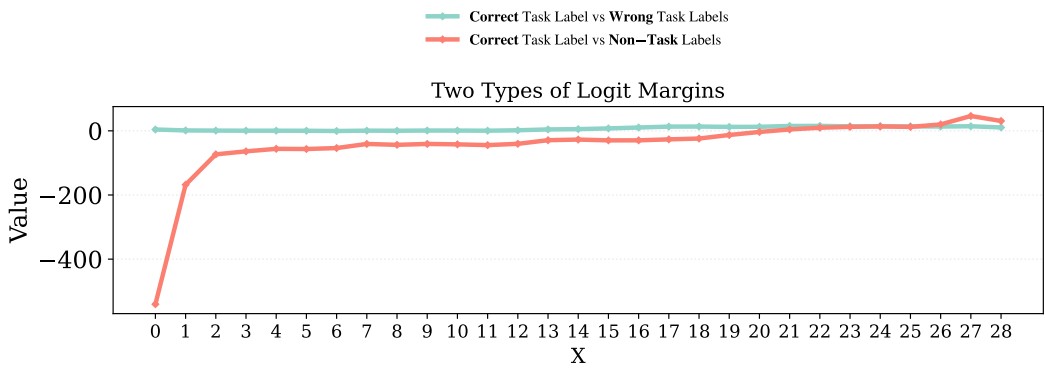

Figure 48: Dynamics of logit margin between 1) correct and incorrect task labels and 2)task labels and task-irrelevant labels across layers on Llama3.2-3B.

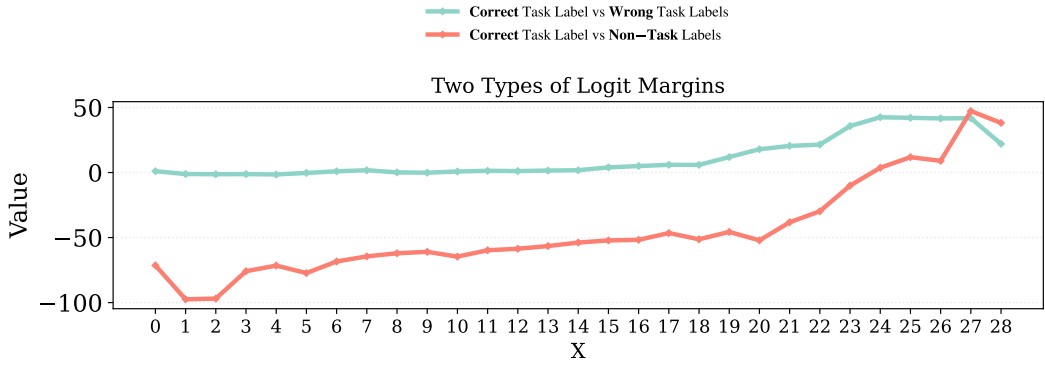

Figure 49: Dynamics of logit margin between 1) correct and incorrect task labels and 2)task labels and task-irrelevant labels across layers on Qwen2-7B.

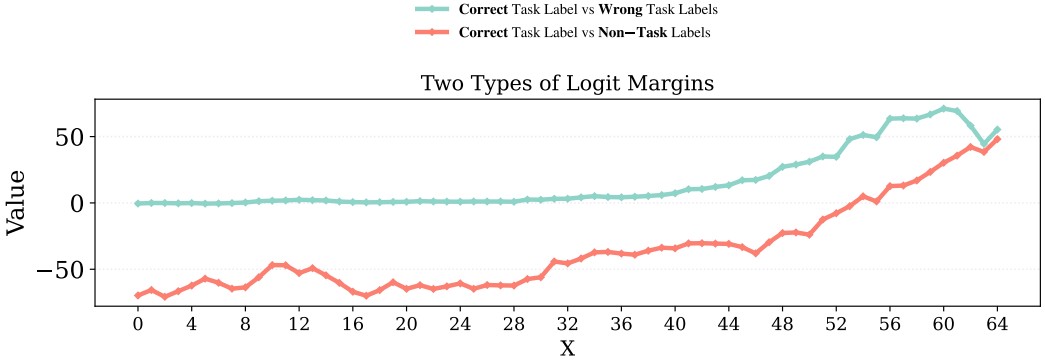

Figure 50: Dynamics of logit margin between 1) correct and incorrect task labels and 2)task labels and task-irrelevant labels across layers on Qwen2.5-32B.

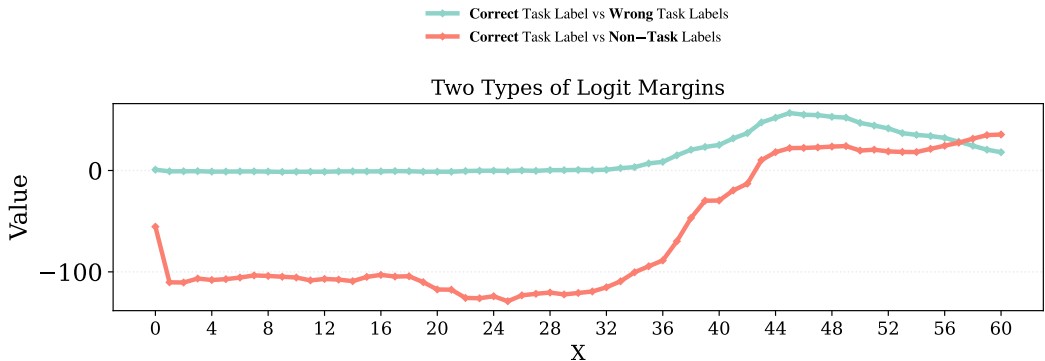

Figure 51: Dynamics of logit margin between 1) correct and incorrect task labels and 2)task labels and task-irrelevant labels across layers on Yi-34B.

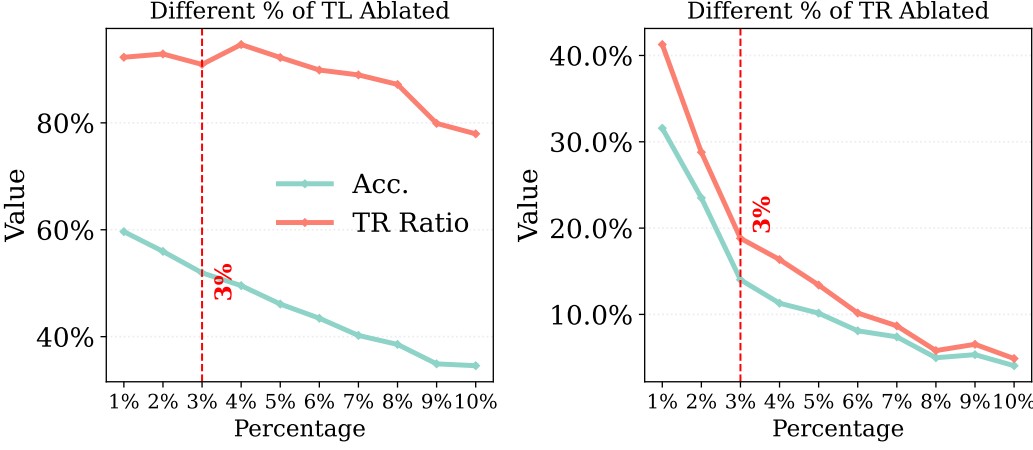

Figure 52: Dataset average accuracy and TR ratio resulted from ablating TL and TR heads at top percentage levels from 1% to 10%.

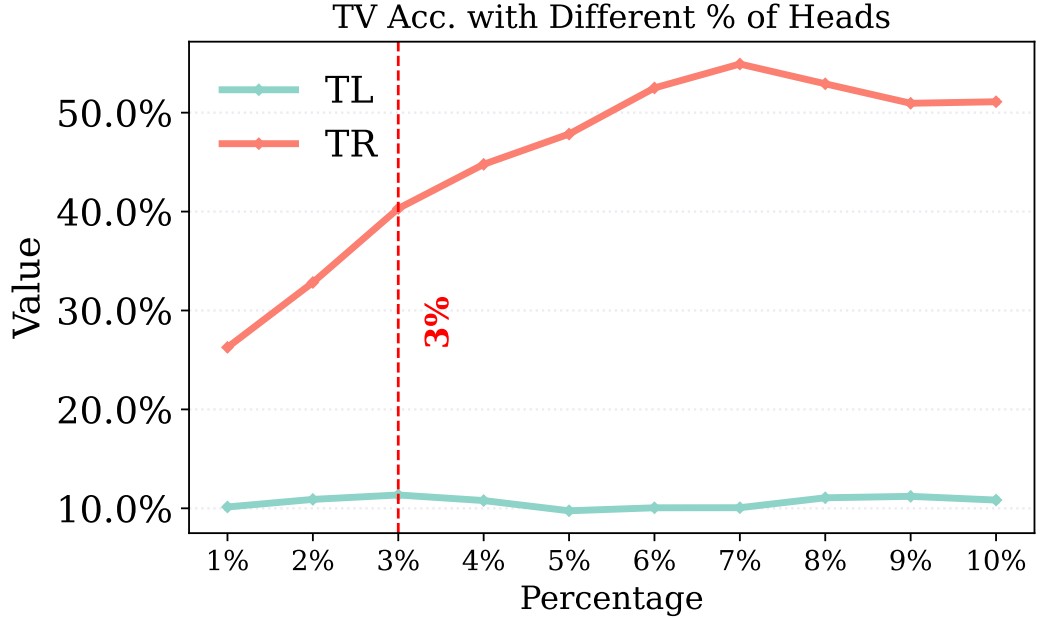

Figure 53: Dataset average accuracy and TR ratio resulted from using the outputs TL and TR heads at top percentage levels from 1% to 10% as task vectors.

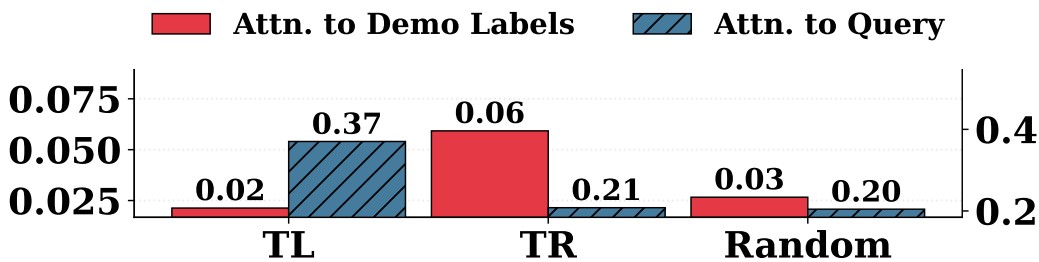

Figure 54: Dataset average cumulative attention weights assigned by the top TL, TR, and random heads of Llama3.1-8B to the demonstration labels and query tokens.

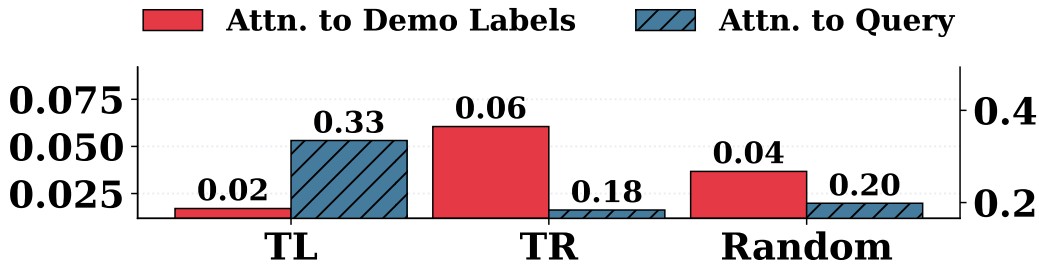

Figure 55: Dataset average cumulative attention weights assigned by the top TL, TR, and random heads of Llama3.2-3B to the demonstration labels and query tokens.

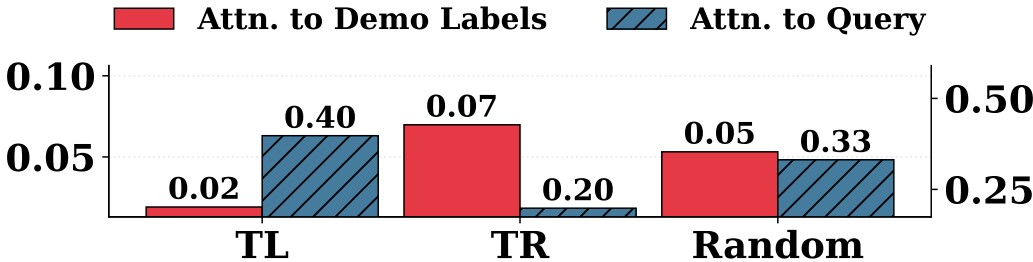

Figure 56: Dataset average cumulative attention weights assigned by the top TL, TR, and random heads of Qwen2-7B to the demonstration labels and query tokens.

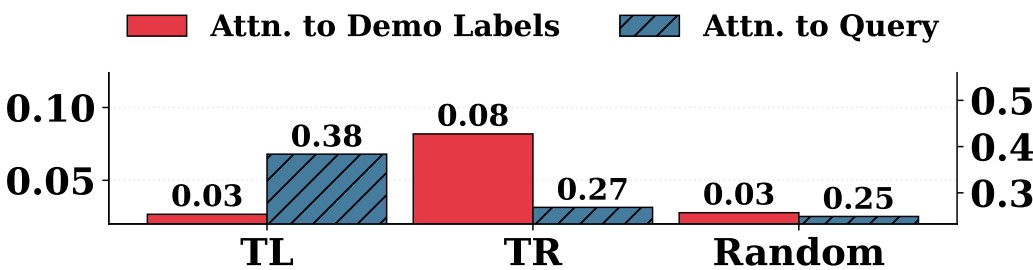

Figure 57: Dataset average cumulative attention weights assigned by the top TL, TR, and random heads of Qwen2.5-32B to the demonstration labels and query tokens.

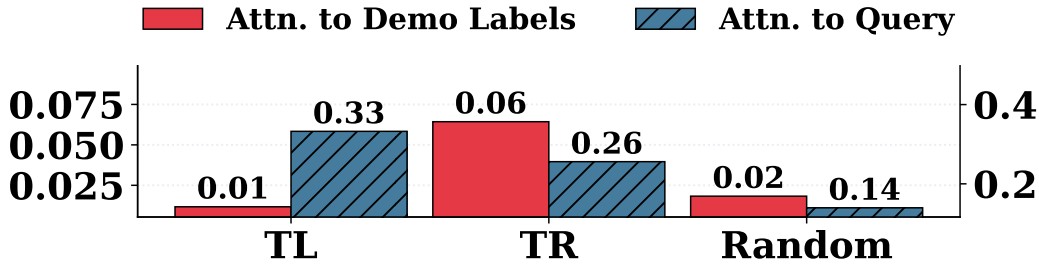

Figure 58: Dataset average cumulative attention weights assigned by the top TL, TR, and random heads of Yi-34B to the demonstration labels and query tokens.

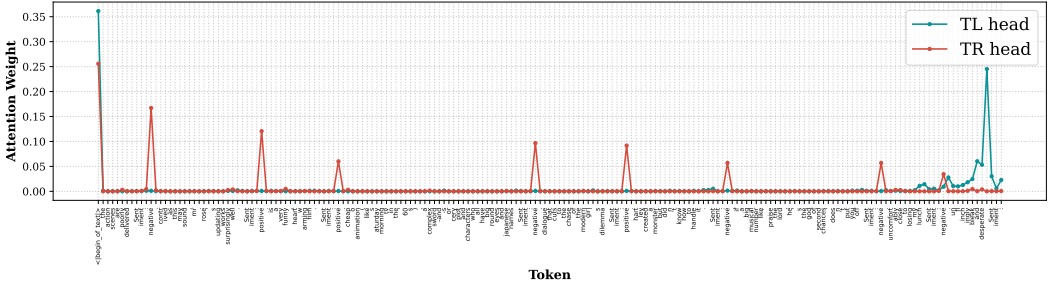

Figure 59: Attention distributions of the top-1 Llama3-8B TL and TR heads on SST-2 over the ICL prompt formed using the second test query of SST-2.

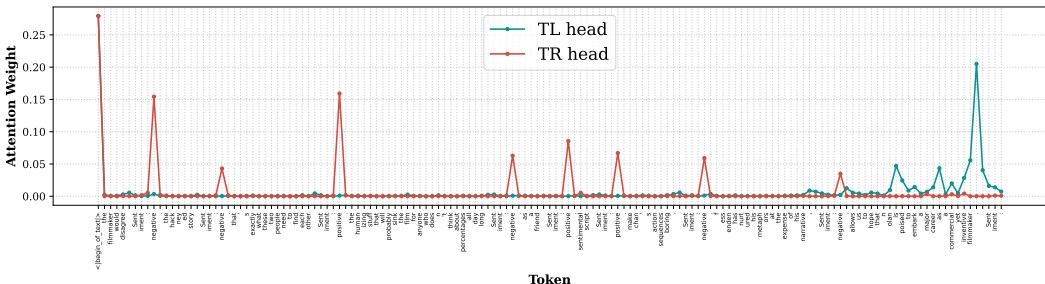

Figure 60: Attention distributions of the top-1 Llama3-8B TL and TR heads on SST-2 over the ICL prompt formed using the test query of SST-2.

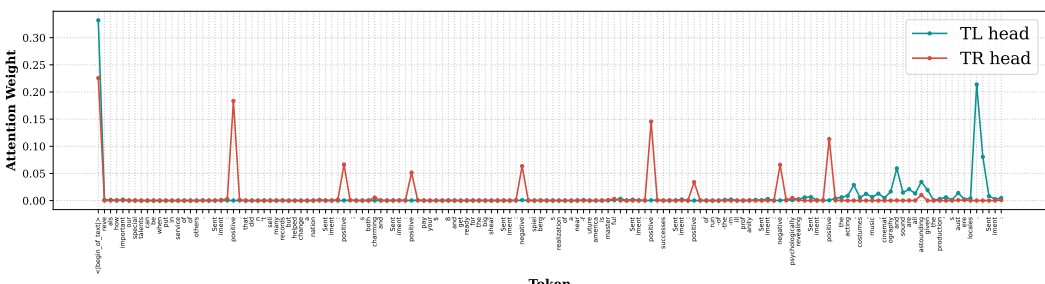

Figure 61: Attention distributions of the top-1 Llama3-8B TL and TR heads on SST-2 over the ICL prompt formed using the fourth test query of SST-2.

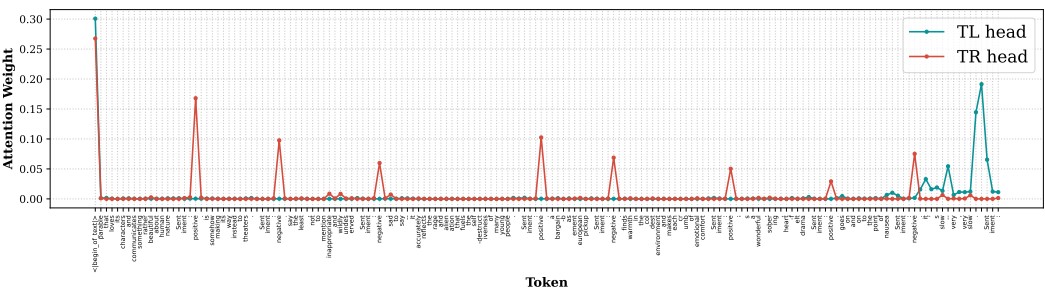

Figure 62: Attention distributions of the top-1 Llama3-8B TL and TR heads on SST-2 over the ICL prompt formed using the fifth test query of SST-2.

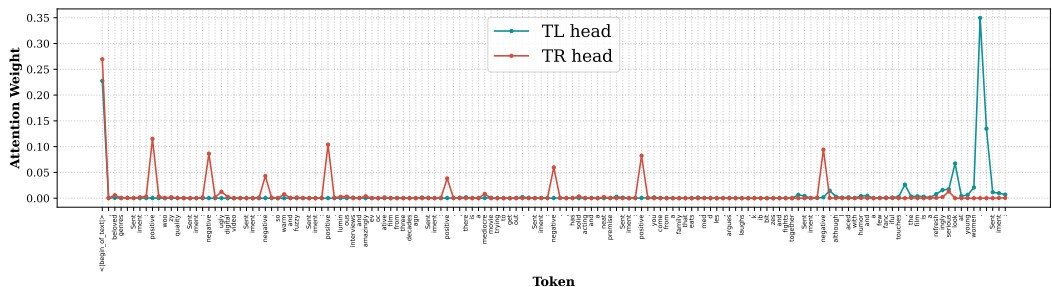

Figure 63: Attention distributions of the top-1 Llama3-8B TL and TR heads on SST-2 over the ICL prompt formed using the sixth test query of SST-2.

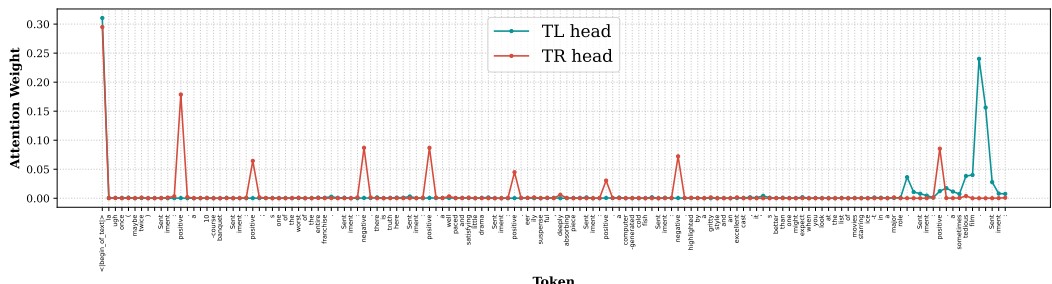

Figure 64: Attention distributions of the top-1 Llama3-8B TL and TR heads on SST-2 over the ICL prompt formed using the seventh test query of SST-2.

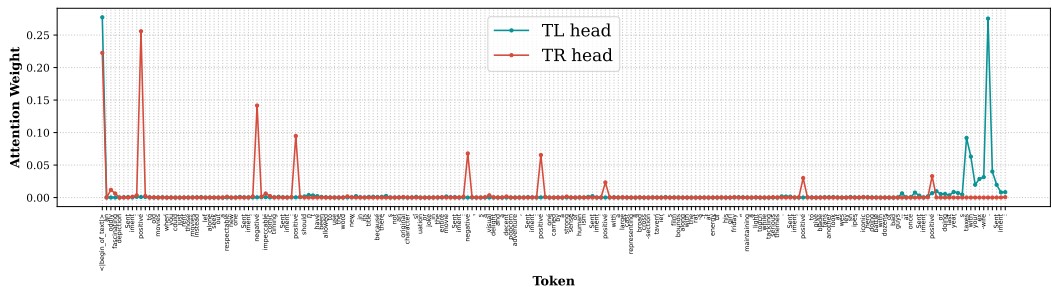

Figure 65: Attention distributions of the top-1 Llama3-8B TL and TR heads on SST-2 over the ICL prompt formed using the eighth test query of SST-2.

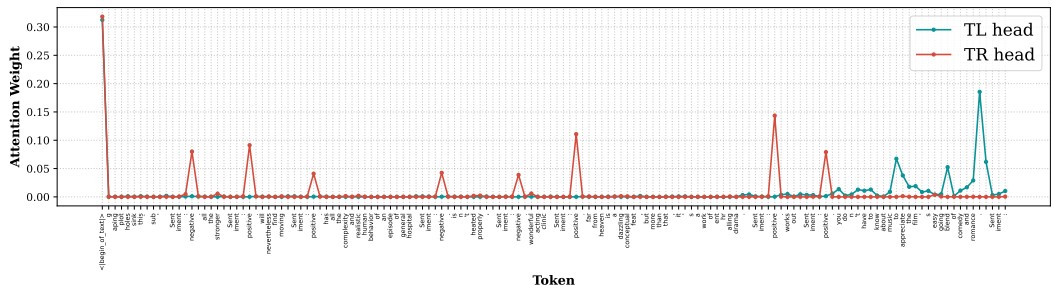

Figure 66: Attention distributions of the top-1 Llama3-8B TL and TR heads on SST-2 over the ICL prompt formed using the ninth test query of SST-2.

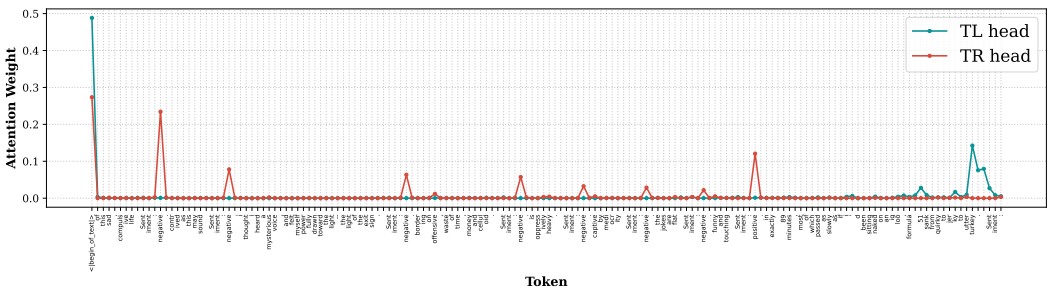

Figure 67: Attention distributions of the top-1 Llama3-8B TL and TR heads on SST-2 over the ICL prompt formed using the tenth test query of SST-2.

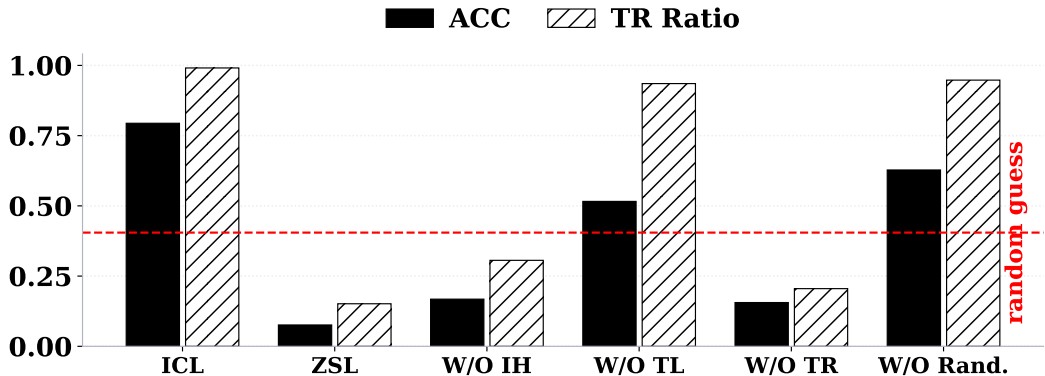

Figure 68: Effects of ablating the top 3% of TR, TL, and IH heads across datasets on Llama3.1-8B.

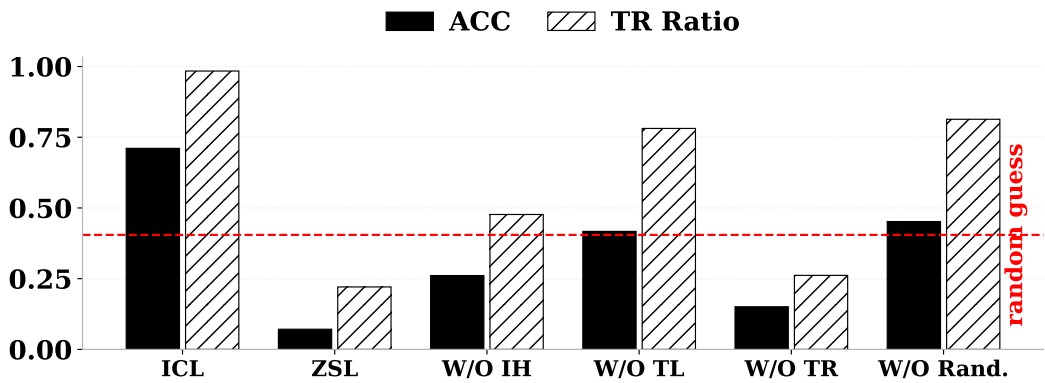

Figure 69: Effects of ablating the top 3% of TR, TL, and IH heads across datasets on Llama3.2-3B.

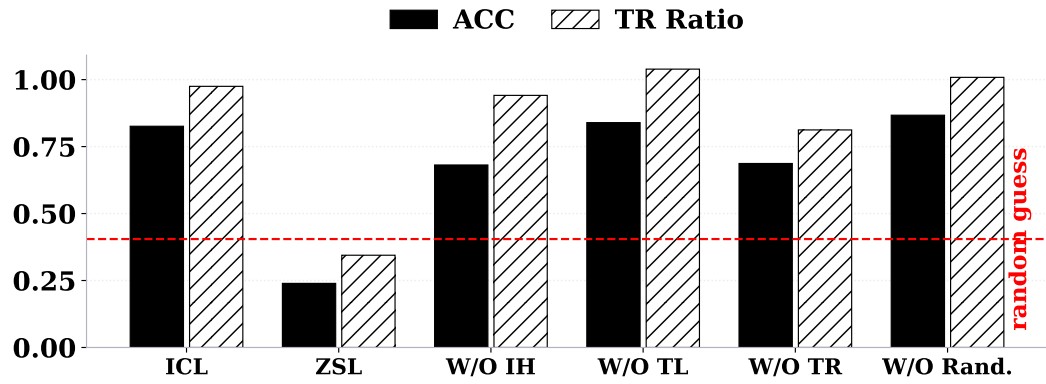

Figure 70: Effects of ablating the top 3% of TR, TL, and IH heads across datasets on Qwen2-7B.

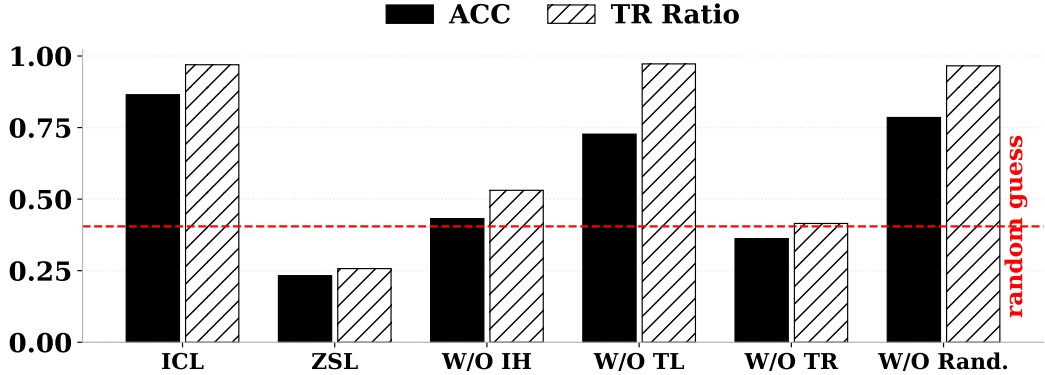

Figure 71: Effects of ablating the top 3% of TR, TL, and IH heads across datasets on Qwen2.5-32B.

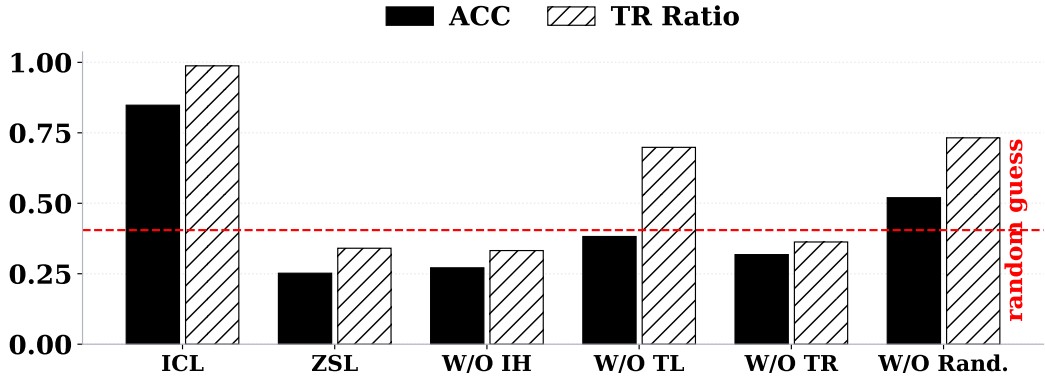

Figure 72: Effects of ablating the top 3% of TR, TL, and IH heads across datasets on Yi-34B.

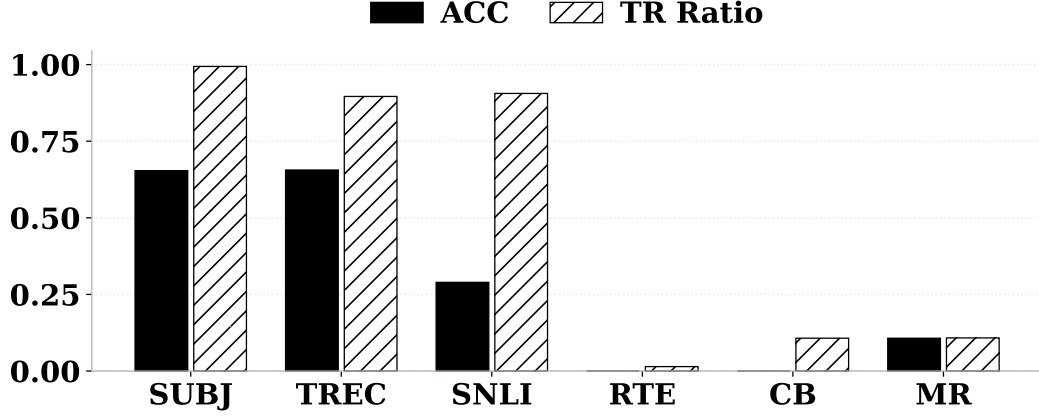

Figure 73: Effects of ablating TR heads identified on SST-2 when transferred to other datasets using Llama3-8B.

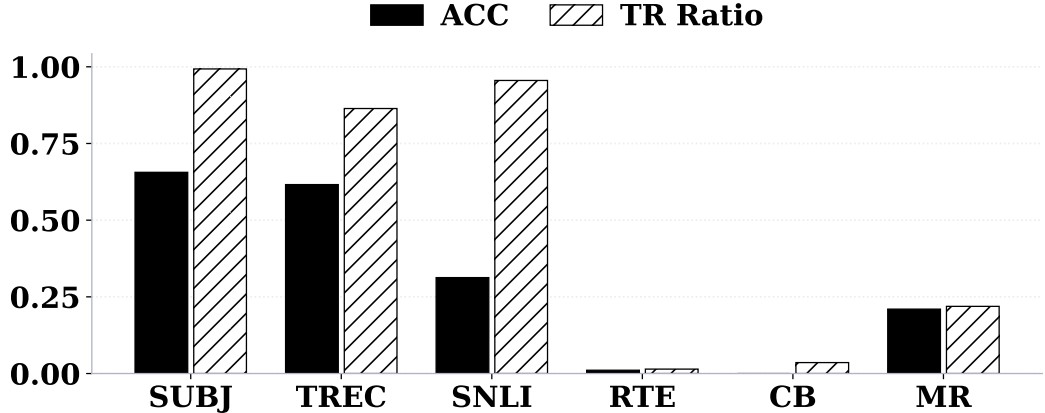

Figure 74: Effects of ablating TR heads identified on SST-2 when transferred to other datasets using Llama3.1-8B.

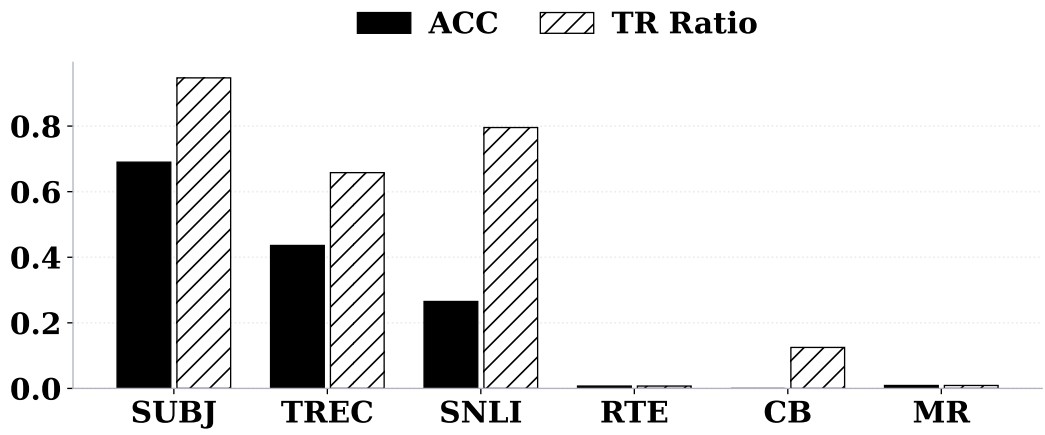

Figure 75: Effects of ablating TR heads identified on SST-2 when transferred to other datasets using Llama3.2-3B.

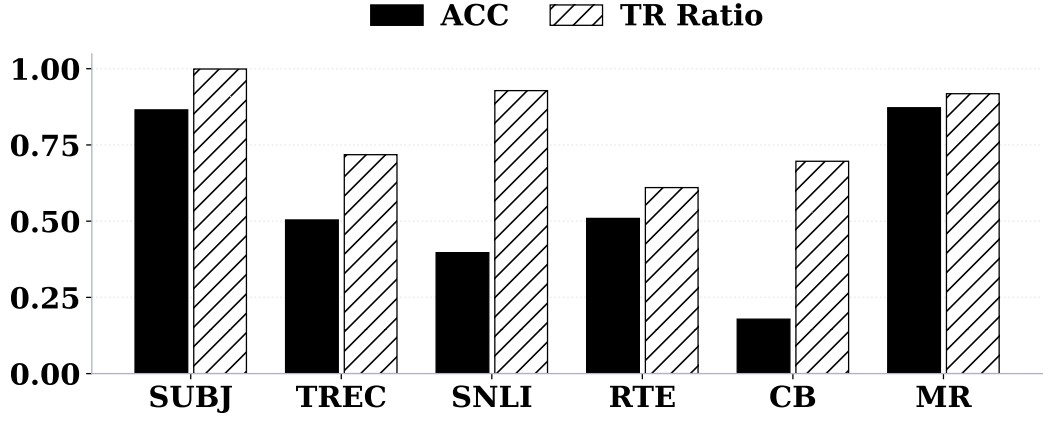

Figure 76: Effects of ablating TR heads identified on SST-2 when transferred to other datasets using Qwen2-7B.

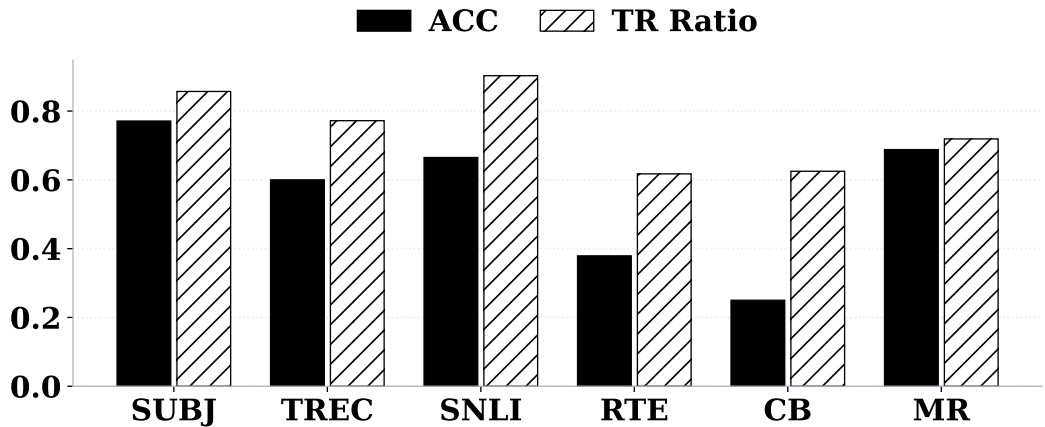

Figure 77: Effects of ablating TR heads identified on SST-2 when transferred to other datasets using Qwen2.5-32B.

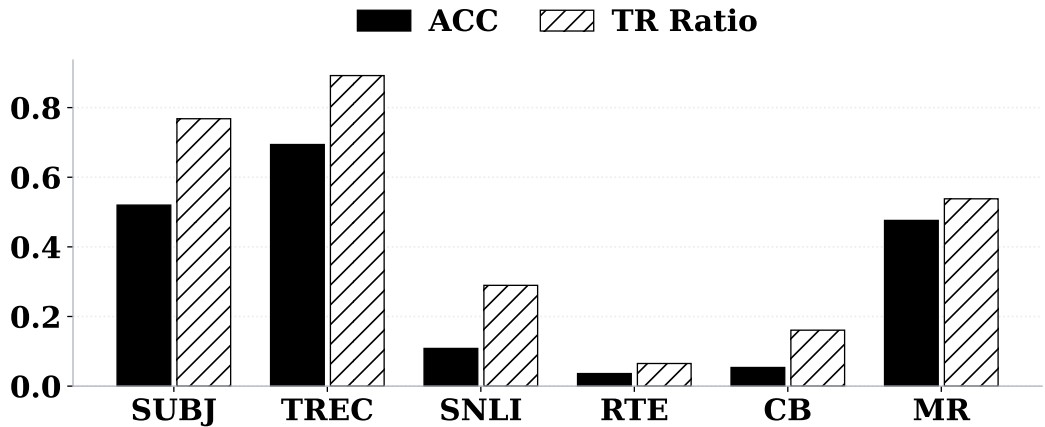

Figure 78: Effects of ablating TR heads identified on SST-2 when transferred to other datasets using Yi-34B.

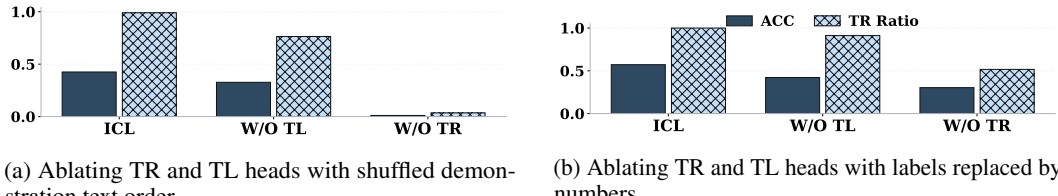

(a) Ablating TR and TL heads with shuffled demonstration text order.

(b) Ablating TR and TL heads with labels replaced by numbers.

Figure 79: Effects of ablating TR and TL heads under perturbed ICL inputs on Llama3.1-8B.

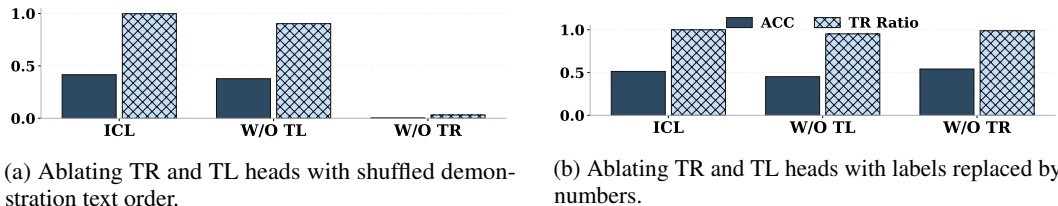

(a) Ablating TR and TL heads with shuffled demonstration text order.

(b) Ablating TR and TL heads with labels replaced by numbers.

Figure 80: Effects of ablating TR and TL heads under perturbed ICL inputs on Llama3.2-3B.

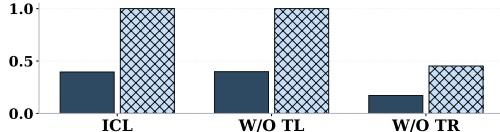
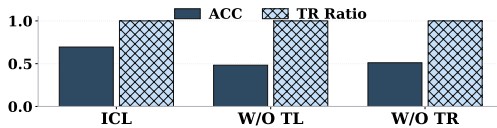

(a) Ablating TR and TL heads with shuffled demonstration text order.

(b) Ablating TR and TL heads with labels replaced by numbers.

Figure 81: Effects of ablating TR and TL heads under perturbed ICL inputs on Qwen2-7B.

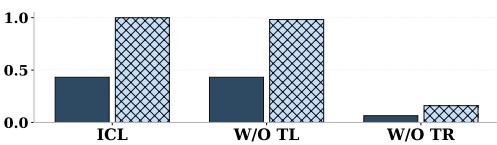
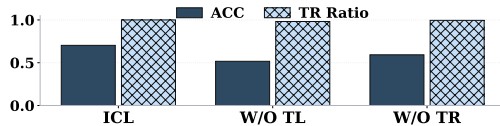

(a) Ablating TR and TL heads with shuffled demonstration text order.

(b) Ablating TR and TL heads with labels replaced by numbers.

Figure 82: Effects of ablating TR and TL heads under perturbed ICL inputs on Qwen2.5-32B.

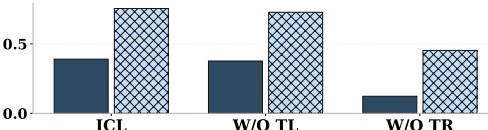
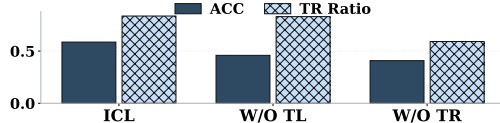

(a) Ablating TR and TL heads with shuffled demonstration text order.

(b) Ablating TR and TL heads with labels replaced by numbers.

Figure 83: Effects of ablating TR and TL heads under perturbed ICL inputs on Yi-34B.

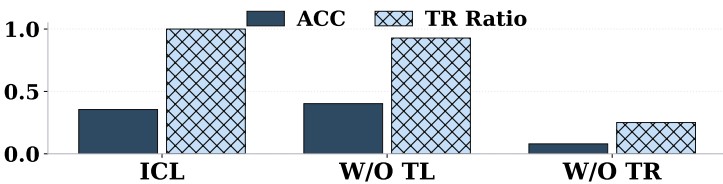

Figure 84: Effects of ablating TR and TL heads on Llama3-8B when demonstration labels are flipped.

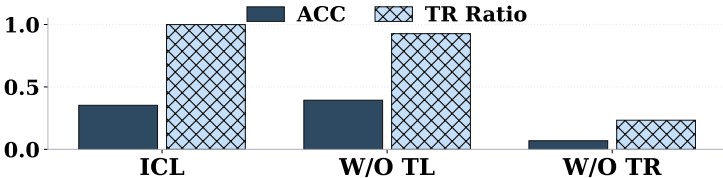

Figure 85: Effects of ablating TR and TL heads on Llama3.1-8B when demonstration labels are flipped.

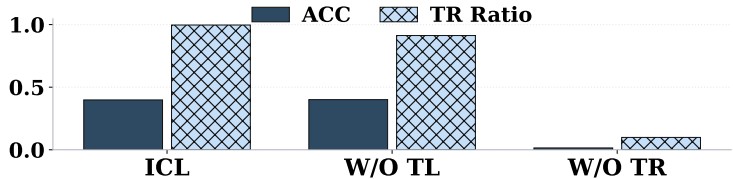

Figure 86: Effects of ablating TR and TL heads on Llama3.2-3B when demonstration labels are flipped.

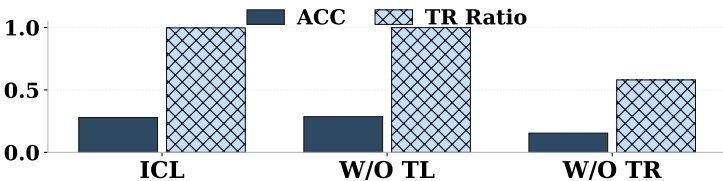

Figure 87: Effects of ablating TR and TL heads on Qwen2-7B when demonstration labels are flipped.

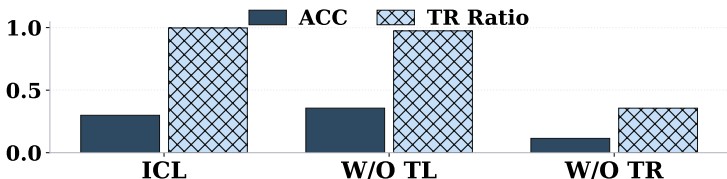

Figure 88: Effects of ablating TR and TL heads on Qwen2.5-32B when demonstration labels are flipped.

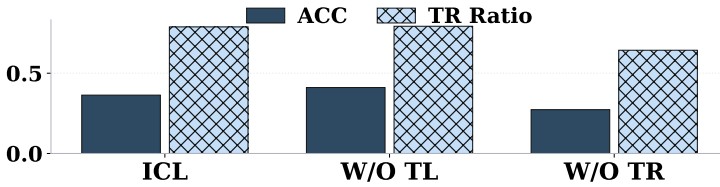

Figure 89: Effects of ablating TR and TL heads on Yi-34B when demonstration labels are flipped.

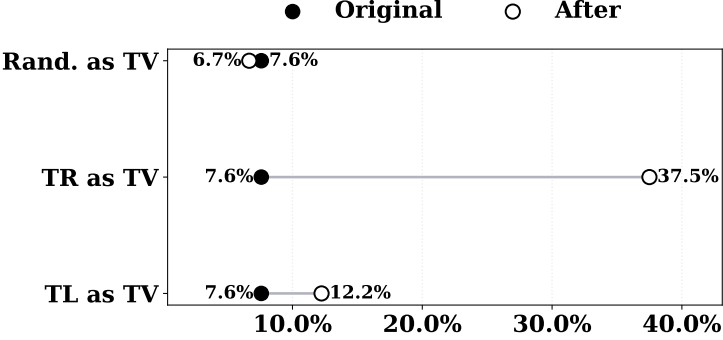

Figure 90: Steering zero-shot hidden states of Llama3.1-8B using task vectors from TR, TL, or random heads.

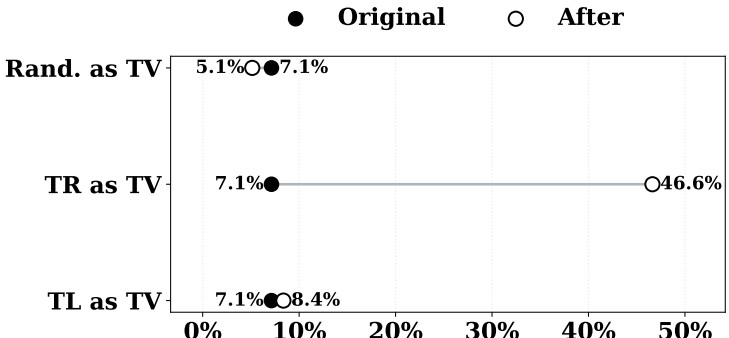

Figure 91: Steering zero-shot hidden states of Llama3.2-3B using task vectors from TR, TL, or random heads.

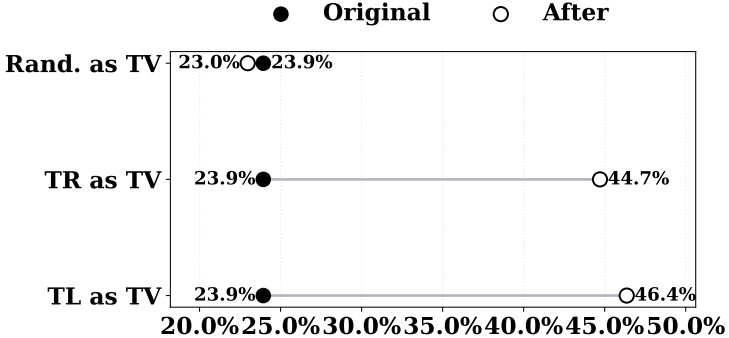

Figure 92: Steering zero-shot hidden states of Qwen2-7B using task vectors from TR, TL, or random heads.

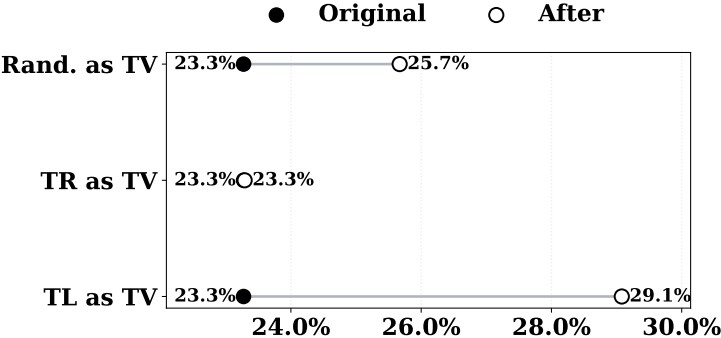

Figure 93: Steering zero-shot hidden states of Qwen2.5-32B using task vectors from TR, TL, or random heads.

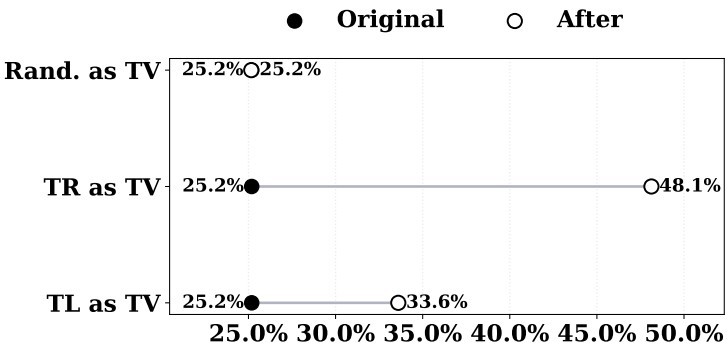

Figure 94: Steering zero-shot hidden states of Yi-34B using task vectors from TR, TL, or random heads.

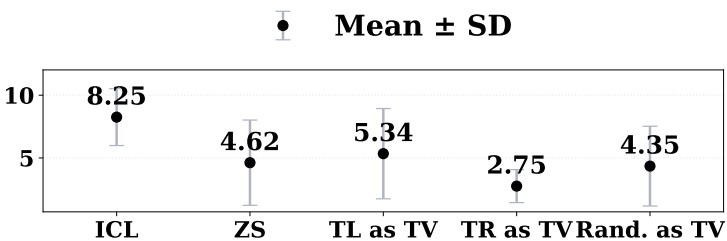

Figure 95: Mean and standard deviation of review ratings with Llama3.1-8B when task vectors from different head types are applied.

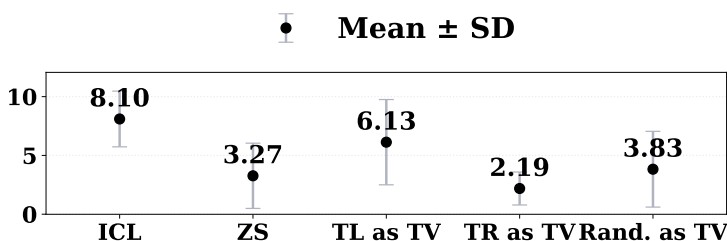

Figure 96: Mean and standard deviation of review ratings with Llama3.2-3B when task vectors from different head types are applied.

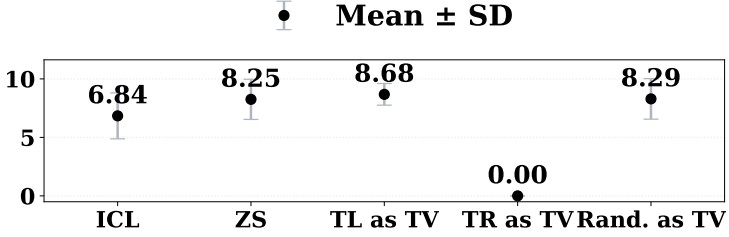

Figure 97: Mean and standard deviation of review ratings with Qwen2-7B when task vectors from different head types are applied.

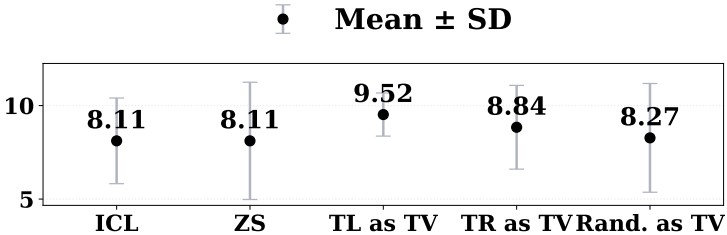

Figure 98: Mean and standard deviation of review ratings with Qwen2.5-32B when task vectors from different head types are applied.

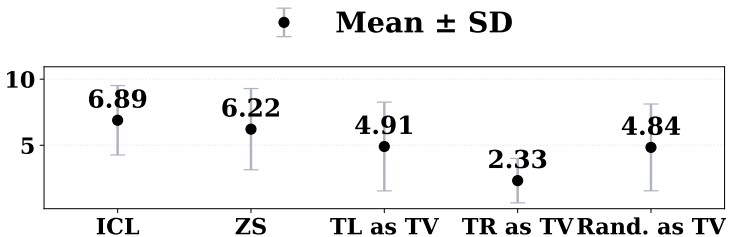

Figure 99: Mean and standard deviation of review ratings with Yi-34B when task vectors from different head types are applied.

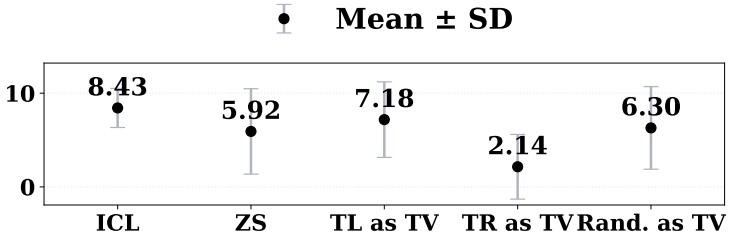

Figure 100: Mean and standard deviation of the ratings of book reviews generated using Llama3-8B when task vectors from different head types identified on the SubjQA dataset are applied.

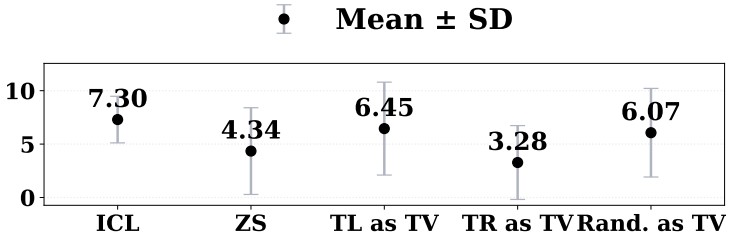

Figure 101: Mean and standard deviation of the ratings of book reviews generated using Llama3.1-8B when task vectors from different head types identified on the SubjQA dataset are applied.

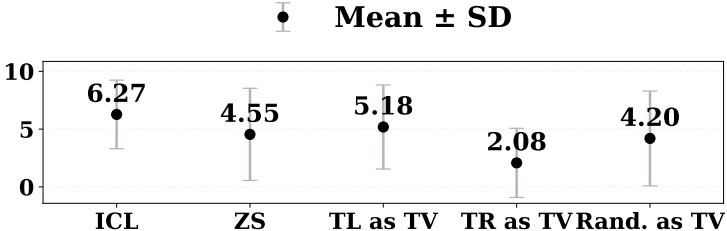

Figure 102: Mean and standard deviation of the ratings of book reviews generated using Llama3.2-3B when task vectors from different head types identified on the SubjQA dataset are applied.

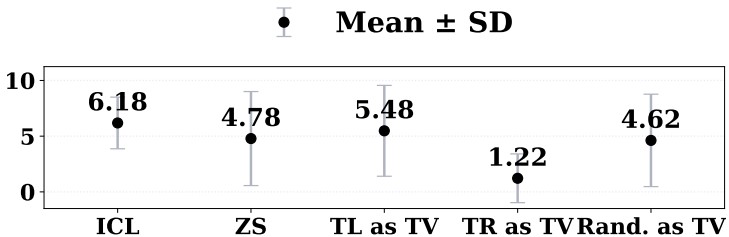

Figure 103: Mean and standard deviation of the ratings of book reviews generated using Qwen2-7B when task vectors from different head types identified on the SubjQA dataset are applied.

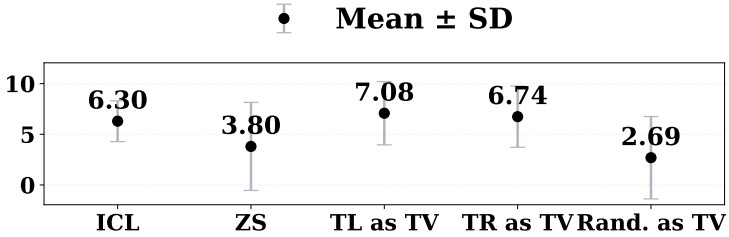

Figure 104: Mean and standard deviation of the ratings of book reviews generated using Qwen2.5-32B when task vectors from different head types identified on the SubjQA dataset are applied.

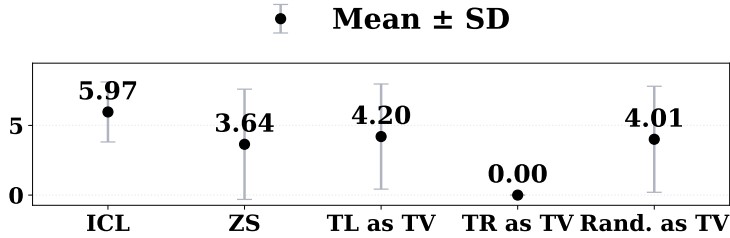

Figure 105: Mean and standard deviation of the ratings of book reviews generated using Yi-34B when task vectors from different head types identified on the SubjQA dataset are applied.

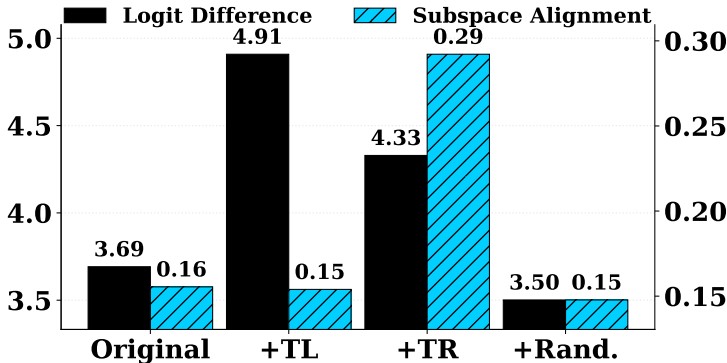

Figure 106: Geometric effects of TR and TL head outputs on hidden states in Llama3.1-8B.

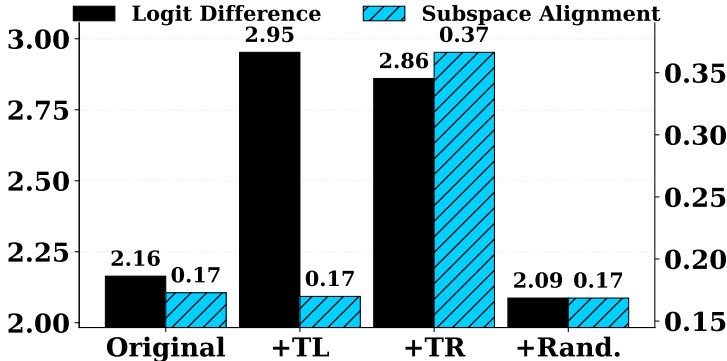

Figure 107: Geometric effects of TR and TL head outputs on hidden states in Llama3.2-3B.

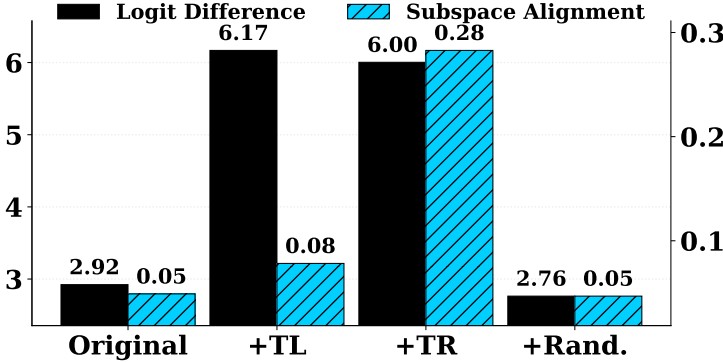

Figure 108: Geometric effects of TR and TL head outputs on hidden states in Qwen2-7B.

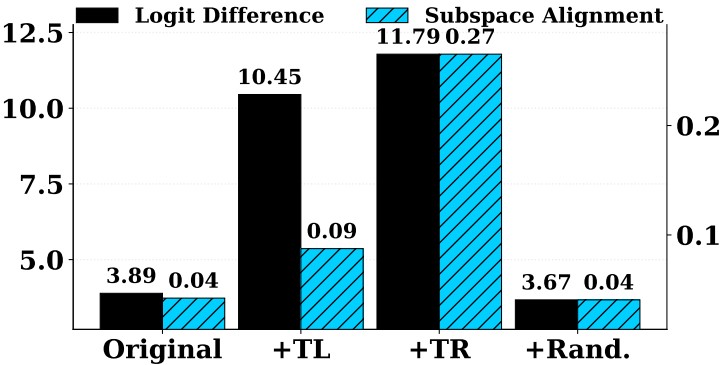

Figure 109: Geometric effects of TR and TL head outputs on hidden states in Qwen2.5-32B.

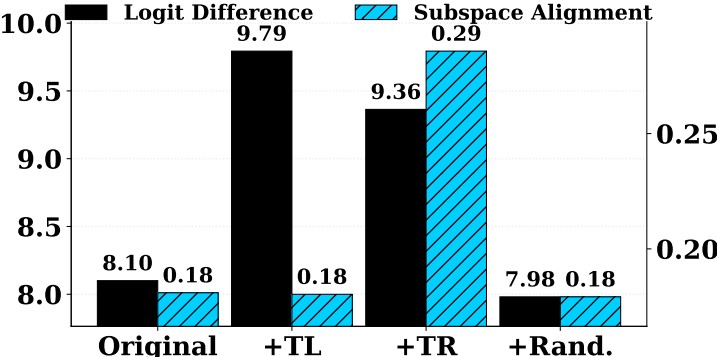

Figure 110: Geometric effects of TR and TL head outputs on hidden states in Yi-34B.

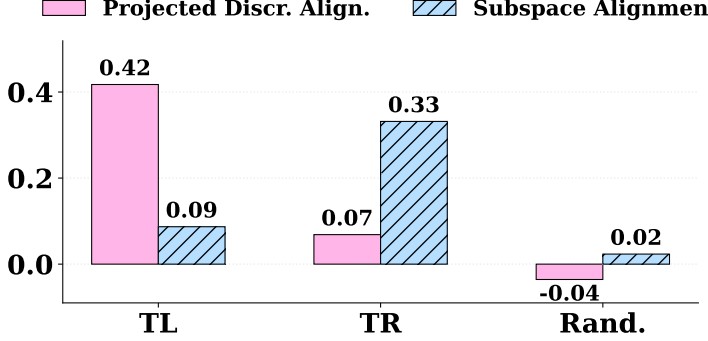

Figure 111: Impact of TL and TR head outputs on hidden states w.r.t. task subspace in Llama3.1-8B.

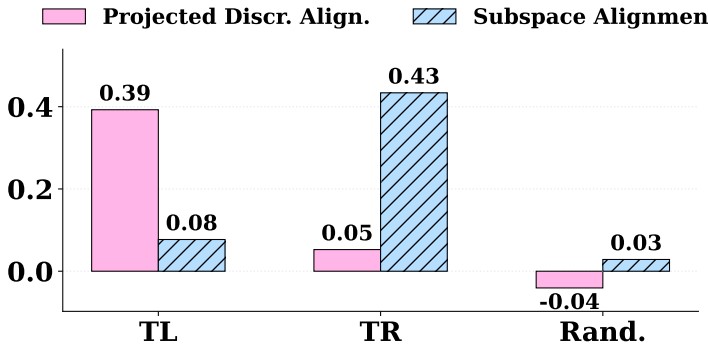

Figure 112: Impact of TL and TR head outputs on hidden states w.r.t. task subspace in Llama3.2-3B.

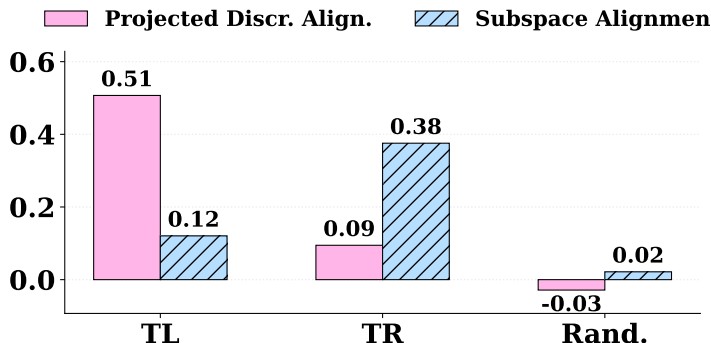

Figure 113: Impact of TL and TR head outputs on hidden states w.r.t. task subspace in Qwen2-7B.

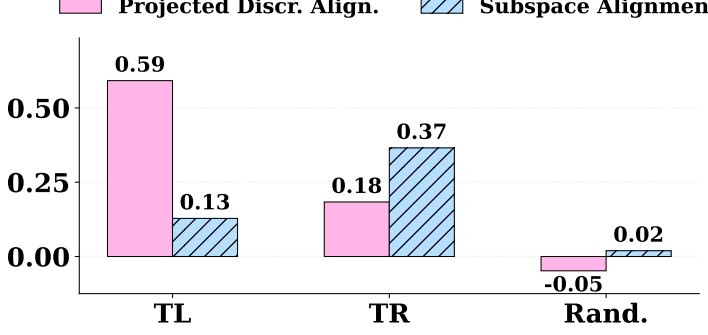

Figure 114: Impact of TL and TR head outputs on hidden states w.r.t. task subspace in Qwen2.5-32B.

Table 4: Jaccard coefficient and Spearman's $\rho$ for TL and TR scores across tasks.

|  | TL Task 1&2 | **TR Task 1&2** |
|---|---|---|
| Jaccard coefficient | 0.1860 | **0.3421** |
| Spearman's $\rho$ | 0.2980 | **0.8365** |

| **Llama3-8B** | **Llama3.1-8B** | **Llama3.2-3B** | **Qwen2-7B** | **Qwen2.5B** | **Yi-34B** |
|---|---|---|---|---|---|
| $1.96 \times 10^{-4}$ | $1.96 \times 10^{-4}$ | $7.63 \times 10^{-6}$ | $7.63 \times 10^{-6}$ | $7.63 \times 10^{-6}$ | $7.37 \times 10^{-4}$ |

Table 5: Statistical significance of the difference between the norms of token unembeddings projected to different subspaces across models.

|  | TL heads mean layer | IHs mean layer | TR heads mean layer | TL < IHs? | IHs < TR heads? |
|---|---|---|---|---|---|
| 0.03 | 16.89 | 16.69 | 26.22 | 0.84985 | 5.6052e-45 |
| 0.05 | 16.94 | 16.74 | 25.08 | 0.80371 | 0.0000e+00 |
| 0.10 | 16.75 | 16.88 | 23.27 | 0.59518 | 0.0000e+00 |

Table 6: Mean layer index of TL heads, IHs, and TR heads across datasets on Llama3-8B, with p-values for distribution differences.

|  | TL heads mean layer | IHs mean layer | TR heads mean layer | TL < IHs? | IHs < TR heads? |
|---|---|---|---|---|---|
| 0.03 | 16.85 | 16.67 | 26.07 | 0.71998 | 8.4078e-45 |
| 0.05 | 16.36 | 16.77 | 25.33 | 0.22671 | 0.0000e+00 |
| 0.10 | 16.12 | 16.79 | 23.25 | 0.049241 | 0.0000e+00 |

Table 7: Mean layer index of TL heads, IHs, and TR heads across datasets on Llama3.1-8B, with p-values for distribution differences.

|  | TL heads mean layer | IHs mean layer | TR heads mean layer | TL < IHs? | IHs < TR heads? |
|---|---|---|---|---|---|
| 0.03 | 14.51 | 15.69 | 23.04 | 0.040803 | 1.3459e-26 |
| 0.05 | 14.25 | 15.78 | 22.59 | 0.0024930 | 3.0489e-35 |
| 0.10 | 14.38 | 15.75 | 20.96 | 0.00044495 | 1.0738e-36 |

Table 8: Mean layer index of TL heads, IHs, and TR heads across datasets on Llama3.2-3B, with p-values for distribution differences.

|  | TL heads mean layer | IHs mean layer | TR heads mean layer | TL < IHs? | IHs < TR heads? |
|---|---|---|---|---|---|
| 0.03 | 19.06 | 18.57 | 24.87 | 0.31254 | 2.6148e-28 |
| 0.05 | 18.01 | 18.62 | 24.64 | 0.37007 | 1.7432e-42 |
| 0.10 | 16.22 | 18.97 | 23.38 | 2.4055e-09 | 1.8049e-41 |

Table 9: Mean layer index of TL heads, IHs, and TR heads across datasets on Qwen2-7B, with p-values for distribution differences.

|  | TL heads mean layer | IHs mean layer | TR heads mean layer | TL < IHs? | IHs < TR heads? |
|---|---|---|---|---|---|
| 0.03 | 46.26 | 44.08 | 56.80 | 0.0012338 | 0.0000e+00 |
| 0.05 | 43.19 | 44.53 | 54.89 | 0.35802 | 0.0000e+00 |
| 0.10 | 39.42 | 45.11 | 51.36 | 1.7597e-20 | 0.0000e+00 |

Table 10: Mean layer index of TL heads, IHs, and TR heads across datasets on Qwen2.5-32B, with p-values for distribution differences.

| | TL heads mean layer | IHs mean layer | TR heads mean layer | TL < IHs? | IHs < TR heads? |
|---|---|---|---|---|---|
| 0.03 | 36.64 | 38.16 | 47.21 | 0.071766 | 0.0000e+00 |
| 0.05 | 35.57 | 38.20 | 46.90 | 0.0011017 | 0.0000e+00 |
| 0.10 | 34.09 | 38.25 | 45.20 | 1.0199e-12 | 0.0000e+00 |

Table 11: Mean layer index of TL heads, IHs, and TR heads across datasets on Yi-34B, with p-values for distribution differences.

| Setting | Generated Review |
|---|---|
| ICL | Poignant character arcs explore relatable themes with depth. Cinematic score heightens emotional impact of pivotal scenes. Timely social commentary addresses important issues with nuance. Strong performances deliver believable emotions and connection. |
| ZS | 1. What is the purpose of this review? 2. What is the author's purpose? 3. How do you know? 4. What is the audience? 5. How do you know? |
| TL as TV | The movie was very entertaining. I enjoyed the movie and the characters. It was a great movie to watch. I would recommend it to others. It was a very entertaining movie. |
| TR as TV | Write a positive review for a movie. The positive review should be within 30 words. |
| Random as TV | Thank you for the positive review. It is always nice to hear when someone enjoyed the film. I am glad that you enjoyed the film and that you took the time to write a review. |

Table 12: Sample reviews generated under different settings with Llama3-8B.

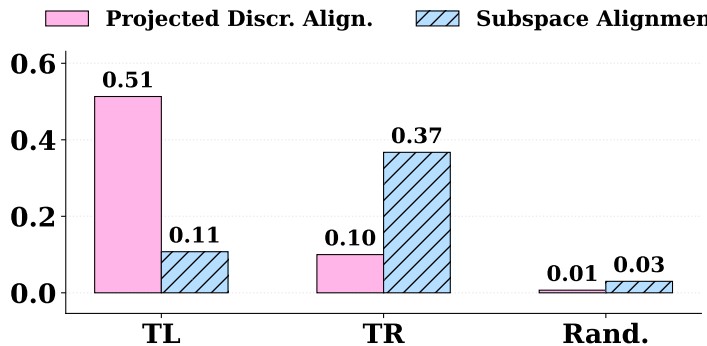

Figure 115: Impact of TL and TR head outputs on hidden states w.r.t. task subspace in Yi-34B.

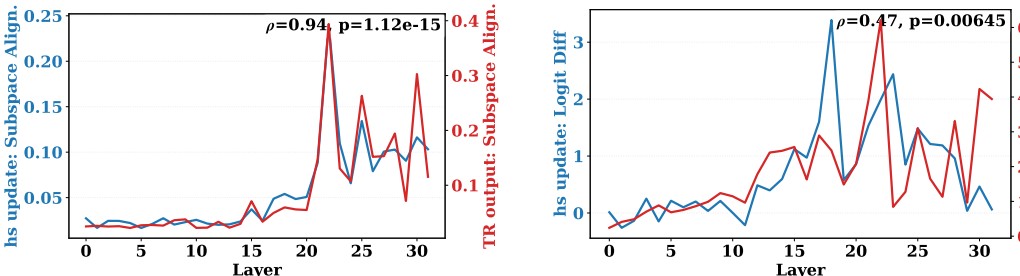

(a) Correlation of hidden state updates with TR heads (subspace alignment).

(b) Correlation of hidden state updates with TL heads (logit difference).

Figure 116: Layerwise correlation of TR and TL head effects on Llama3.1-8B.

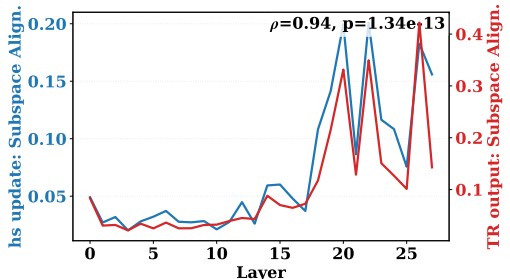 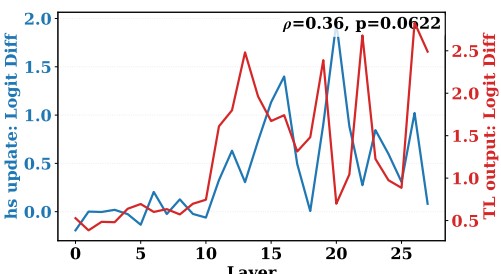

(a) Correlation of hidden state updates with TR heads (subspace alignment).

(b) Correlation of hidden state updates with TL heads (logit difference).

Figure 117: Layerwise correlation of TR and TL head effects on Llama3.2-3B.

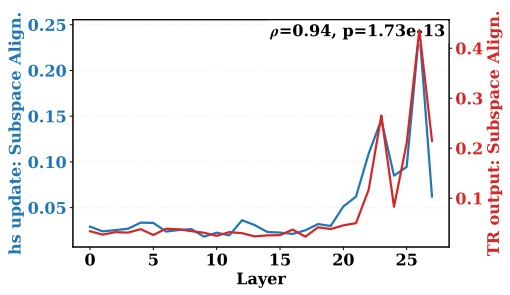 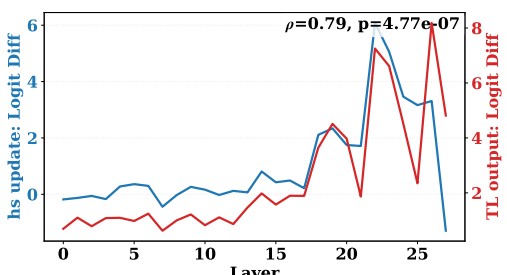

(a) Correlation of hidden state updates with TR heads (subspace alignment).

(b) Correlation of hidden state updates with TL heads (logit difference).

Figure 118: Layerwise correlation of TR and TL head effects on Qwen2-7B.

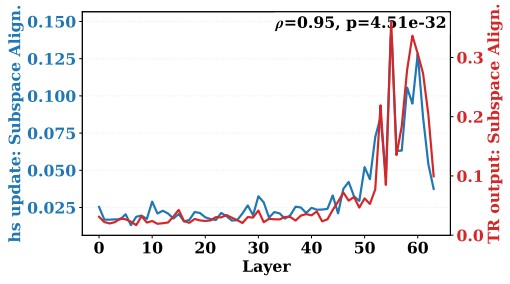 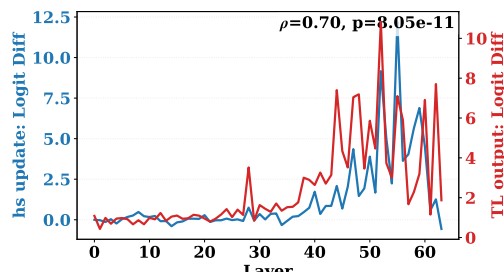

(a) Correlation of hidden state updates with TR heads (subspace alignment).

(b) Correlation of hidden state updates with TL heads (logit difference).

Figure 119: Layerwise correlation of TR and TL head effects on Qwen2.5-32B.

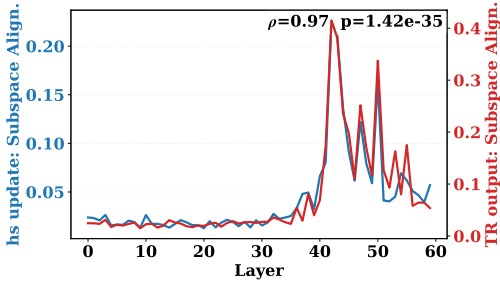

(a) Correlation of hidden state updates with TR heads (subspace alignment).

(b) Correlation of hidden state updates with TL heads (logit difference).

Figure 120: Layerwise correlation of TR and TL head effects on Yi-34B.

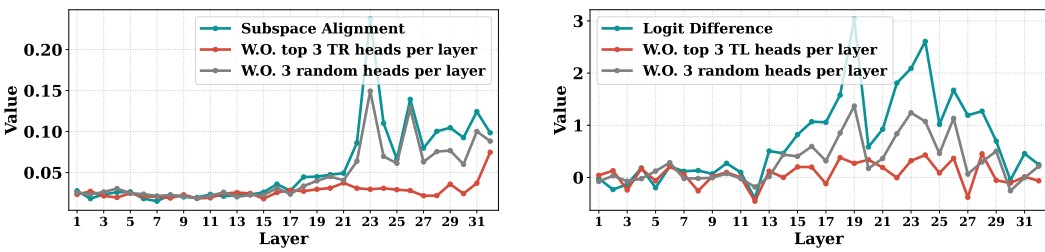

Figure 121: Layerwise ablation of TR and TL heads (top 3 per layer) in Llama3-8B.

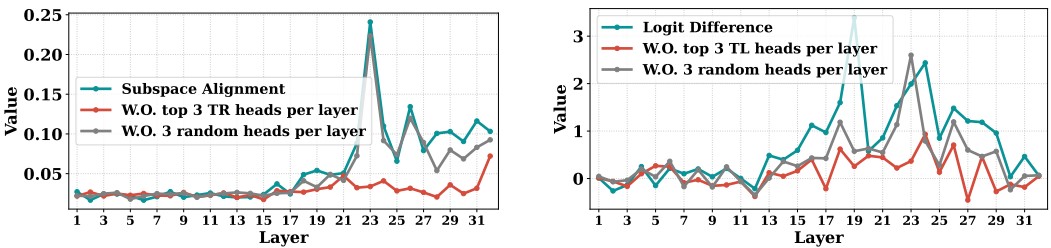

Figure 122: Layerwise ablation of TR and TL heads (top 3 per layer) in Llama3.1-8B.

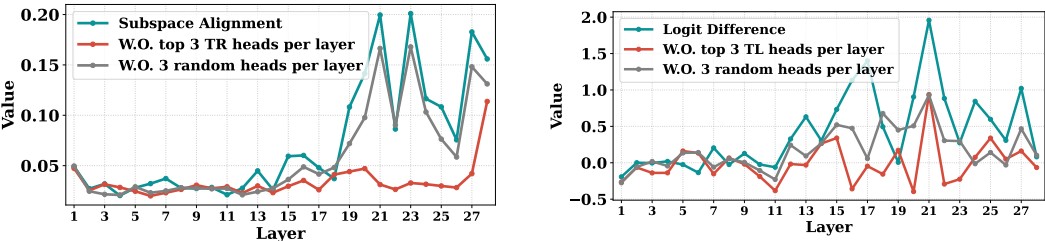

Figure 123: Layerwise ablation of TR and TL heads (top 3 per layer) in Llama3.2-3B.

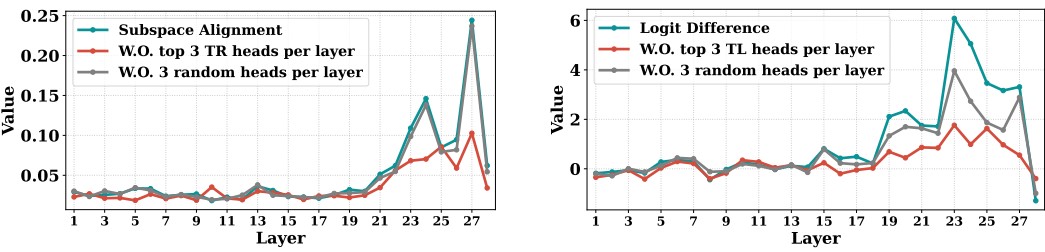

Figure 124: Layerwise ablation of TR and TL heads (top 3 per layer) in Qwen2-7B.

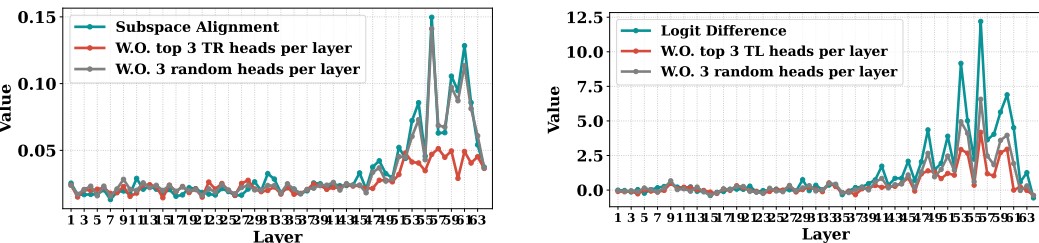

Figure 125: Layerwise ablation of TR and TL heads (top 3 per layer) in Qwen2.5-32B.

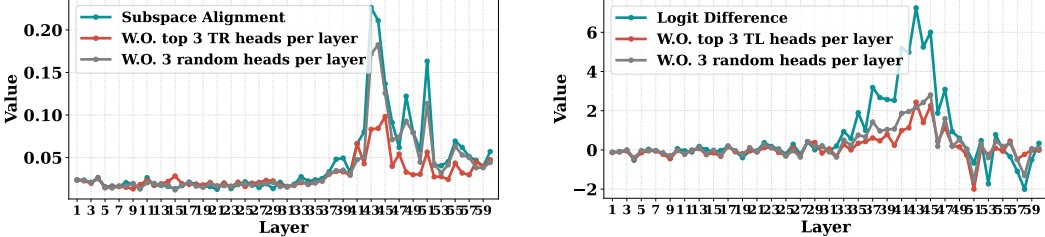

Figure 126: Layerwise ablation of TR and TL heads (top 3 per layer) in Yi-34B.

