# OpenReview forum: "Localizing Task Recognition and Task Learning in In-Context Learning via Attention Head Analysis"
_ICLR.cc/2026/Conference — ICLR 2026 Poster_

### Official Review · Reviewer_YgDM · 2025-10-26

**Soundness:** 4
**Presentation:** 3
**Contribution:** 3
**Rating:** 8
**Confidence:** 4

**Summary:**

This work investigates in-context learning (ICL) through attention-head analysis. It introduces Task Subspace Logit Attribution (TSLA), a framework for identifying Task Learning (TL) and Task Recognition (TR) heads, addressing limitations of prior methods such as Direct Logit Attribution (DLA) and Lieberum et al. (2023). The authors report TR and TL heads across multiple ICL datasets and models. Ablations show that randomizing the outputs of TR or TL heads alters both accuracy and Task Recognition Score, suggesting TR and TL heads operate independently. Steering experiments and geometric analysis indicate that zero-shot ICL failures are largely attributable to task recognition. The authors conclude that TR heads map the logit vector into a task-label subspace, while TL heads rotate it toward the correct label. Overall, the paper offers a comprehensive head-level analysis that reconciles findings across prior work.

**Strengths:**

1. The motivation for TSLA is clear and compelling. In Appendix C, the overlap analysis with induction heads (IH) demonstrates how TSLA addresses limitations of prior approaches; the method seems broadly useful for future head analyses.

2. The experimental suite (steering, ablations, semantic scoring) is well motivated and yields rich findings. In particular, the “task-type dependence of steering effectiveness” experiment (Figure 8) supports the claim that TL heads contribute to parsing the semantic coherence of labels.

**Weaknesses:**

1. The analysis focuses on attention-head outputs; it is unclear how other components (e.g., MLPs) contribute to ICL or how attention patterns differ beyond heads.

2. The scope of ICL considered is narrow, and labels of datasets are mostly binary or few-class. ICL often involves larger label sets or more open-ended tasks (e.g., domain-knowledge QA), which are not covered.

3. While the paper presents many interesting points, some connections to prior work feel scattered. Readers without substantial background may struggle to pinpoint the novel contributions and how the results reconcile earlier findings.

**Questions:**

\textbf{1. Head-level contribution granularity.} As I understand it, ablations and task-vector analyses are applied to the full sets of TR and TL heads selected by a threshold (e.g., 3% or 10%). What are the contributions of individual heads within these sets? Do they contribute equally, additively, or with varying importance? Such analysis could clarify how identified heads work collectively and speak to the robustness of the head identification.

 \textbf{2.Scalability to larger label spaces / open-ended tasks. Can you demonstrate TSLA’s efficacy on tasks with many candidate labels (e.g., math/code proofs, program synthesis with many valid programs) or comparable large-output-space settings?

 \textbf{3.Figure 6 baseline.} In the binary semantic classification setup (positive/negative), random guessing would yield 0.5 accuracy. Why is ICL accuracy without TL below 0.5 in Figure 6?

---

> ### Author Response · Authors · 2025-11-23
>
> **Dear Reviewer YgDM**
>
> Thank you for your comprehensive and considerate assessment of our work. Your well-reasoned critiques and helpful suggestions provide valuable guidance for improving the manuscript. Our point-by-point responses are below.
>
> **W1: Focusing Only on Attention Heads and Lack of Attention-Pattern Analysis**
>
> We focus on attention heads because a substantial body of prior work has shown that they are central to the mechanism of ICL and, more broadly, to many Transformer functionalities [1–4]. In contrast, the literature linking MLP components to ICL is relatively limited, suggesting that attention heads are the more salient locus for explaining ICL behavior, including its TL and TR components.
>
> We also agree that analyzing attention patterns of TL and TR heads would enrich the paper and sharpen mechanistic understanding. As described in Revision Note 4, we investigate attention patterns in Section 4.1 using two metrics that quantify attention allocation: (i) total attention to demonstration labels and (ii) total attention to query tokens. The results show that top TR heads place comparatively more attention on demonstration labels, consistent with their role in recognizing the task label space from demonstrations. In contrast, top TL heads exhibit a local attention pattern concentrated on query tokens. This matches our expectation: to map a query to the correct label, TL heads must first extract the query’s semantic content. Together, these findings clarify how TR and TL heads realize their respective functions through distinct attention behaviors.
>
> **W2 & Q2: Include More Generation Tasks**
>
> We agree that evaluating more realistic generative settings would strengthen the relevance of our work. Accordingly, in Revision Note 1 we add SubjQA in addition to the movie-review generation experiments that produce Figure 8 (Sec. 4.3). SubjQA is a book-review generation task where each example provides a sentiment label and a corresponding labeled review. It is more challenging than the movie-review task in two ways: (1) the label space is substantially richer (e.g., *wrenching*, *anticlimactic*, *captivating*), requiring finer-grained sentiment control than simple positive/negative generation; and (2) the label reviews are human-written, making them more diverse than GPT-generated labels and harder for the model to learn from.
>
> We apply the same task-vector steering protocol to SubjQA (Appendix I.3). Models again generate higher-quality reviews when steered with vectors constructed from TL-head outputs, indicating that TL heads robustly capture text–label associations even in more complex generative scenarios. This is consistent with our conclusion in Sec. 4.3.
>
> **W3: Lack of Connection with Previous Works**
>
> Thank you for encouraging us to more clearly articulate how our findings relate to—and resolve gaps in—prior work. As noted in Revision Note 8, we substantially revised the Introduction and Related Work sections to provide a more principled review of existing literature, clarify where our work sits within it, and explicitly explain how our results address outstanding limitations in previous studies.
>
> **Q1: Fine-Grained Analysis of Head Contributions and Sensitivity to the Head-Selection Threshold**
>
> We agree that the head-selection threshold is an important choice, as it can affect the strength and stability of the observed effects. Please see Revision Note 6 for our analysis. Concretely, we vary the selection percentage from 1% to 10% in 1% increments and repeat the ablation and task-vector steering experiments at each level. The results show that increasing the threshold makes the ablation/steering effects stronger, while preserving the same qualitative patterns observed with our default 3% threshold. This suggests that top TR and TL heads identified at different percentage levels contribute additively (with varying magnitudes), supporting the use of a percentage-based selection rule rather than restricting to a small fixed set of heads.
>
> **Q3: Regarding Figure 6 (Now Figure 8)**
>
> In this figure we report dataset-averaged results. As a consequence, the random-guess baseline is not 0.5 (as in a single binary-classification task), but 40.48%, as noted in the footnote on page 7. This value corresponds to the baseline shown as the red line in Figure 7 (former Figure 5).
>
> We again appreciate the time and attention you devoted to reviewing our manuscript. We welcome further discussion, and if you feel that our responses have addressed your concerns, we would be grateful if you could take this into account when updating your evaluation.
>
> **References**
>
> [1] Zheng, Zifan, et al. "Attention heads of large language models..."
>
> [2] Cabannes, Vivien, et al. "Iteration head: A mechanistic study of chain-of-thought."
>
> [3] Cho, Hakaze, et al. "Mechanism of Task-oriented Information Removal in In-context Learning."
>
> [4] Singh, Aaditya K., et al. "What needs to go right for an induction head?..."

---

> > ### Comment · Reviewer_YgDM · 2025-11-27
> >
> > Thank you for the detailed answers that addressed all of my previous questions. I will maintain my scores to support your paper.

---

> > > ### Author Response · Authors · 2025-11-27
> > >
> > > We are very glad to hear that our rebuttal has addressed all of your concerns. Thank you again for your careful review and for maintaining your score in support of our paper. We sincerely appreciate your time and thoughtful feedback.

---

### Official Review · Reviewer_iScF · 2025-10-27

**Soundness:** 3
**Presentation:** 3
**Contribution:** 2
**Rating:** 4
**Confidence:** 3

**Summary:**

This paper investigates the mechanisms behind in-context learning (ICL) in large language models by bridging the gap between component-level analysis (focusing on attention heads) and the holistic view that decomposes ICL into Task Recognition (TR) and Task Learning (TL). The authors propose a novel method, Task Subspace Logit Attribution (TSLA), to identify specific attention heads responsible for TR and TL. Through various experiments including ablation, steering, and geometric analysis of hidden states, they demonstrate that TR heads primarily function to align model representations with the subspace defined by potential task labels , facilitating recognition of the valid output space, while TL heads operate within this subspace, rotating representations towards the correct specific label for accurate prediction. This framework successfully localizes the distinct TR and TL functions to specialized attention heads and reconciles previous findings on mechanisms like induction heads (identified mainly as TR contributors) and task vectors.

**Strengths:**

- The paper introduces Task Subspace Logit Attribution, a novel set of metrics designed to identify attention heads specialized for TR and TL components of in-context learning, offering a more geometrically grounded approach than prior methods
- The effectiveness and independence of the identified TR and TL heads are thoroughly validated through a comprehensive suite of experiments, including correlation analyses, ablation studies under various input perturbations, steering experiments, and geometric analyses of hidden states. This extensive validation across multiple models and datasets lends strong support to the proposed framework.

**Weaknesses:**

- While the specific TSLA metrics are new and the experimental validation is extensive, the core conceptual approach, i.e.,localizing distinct cognitive functions (TR/TL) to specific attention heads, builds significantly upon prior work that also attributes ICL functions to heads

**Questions:**

- The paper motivates the use of projection onto the task subspace by stating that demonstration labels might be arbitrary and not capture full task semantics. Could the authors elaborate on how projecting onto this specific subspace addresses this issue? Does this implicitly assume that unembeddings of semantically related tokens (e.g., "positive", "favourable", "good") lie close together or within this subspace? If so, could this assumption be further validated, perhaps experimentally, to strengthen the justification for the TR score metric?
- The core proposal involves the TSLA metrics for identifying TR and TL heads. Could the authors further highlight the primary novel contributions of this work beyond the introduction of these specific metrics?
- Could the authors provide a more detailed comparative analysis or specific examples demonstrating how TSLA overcomes the limitations of DLA, such as sensitivity to demonstration label choice and better isolation of true Task Learning effects?

---

> ### Author Response · Authors · 2025-11-23
>
> **Dear Reviewer iScF,**
>
> Thank you for taking the time to provide such a thorough review. We greatly appreciate your detailed feedback. Below, we respond to each of your comments.
>
> **W1 & Q2: Similarity with Existing Studies and Core Contributions**
>
> We understand your concern and would like to clarify our positioning. We analyze attention heads for explaining ICL **because** substantial prior work has established their importance for ICL and other Transformer behaviors [1–4], motivating a head-level treatment.
>
> However, despite this literature, important gaps remain. The TR/TL decomposition of ICL was previously identified via heuristic input perturbations, but to our knowledge no work has localized and analyzed the specific heads responsible for these two components. Many existing ICL–head studies use coarse ablation criteria [5,6], selecting heads whose removal most reduces logits of chosen labels. This reveals **how much** a head matters, but not **how** it contributes—namely, whether its effect operates through TL or TR. Likewise, task-vector approaches typically assess post-steering accuracy without examining how steering changes forward computation, leaving the mechanism unclear.
>
> Our primary contributions are aimed at filling these gaps. We propose the theoretically grounded TSLA method, which enables robust and accurate identification of TL and TR heads—components that were previously only studied via coarse input-level perturbations. We then carry out ablation and steering experiments *within the TL/TR framework* to reveal **how** the discovered heads influence ICL through concrete effects on the model’s forward computation. Concretely:
>
> - We design metrics that separately quantify ablation effects on the TL and TR channels, enabling more fine-grained interpretation.
> - We analyze how task-vector steering changes the geometric evolution of layer-wise hidden-state updates, and connect these observations to the theoretical basis of TSLA.
>
> We thank you for prompting us to present these contributions more clearly. Following your advice, we revised the Introduction and Related Work sections (Revision Note 8) to better highlight our novelty and explain how our experiments address outstanding limitations of prior research.
>
> **Q1: Projection onto a Subspace and Verifying the Subspace Hypothesis**
>
> The key intuition for selecting TR heads based on the projected norm of their outputs onto the task-unembedding subspace is robustness. If heads are selected based on how strongly they promote a *specific label’s* logit (as in DLA), the selection can overfit to that label choice: heads that align well with the unembedding of one label may align poorly after label replacement. In contrast, selecting heads based on alignment with the **entire task-label subspace** (i.e., via the projected norm onto that subspace) reduces this overfitting risk, because the selected heads are likely to be well aligned with *all* candidate labels and therefore more robust to label changes. We refer you to Appendix C for empirical evidence.
>
> We agree the shared-subspace assumption should be verified. As described in Revision Note 2, we test semantically related labels (e.g., *positive/negative/pleasing/disappointing*) by comparing each label’s projected norm onto the subspace spanned by the others against a matched random-token subspace. The semantically related set yields consistently larger projected norms with a highly significant gap, thereby verifying the hypothesis.
>
> **Q3: Robustness of TSLA**
>
> We refer you to Appendix C, where we demonstrate TSLA’s robustness through extensive comparisons with DLA. Briefly, we first identify TR heads on SST-2 using both methods with *positive/negative* as labels. We then swap the labels to *favourable/unfavourable* and ablate the two corresponding TR-head sets, evaluating the impact on accuracy and TR ratio under the new labels. Ablating TSLA-identified TR heads yields substantially larger drops, indicating that TSLA’s identification is more robust to label changes. Appendix C also documents additional advantages of TSLA over DLA, including improved identification of TL heads and the ability to explain the role of induction heads.
>
> Thank you again for your detailed feedback. If our responses have addressed your concerns, we would be grateful if you could reflect that in your updated review.
>
> **References**
>
> [1] Zheng, Zifan, et al. "Attention heads of large language models..."
>
> [2] Cabannes, Vivien, et al. "Iteration head: A mechanistic study of chain-of-thought."
>
> [3] Cho, Hakaze, et al. "Mechanism of Task-oriented Information Removal in In-context Learning."
>
> [4] Singh, Aaditya K., et al. "What needs to go right for an induction head?..."
>
> [5] Crosbie, Joy, and Ekaterina Shutova. "Induction heads as an essential mechanism for pattern matching..."
>
> [6] Kahardipraja, Patrick, et al. "The Atlas of In-Context Learning: How Attention Heads Shape In-Context Retrieval Augmentation."

---

> ### Author Response · Authors · 2025-11-27
>
> **Dear Reviewer iScF**
>
> We are truly grateful for the careful attention and thoughtful feedback you have provided. With the discussion phase nearing its end, we hope our answers have satisfactorily clarified your questions. Please feel free to reach out should you need additional information. Thank you again for your support.
> Wishing you a pleasant day!

---

### Official Review · Reviewer_Rweb · 2025-11-03

**Soundness:** 3
**Presentation:** 3
**Contribution:** 3
**Rating:** 6
**Confidence:** 3

**Summary:**

This paper investigates the mechanistic underpinnings of ICL by identifying and analyzing specific types of attention heads associated with the Task Recognition (TR) and Task Learning (TL) functions of ICL described by previous work. The authors propose a novel method, Task Subspace Logit Attribution (TSLA), to identify attention heads specialized for TR and TL. Through a series of ablation, steering, and representational analyses analyses, the paper argues for a clear functional dissociation, where TR heads are shown to align hidden states with the "task subspace" (enabling the model to recognize the set of possible answers) and TL heads are shown to rotate representations within the task subspace to align the hidden state with the correct label's unembedding (in effect, "learning" the specific text-label mapping for the query). The authors then show that TR heads are strongly correlated with induction heads and are largely task-invariant, while TL heads are highly task specific. The framework provides a unified account of ICL, positioning prior findings on induction heads and task vectors as primarily implementing the TR component of ICL.

**Strengths:**

1. *Novel Method:* The paper introduces Task Subspace Logit Attribution (TSLA), a robust method for identifying specialized heads. Unlike prior approaches, TSLA defines the task via a subspace spanned by label unembeddings. This makes it less sensitive to the specific surface forms of labels (e.g., "positive" vs. "favorable").
2. *Clearly dissociates ICL mechanisms into TR and TL:* The primary strength is the clean dissociation of ICL into TR and TL components and the localization of these functions to distinct, relatively non-overlapping attention heads. The introduction of the "TR Ratio" as an evaluation metric allows the ablation experiments to cleanly separate a failure of label-space recognition from a failure of text-label mapping.
3. *Task-Dependent Steering:* The steering experiments in Section 4.3 demonstrate that TR heads are key for restoring zero-shot classification (where label space recognition is the bottleneck), while TL heads are more effective for generation tasks (which depend on learning a mapping to an open-ended, rather than fixed, label space).
4. *Unifying Framework:* The paper successfully situates prior work on induction heads (IHs) and task vectors within this framework, hypothesizing that they are primarily mechanisms for Task Recognition.

**Weaknesses:**

1. *Clarity and Flow:* The paper's primary weakness is its clarity of presentation. It reads as a collection of distinct analyses (e.g., TSLA definitions, correlation analysis, ablation, steering, geometric verification) rather than a single, cohesive narrative. The transition between the TSLA identification section (Section 3.2) and the geometric verification analysis (Section 4.3) is particularly abrupt. While these analyses are related, the connection could be drawn more clearly to show how the identification method logically predicts the subsequent geometric effects.
2. *Theoretical Justification:* While theorem 1 is interesting, it feels disconnected from the main mechanistic argument around TR heads. The paper would benefit from a more intuitive explanation in the main text of why this probabilistic guarantee for the TR score is important, rather than presenting it as a standalone theoretical result.
3. *Limited Scope of TL Mechanism:* The paper successfully identifies TL heads but provides less detail on how these heads compute the text-label mapping, beyond saying that they "rotate" the hidden state. This is, of course, a difficult research question given the variability of TL heads across tasks, but the mechanism for TL heads are comparatively less specified than TR heads.

**Questions:**

1. *Elaborating the TL Head Computation*: The paper identifies TL heads as task-specific mechanisms that perform an "in-subspace rotation". However, the computation that determines the direction and magnitude of this rotation remains unspecified. Given that TL heads are more task-specific than TR heads, did the authors investigate what these heads attend to in order to compute this mapping? For example, do they consistently attend to specific demonstration examples (e.g., the most recent one, or one semantically similar to the query)? Any initial analysis or speculation here would significantly strengthen the paper's mechanistic account of TL heads.

2. *Reconciling with "Just-in-Time" Task Representations*: Recent work on composite and longer-generation tasks (e.g., Li et al., 2025 https://arxiv.org/abs/2509.04466) suggests that task representations are not monolithic. Instead, they are computed "just-in-time" for minimal task scopes, and their effect fades over multiple generation steps. How might this finding be integrated into the TR/TL framework? Is this "fading" effect best understood as a stable TR mechanism but a fleeting TL mechanism that must be recomputed for each new subtask? This seems like a great way to test whether these ideas presented in the author's paper can provide insight into cases where the traditional task vectors account falls short.

---

> ### Author Response · Authors · 2025-11-23
>
> **Dear Reviewer Rweb**
>
> We sincerely appreciate your careful and insightful review. Your constructive comments and suggestions are invaluable and will meaningfully improve the quality of our manuscript. We address each of your points in detail below.
>
> **Weakness 1&2: Clarity and Flow of Writing, and Lack of Explanation for the Theorem**
>
> Thank you for these suggestions, which have helped us improve both the presentation and overall quality of the manuscript. Please refer to Revision Note 8 for a detailed description of the revisions made in response to your feedback. In summary, we provide a more structured presentation of our findings in the Introduction and Related Work sections, add intuitive explanations of the theorem and clarify how it connects to our experimental results (so that it does not read as a standalone component), and insert connecting sentences between experimental-result paragraphs to highlight how our findings are interrelated.
>
> **Weakness 3 & Question 1: Elaborating the Mechanism of TL Heads**
>
> We appreciate your view that the mechanisms underlying TL and TL heads merit a more detailed exposition. We refer you to Revision Notes 4 and 8. Specifically, we add further explanation of the TL metric and the operations performed by TL heads in Section 3.2. In addition, we conduct experiments in Section 4.1 to examine the distinct attention patterns of TL and TR heads. These results show that top TL heads exhibit a local attention pattern that focuses on query tokens. This is expected: to map a query to the correct label, TL heads must first capture the query’s semantic information. Taken together, these findings shed light on how TL heads implement the functionality of learning the text–label mapping.
>
> **Question 2: Regarding the “Just-in-Time Task Representations” Paper**
>
> Thank you for pointing us to this interesting paper, which enabled us to extend our TL/TR analysis to the composite-task setting studied there. We kindly refer you to Revision Note 5. In particular, we consider the Country–Capital + Antonym composite task from that work, with prompts formatted as “France, big -> Paris, small.” We identify TL and TR heads for the two constituent tasks by computing TSLA metrics for head outputs at the two label positions (for “Paris” and “small,” respectively). We then perform cross-task ablations: for example, we ablate TL/TR heads identified on the Antonym task while the model generates the first-task label using queries like “France, big ->”, and vice versa for the Antonym label using queries like “France, big -> Paris, …”. The results show that ablating TL heads has minimal cross-task effect, whereas ablating TR heads leads to a significant degradation across tasks. We further show that the TR heads identified for the two tasks largely overlap, while the TL heads are task-specific. This suggests that in composite-task settings, separate TL heads are required to learn the mapping for each constituent task and promote the correct task-specific labels, while a shared set of TR heads tracks the full composite label space. This not only matches your intuition about robust TR and more transient TL mechanisms, but also echoes our findings in Figure 6, where TR heads identified across datasets are highly correlated while TL heads remain distinct.
>
> We sincerely thank you once more for your thoughtful review and the time you dedicated to our work. We welcome any further discussion. If you feel that our responses have resolved your concerns, we kindly ask you to take this into account when updating your evaluation.

---

> ### Author Response · Authors · 2025-11-27
>
> **Dear Reviewer Rweb**
>
> Thank you for the considerable effort and insight you have contributed throughout the review process. With the discussion phase nearing its end, we hope our replies have adequately addressed all of your concerns. If anything remains unclear, we would be glad to elaborate further. We are grateful for your thoughtful comments.
> Have a great day!

---

### Official Review · Reviewer_3pGh · 2025-11-05

**Soundness:** 3
**Presentation:** 3
**Contribution:** 3
**Rating:** 8
**Confidence:** 4

**Summary:**

This paper proposes a new method called TSL to identify and localize attention heads that are specialized in Task Recognition (TR) vs. Task Learning (TL). TSL accounts for the label subspace, not just specific tokens, and the logit gap between correct / incorrect labels, which makes it more robust that prior methods in identifying important heads. They discover that TR and TL are handled by distinct heads, and also find that induction heads are largely a subset of TR heads, while task vectors primarily work by injecting the missing TR signal.

**Strengths:**

This paper elegantly provides a mechanistic understanding of ICL properties by providing an attention heads-level explanation to higher-level ICL mechanisms of task recognition and learning.
The TSL method is a clever improvement to find head attributions that are not brittle to the exact choice of label words.
The paper provides thorough empirical evidence to validate the effects of TR and TL heads.
The paper also provides an interesting discussion of the implications of results with prior literature on induction heads and task vectors, that have been widely discussed.

**Weaknesses:**

The analyses are primarily on classification tasks with small label spaces, so it is unclear how this TR/TL decomposition applies to more complex, real-world tasks. However, this critique does not prevent me from positively recommending the paper, as prior works on ICL understanding have used similar families of tasks.

I'd like to see more explanation on how robust the task subspaces are to the demonstration labels used to discover the subspace, perhaps a follow up analysis to test the robustness of TSL by using synonymous but different label sets in the demonstrations and test query.

**Questions:**

Minor: the use of bold text is a bit unusual and excessive, I suggest being more selective in bolding text.

Q: TR heads appearing deeper than TL feels surprising to me intuitively. Out of curiosity, do you have hypotheses why that may be the case, that recognizing the task comes after learning it?

---

> ### Author Response · Authors · 2025-11-23
>
> **Dear Reviewer 3pGh**
>
> Thank you for your insightful review of our paper. We greatly appreciate the thoroughness of your comments and the helpful suggestions, which will certainly strengthen our work. Below, we provide our responses to the points you have kindly raised.
>
> **Weakness 1: Experiments on more complex generation tasks**
>
> We fully concur with your remark that experiments on more generative task settings that better match real-world scenarios will improve the relevance and significance of our work. We kindly refer you to Revision Note 1, where we additionally include the SubjQA task on top of the movie review generation task previously presented in Section 4.3. SubjQA is a book review generation task: each data point contains a sentiment label and a labeled book review reflecting that sentiment. However, it differs from our movie review generation task in two critical ways. **1)** The sentiment labels are far more diversified, including labels such as *wrenching*, *anticlimactic*, *captivating*, and others, making it a much more challenging task since the model needs to grasp fine-grained sentimental nuances while generating reviews. In contrast, the movie review generation task only requires generating positive/negative reviews. **2)** The label reviews are written by humans, which makes them more variable than the GPT-generated movie review labels, thereby making it harder for the model to learn the association between the sentiment and the review.
>
> We conduct the same task-vector steering as in Section 4.3 on this dataset in Appendix I.3. The results indicate that models still generate better reviews when steered with task vectors constructed from TL-head outputs. This reflects that the capability of our TL heads to grasp the association between demonstration texts and labels holds in more complicated generation scenarios, which validates our conclusion in Section 4.3.
>
> **Weakness 2: Demonstrating the robustness of TSLA and its advantage over DLA**
>
> We would like to bring your attention to Appendix C, where we provide extensive comparisons between our TSLA and DLA in terms of their ability to identify TL and TR heads and their robustness to changes in demonstration labels. Our findings are threefold. **1)** The TL and TR heads identified using DLA are largely repetitive and overlap heavily with the TR heads identified using TSLA, indicating that DLA cannot effectively identify TL heads. **2)** Neither the TL nor TR heads identified using DLA overlap significantly with induction heads; thus, DLA cannot explain the significance of induction heads for ICL. In contrast, using TSLA we find that induction heads are highly similar to TR heads. **3)** Under a demonstration-label shift on the SST-2 dataset from positive/negative to favourable/unfavourable, ablating TR heads identified using DLA leads to far smaller drops in accuracy and TR ratio than ablating their counterparts identified using TSLA, which demonstrates the robustness of TSLA-based identification to label shifts.
>
>
> **Question 1: Reduce the use of bold texts**
> Thank you for your suggestion, which helps us improve the visual effect of our paper. We refer you to Revision Note 7, where we explain how we reduced the use of bold text.
>
> **Question 2: Why TL heads appear in earlier layers than TR heads**
> We understand your concern that this finding appears counterintuitive given the literal meanings of task learning and task recognition. This is because, in our TSLA framework, we define task learning and task recognition based on how they affect the model’s forward computation and influence token logits at the output. From this logit-based perspective, TL (and therefore TL-head outputs) increases the logit margin between correct and incorrect task labels to promote prediction of the correct label, whereas TR (and TR-head outputs) increases the logit margin between all task labels and the remaining unrelated labels, thereby causing the model to recognize the task-label set and predict solely from it. We refer you to Revision Note 3, where we discuss this phenomenon and provide experiments to verify it.
>
> Thank you again for the care and time you’ve invested in reviewing our paper. We appreciate the opportunity to continue the dialogue. If you find that we have adequately addressed your concerns, we would be grateful if you could reflect that in your updated assessment.

---

> ### Author Response · Authors · 2025-11-27
>
> **Dear Reviewer 3pGh**
>
> We sincerely appreciate the time and thought you have devoted to reviewing our manuscript. As the discussion period comes to a close, we hope our responses have fully resolved your questions. Please don’t hesitate to reach out if any further clarification is needed. Thank you again for your valuable feedback.
> Wishing you a wonderful day!

---

> > ### Comment · Reviewer_3pGh · 2025-11-27
> >
> > Thank you for your clarifications, I appreciate it! I will maintain my (already high) original score.

---

> > > ### Author Response · Authors · 2025-11-28
> > >
> > > Thank you for your considerate follow-up and for the time you dedicated to reviewing our work. We greatly appreciate your positive evaluation and your decision to uphold your scores. Your feedback has been instrumental in helping us strengthen the clarity and overall quality of the paper.
> > >
> > > We sincerely appreciate your continued support.

---

### Author Response · Authors · 2025-11-23
**Revision Note**

Dear PCs, SACs, ACs, and Reviewers,

Thank you for your valuable suggestions. **To address the comments raised by the reviewers, we have made the following adjustments to the paper.** (Revisions are highlighted in olive.)

1. **Reviewer 3pGh, YgDM: Include more open-ended generation tasks (Appendix I.3)**
   We agree with the reviewers’ suggestion that experiments on more complex generation tasks—beyond our movie-review generation task—would further increase the paper’s practical significance. We therefore include the **SubjQA** dataset, a review-generation task with richer sentiment requirements and real human-written label reviews (whereas the movie task uses GPT-generated labels) in Appendix I.3. Experiments support the effectiveness of TL-head task vectors, consistent with our claim in Section 4.3.

2. **Reviewer iScF: Verifying that semantically related tokens’ unembeddings lie in common subspaces (Appendix F)**
   We concur that it is crucial to verify the assumption that semantically related tokens’ unembeddings lie in common subspaces, since this underpins our TSLA framework. We conduct experiments for all models (Appendix F), confirming the assumption holds across models.

3. **Reviewer 3pGh: Explanation for why TL heads occur in earlier layers than TR heads (Appendix G.7)**
   We understand the concern that heads responsible for task learning appearing earlier than those governing task recognition may seem counterintuitive. We provide further explanations, justifications, and experimental results in Appendix G.7 to clarify this observation.

4. **Reviewer Rweb, YgDM: Analyzing attention patterns of TL and TR heads to elucidate their mechanisms (Section 4.1)**
   We appreciate the comment that analyzing attention patterns is a clear way to understand how TL and TR heads achieve their functions. We add this analysis in Section 4.1; see Figures 4 and 5 which indicate that TR heads attend more to demonstration labels to recognize the task label space, while TL heads attend more to query tokens to facilitate the semantic matching of query to its correct label, thereby advancing TR and TL, respectively.

5. **Reviewer Rweb: Reconciling with the “Just-in-Time” Task Representation paper (Appendix H.4)**
   We thank the reviewer for bringing this paper to our attention. It shares a common theme with our research and allows us to extend our TR/TL framework to composite-task ICL settings studied there. We present a detailed discussion and corresponding experiments in Appendix H.4 to investigate how TL and TR operate for composite tasks.

6. **Reviewer YgDM: Granularity of heads’ influence and ablation studies for the head-selection threshold (Appendix G.1)**
   We agree on the importance of examining how the threshold percentage affects the joint influence of selected heads. In Appendix G.1, we sweep the threshold from 1% to 10% and repeat all ablations and experiments, which confirm that our conclusions hold regardless of the specific threshold choice.

7. **Reviewer 3pGh: Reduce the use of bold texts**
   We agree that our previous use of bold text was excessive. We revised the paper so bold font is used only when important concepts or metrics are first introduced.

8. **Reviewer Rweb, iScF, YgDM: Revise the writing and presentation of the paper**
   We are deeply grateful for the reviewers’ suggestions, which enabled substantial improvements to organization and clarity. Specifically, we made the following adjustments:

   - **More clearly presenting the core contributions and their relationship to existing literature (Reviewer iScF, YgDM)**
     We revised the Introduction and Related Works sections to (1) add a concise summary of core contributions, (2) reorganize related work, and (3) state how our experiments address prior limitations.

   - **Providing an intuitive explanation for Theorem 1 (Reviewer Rweb)**
     We add a clearer intuitive explanation of the theorem and how it supports the TR score metric in Lines 192–207 of Section 3.2.

   - **Elaborating the TL heads’ mechanism (Reviewer Rweb)**
     We provide further explanation of TL heads’ computation mechanism in Lines 216–223, drawing on Figure 1(C) and additional attention-pattern evidence in Section 4.1.

   - **Strengthening the connections between theory and experiments, and among experiments (Reviewer Rweb)**
     We revise the Introduction and Related Works to clarify how TSLA motivates our experimental design. We also add a roadmap paragraph (Lines 228–233) at the end of Section 3 explaining how TSLA’s construction predicts our findings and how we validate these predictions. Finally, at the beginnings of Sections 4.1, 4.2, and 4.3, we add brief bridge sentences that make the logical progression across experiments explicit.

We sincerely appreciate the time and consideration you devoted to reviewing our manuscript. Your thoughtful feedback has substantially strengthened the paper.

Sincerely,
The Authors

---

### Author Response · Authors · 2025-12-02
**Summary of Reviewer-Author Discussions: Part 1**

**Dear PCs, SACs, ACs, and Reviewers,**

Thank you very much for your valuable contributions to our work. Given the extremely unconventional status this year and the length of our detailed rebuttal and revision, to assist the newly assigned AC in decision making, we provide below a summary of the key points from the reviews and the reviewer–author discussions.

# **Part 1 — Strengths and Positive Reviewer Feedback**

Across the initial reviews and the discussion phase, reviewers expressed strong agreement that the paper makes a meaningful mechanistic contribution to understanding in-context learning, as reflected by the high average rating the paper receives (6.5). Specifically, two reviewers gave high initial ratings (8), and both further explicitly confirmed during the discussion that our rebuttal fully resolved their concerns and that they would maintain their strong scores.

## **1. Insightful framework linking head-level mechanisms to high-level ICL functions**

All four reviewers highlighted that the proposed **Task Subspace Logit Attribution (TSLA)** provides an effective and principled way to connect attention-head-level computations with the high-level ICL mechanisms of **Task Recognition (TR)** and **Task Learning (TL)**. They emphasized that TSLA:

- robustly identifies TR and TL heads using label-subspace geometry,
- is less brittle to label surface forms than past methods,
- and offers a clear mapping from microscopic head behavior to macroscopic ICL functions.

Reviewer 3pGh praised TSLA as an elegant mechanistic explanation of ICL.
Reviewer Rweb emphasized the clarity of the TR/TL interpretation enabled by TSLA.
Reviewer iScF noted the geometric grounding as a principled advance.
Reviewer YgDM viewed the approach as broadly useful for future head-level analyses.

## **2. Strong empirical evidence for a functional TR/TL dissociation**

Reviewers agreed that the paper presents compelling and comprehensive evidence that TR and TL correspond to distinct sets of heads that implement different pieces of the ICL process:

- **TR heads** align representations with the task-label subspace (recognizing the valid output space).
- **TL heads** rotate representations within this subspace toward the correct label (performing the text–label mapping).

They found that this dissociation:

- clarifies the function of induction heads (as TR-related),
- explains the behavior of task vectors (injecting TR information),
- and unifies several previously disconnected findings in the mechanistic ICL literature.

Reviewer Rweb described this dissociation as the paper’s primary strength.
Reviewer 3pGh and YgDM emphasized that it provides a coherent mechanistic account that prior work lacked.

## **3. Extensive and carefully designed experimental validation**

Reviewers consistently emphasized that the empirical suite is broad, rigorous, and unusually comprehensive for mechanistic interpretability. They highlighted that:

- **Ablation and steering** cleanly separate TR vs. TL functionality.
- **Correlation and semantic-scoring** analyses show TR heads capture label-space structure, while TL heads capture text–label alignment.
- **Geometric analyses** confirm distinct representational effects predicted by TSLA.
- **Attention-pattern studies** (added in revision) reveal that TR heads focus on demonstration labels and TL heads on query tokens.

Overall, reviewers found the results rich, interpretable, and robust across models, datasets, and perturbations, reinforcing the credibility of the proposed mechanism.

---

## **4. Effective unification and clarification of prior literature**

Reviewers agreed that the TR/TL framework provides a coherent explanation that ties together previously scattered findings. They noted that it:

- accounts for induction heads as TR-related,
- explains task vectors as injecting TR information,
- and contextualizes prior representational results (e.g., label-subspace geometry) within a unified mechanism.

This integrative perspective was viewed as an important conceptual contribution that clarifies several earlier observations in ICL research.

---

## **Positive responses during discussion**

Reviewers **3pGh** and **YgDM** explicitly stated that our rebuttal resolved their concerns and confirmed that they would **retain their strong scores of 8**.
The other two reviewers did not post follow-up messages but **posed no new questions**, leaving their assessments unchanged. No reviewer expressed remaining concerns after the discussion period.

---

> ### Author Response · Authors · 2025-12-02
> **Summary of Reviewer-Author Discussions: Part 2**
>
> # **Part 2 — How We Addressed Reviewers’ Concerns**
>
> We revised the paper extensively to address all reviewer comments through additional analyses, expanded experiments, and improved exposition. Below we summarize how each major concern was resolved.
>
> ## **1. Need for richer, more open-ended generation tasks (3pGh, YgDM)**
> We added experiments on **SubjQA**, a challenging review-generation task with diverse human-written sentiment labels.
> Results (Appendix I.3) show that **TL-head task vectors continue to improve generative performance**, validating our conclusions beyond simple positive/negative review generation.
>
> ## **2. Verifying the semantic-unembedding subspace assumption (iScF)**
> We conducted cross-model experiments (Appendix F) confirming that semantically related unembeddings cluster in shared subspaces, supporting TSLA’s theoretical foundation.
>
> ## **3. Clarifying why TL heads appear earlier than TR heads (3pGh)**
> Appendix G.7 now provides theoretical justification and empirical evidence showing that TL and TR differ in *logit-level functions*:
> TL increases the margin **within** the task subspace (label mapping), while TR increases the margin **between** task vs. non-task labels (task recognition), explaining the observed layer ordering.
>
> ## **4. Providing mechanistic insight through attention patterns (Rweb, YgDM)**
> We added attention-pattern analyses in Section 4.1 (Figures 4 and 5):
> - **TR heads** attend more to **demonstration labels**, enabling recognition of the task-label space.
> - **TL heads** attend more to **query tokens**, supporting semantic extraction for label mapping.
>
> This directly illuminates how TR and TL heads implement their respective roles.
>
> ## **5. Integrating TSLA with “Just-in-Time Task Representations” (Rweb)**
> Appendix H.4 extends our framework to composite tasks (Country–Capital + Antonym).
> Findings show:
> - TL heads are **task-specific**,
> - TR heads are **shared across subtasks**,
> - ablating TR heads disrupts both subtasks.
> This demonstrates that our TR/TL decomposition generalizes beyond basic ICL settings.
>
> ## **6. Robustness and superiority of TSLA over DLA (3pGh, iScF)**
> We refer reviewers to **Appendix C**, which provides extensive comparisons showing that TSLA is both **more robust** and **more accurate** than DLA in identifying TR and TL heads.
> Key findings include:
> - TSLA reliably identifies TL heads, whereas DLA predominantly recovers TR-like heads.
> - TR heads identified by TSLA align strongly with induction heads; DLA does not capture this structure.
> - Under label-surface shifts (e.g., *positive/negative* → *favourable/unfavourable*), ablating TSLA-identified TR heads produces substantially larger drops in accuracy and TR ratio, demonstrating **significant robustness to label changes**.
>
> These results strengthen the theoretical and empirical foundation of TSLA relative to prior attribution methods.
>
> ## **7. Assessing threshold robustness and head-level contribution granularity (YgDM)**
> Appendix G.1 sweeps the head-selection threshold from 1%–10%.
> All experiments preserve the same qualitative conclusions, demonstrating **robustness** and showing that additional identified heads contribute additively.
>
> ## **8. Improving clarity, organization, and exposition (Rweb, iScF, YgDM, 3pGh)**
> We substantially revised writing throughout the paper:
>
> - reduced unnecessary bold text (Revision Note 7),
> - reorganized Introduction and Related Works to highlight core contributions (Revision Note 8),
> - added an intuitive explanation for Theorem 1 (Lines 192–207),
> - elaborated on TL-head mechanisms (Lines 216–223),
> - added a roadmap linking theory to experiments (Lines 228–233),
> - and inserted connective paragraphs across Sections 4.1–4.3.
>
> These changes significantly improved clarity and narrative coherence.
>
> ---
>
> We sincerely thank all reviewers for their thoughtful feedback and valuable suggestions. Their input has strengthened the paper both conceptually and empirically. We hope this summary assists the AC in evaluating our revisions.
>
> Sincerely,
> **The Authors**

---

### Meta-Review · Area_Chair_uBkt · 2026-01-07

**Summary:**

1. The evaluation focuses too much on small-label classification and may not generalize to more complex or real-world / open-ended generation tasks or large label spaces.
2. TSLA’s robustness and advantage over prior attribution methods (especially DLA) needs clearer evidence, including robustness to label-word changes.
3. The paper’s narrative/positioning is hard to follow, and the novel contributions (and connections to prior work) are not clearly presented.
4. Theoretical results (e.g., Theorem 1) feel disconnected and need an intuitive explanation of why they matter for the mechanistic story.
5. Mechanistic understanding of TL heads is underdeveloped (beyond “rotation in the task subspace”).
6. The scope is limited to attention-head outputs; contribution of other components (e.g., MLPs) and richer attention-pattern analysis is missing.
7. Granularity/sensitivity: results rely on selecting sets of heads by a threshold; reviewers want to know whether effects are stable across thresholds and how individual heads contribute.
8. Conceptual novelty concern: “localizing TR/TL to heads” may be too close to prior work; authors should more explicitly state what is new beyond introducing TSLA metrics.
9. Implicit assumption about task subspace.

**Reviewer Concerns:**

The reviewer concerns are mostly addressed by the rebuttal through clarifications and additional experiments.

**Reviewer Scores:**

1. The Reviewer 3pGh's score would remain 8 as made clear by the reviewer in the follow-up response.
2. The Reviewer Rweb's score would remain 6 or potentially be increased as the concerns have been addressed.
3. The Reviewer iScF's score would increase to 5 or 6 as the concerns have been addressed.
4. The Reviewer YgDM's score would remain 8 as made clear by the reviewer in the follow-up response.

---

### Decision · Program_Chairs · 2026-01-26

Accept (Poster)